# FLIP2: Expanding Protein Fitness Landscape Benchmarks for Real-World Machine Learning Applications

**Kieran Didi** [1 2]   **Sarah Alamdari** [3]   **Alex X. Lu** [3]   **Bruce Wittmann** [4]   **Kadina E. Johnston** [5]   **Ava P. Amini** [3]
**Ali Madani** [6]   **Maya Czeneszew** [1]   **Christian Dallago** [1 7 8 9]   **Kevin K. Yang** [3]

## Abstract

Machine learning methods that predict protein fitness from sequence remain sensitive to changes in data distributions, limiting generalization across common conditions encountered in protein engineering. Practically, protein engineers are thus left wondering about the effective utility of ML tools. The FLIP benchmark established protocols for testing generalization under some domain shifts, but it was limited to measurements of thermostability, binding, and viral capsid viability. We introduce FLIP2, a protein fitness benchmark spanning seven new datasets, including enzymes, protein-protein interactions, and light-sensitive proteins, as well as splits that measure generalization relevant to real-world protein engineering campaigns. Evaluating a suite of benchmark models across these datasets and splits reveals that simpler models often matched or outperformed fine-tuned protein language models on FLIP2, challenging the utility of existing transfer learning techniques. Provenance for all datasets has been recorded and we redistribute all data CC-BY 4.0 to facilitate continued progress.

[1]NVIDIA [2]Department of Computer Science, University of Oxford, Oxford, UK [3]Microsoft Research, Cambridge, MA, 02142, USA [4]Microsoft Office of the Chief Scientific Officer, Redmond, WA, 98052 [5]California Institute of Technology, Pasadena, CA, 91125, USA [6]Profluent Bio, Berkeley, CA, USA [7]Department of Cell Biology, Duke University, Durham, NC, 27705, USA [8]Department of Biostatistics and Bioinformatics, Duke University, Durham, NC, 27705, USA [9]DiscoveryAI, Duke University, Durham, NC, 27705, USA. Correspondence to: Christian Dallago <cdallago@nvidia.com>, Kevin K. Yang <yang.kevin@microsoft.com>.

*Proceedings of the 43rd International Conference on Machine Learning*, Seoul, South Korea. PMLR 306, 2026. Copyright 2026 by the author(s).

## 1. Introduction

Machine learning has revolutionized our ability to predict, design, engineer, and understand protein sequence, structure, and function (Wu et al., 2021; Bordin et al., 2023; Ferruz et al., 2023; Nambiar et al., 2025). In particular, machine learning methods that predict protein fitness from sequence have enabled the engineering of optimized proteins with fewer wet-lab experiments compared to traditional directed evolution approaches (Romero et al., 2013; Yang et al., 2019; Wu et al., 2019; Wittmann et al., 2021; Jiang et al., 2024). The advance of machine learning methods relies on the development and maintenance of benchmarks that reflect the realities of the applications in which those methods will be deployed. Fitness Landscape Inference for Proteins (FLIP) (Dallago et al., 2021) took early steps toward establishing such benchmarks for machine-learning-assisted protein engineering, providing train-test-validation splits that mimic wet-lab engineering.

However, FLIP was not comprehensive, and several critical applications of machine-learning-assisted protein engineering continue to lack benchmark coverage. For example, the FLIP datasets covered thermostability, binding, and viral capsid viability, with limited representation of the enzymatic functions that are central to many biotechnology applications (Miller et al., 2022). Furthermore, the FLIP splits did not capture key practical constraints found in protein engineering. For instance, engineering campaigns often possess abundant data for one protein but need to optimize a homologous target with little to no data – a scenario requiring generalization across wild-type backgrounds (Sela et al., 2024). A wild type protein is a starting point for engineering: many engineering campaigns consist of finding mutations that improve the fitness of a wild type or combining segments of different wild-type proteins. Similarly, initial screens often saturate specific regions (e.g., active sites), requiring models to predict effects in unobserved structural contexts or distal positions to unlock novel function (Johnston et al., 2024).

Here, we present FLIP2, which updates FLIP with new datasets, split strategies, and baseline evaluations. FLIP2 introduces seven new sequence-fitness datasets that significantly broaden the functional diversity of benchmarked

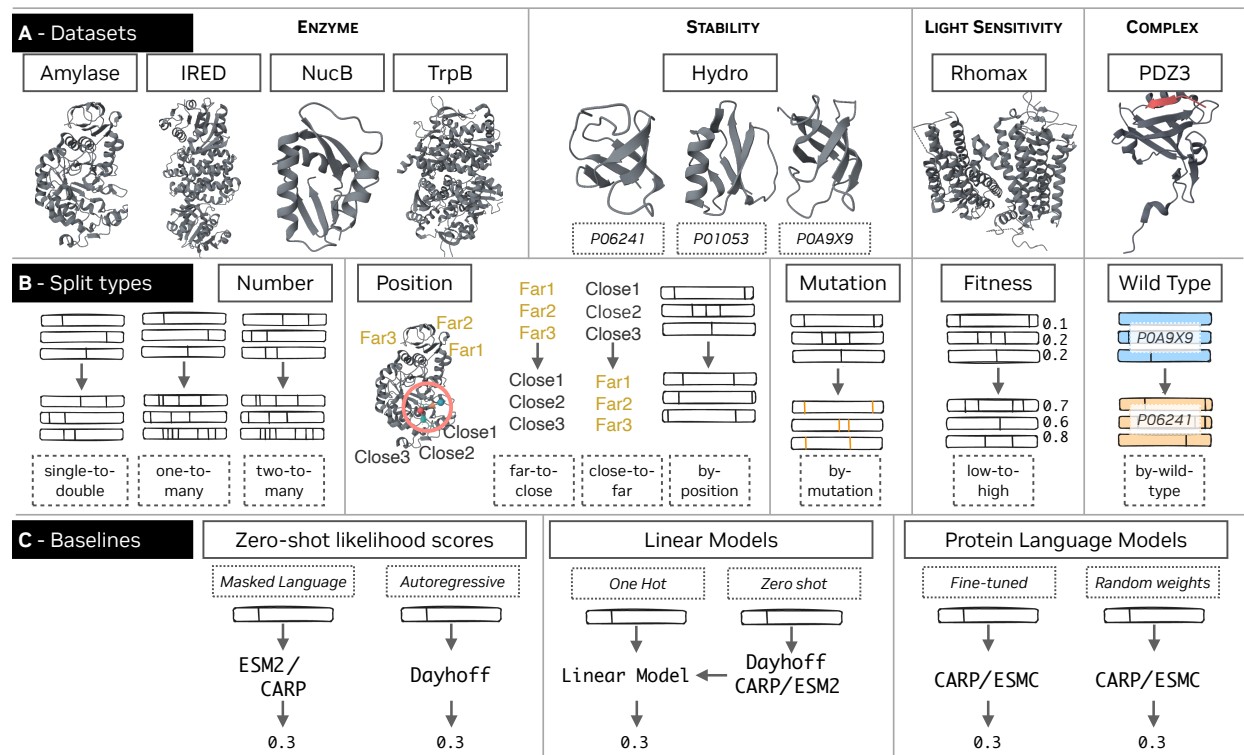

*Figure 1.* **The FLIP2 benchmark**. (A) The FLIP2 datasets. Solid boxes indicate the name of the dataset. Dashed boxes indicate individual protein identifiers. (B) The FLIP2 splits (dashed boxes) and split types (solid boxes). (C) Baseline sequence-to-fitness prediction methods.

proteins (Figure 1A). These new datasets include measurements of enzymes acting on small molecules and nucleic acids, a light-sensitive protein, and protein-protein interactions (PPIs) involving intrinsically disordered regions. Beyond mutation count and fitness-based splits, FLIP2 implements splits designed to mimic additional common phases in an engineering campaign, including generalization across different starting sequences, to unseen mutations, and to mutations at new positions (Figure 1B). We then evaluated zero-shot protein language model (pLM) sequence-likelihood scores, ridge-regression baselines, and fine-tuned pLMs on these splits (Figure 1C). Performance is measured by Spearman's $\rho$ and normalized discounted cumulative gain (NDCG) on held-out test sets. These baselines confirmed that the FLIP2 splits are more challenging than random splits with the same number of training examples. Further, simpler models often matched or outperformed fine-tuned protein language models, challenging the utility of existing transfer learning techniques for fitness prediction.

## 2. Related Work

### 2.1. Protein Fitness Prediction Benchmarks

Early efforts to evaluate machine learning models of protein sequence-fitness landscapes focused on small, task-specific datasets that often lacked standardized evaluation proto-

cols (Bloom, 2014; Fox et al., 2007; Romero & Arnold, 2009; Romero et al., 2013). TAPE (Tasks Assessing Protein Embeddings) (Rao et al., 2019) marked an important milestone by providing a multi-task benchmark for protein sequence understanding, although it included only limited fitness prediction tasks. The FLIP benchmark (Dallago et al., 2021) addressed these limitations by curating datasets and splits specifically for protein engineering applications. FLIP included three primary fitness landscapes: AAV capsid variants with viral viability measurements, GB1 domain variants with both stability and binding data, and thermostability measurements across multiple protein families. Importantly, FLIP introduced split strategies designed to test model performance in the low-resource and extrapolative settings typical of protein engineering campaigns. FLIP served two important functions: (1) it provided easy-to-use and -measure benchmarks, and (2) it better represented what a bench scientist would realistically face during a protein engineering campaign.

Following FLIP, several benchmarks addressed specific aspects of protein engineering. PEER (Xu et al., 2022) expanded multi-task evaluation to include protein-protein interactions and protein-ligand binding. More recently, specialized benchmarks have emerged for enzyme reaction classification (Yang et al., 2024a; Hua et al., 2024), antibody optimization (Chungyoun et al., 2024; Liu et al., 2025; Zhao

*Table 1.* The FLIP2 datasets and splits

| Dataset | $n_{\text{total}}$ | Split | Split type | $n_{\text{train}}$ | $n_{\text{validation}}$ | $n_{\text{test}}$ |
|---|---|---|---|---|---|---|
| Alpha Amylase (Amylase) | 3706 | one-to-many | number | 412 | 77 | 3217 |
| | | close-to-far | position | 1426 | 356 | 1924 |
| | | far-to-close | position | 1539 | 385 | 1782 |
| | | by-mutation | mutation | 2403 | 601 | 702 |
| Imine Reductase (IRED) | 17143 | two-to-many | number | 3746 | 662 | 4178 |
| Nuclease B (NucB) | 55760 | two-to-many | number | 6982 | 2027 | 48304 |
| $\beta$-subunit of Tryptophan Synthase (TrpB) | 228298 | one-to-many | number | 304 | 76 | 227918 |
| | | two-to-many | number | 8633 | 2158 | 217507 |
| | | by-position | position | 31557 | 7889 | 188852 |
| Hydrophobic Core (Hydro) | 24935 | three-to-many | number | 1160 | 290 | 23485 |
| | | low-to-high | fitness | 9974 | 2493 | 12468 |
| | | to-P06241 | wild-type | 11970 | 2993 | 9972 |
| | | to-P0A9X9 | wild-type | 13148 | 3287 | 8500 |
| | | to-P01053 | wild-type | 14778 | 3694 | 6463 |
| Rhodopsin (Rhomax) | 884 | by-wild-type | wild-type | 584 | 116 | 184 |
| PDZ3 | 734 | single-to-double | number | 124 | 31 | 579 |

et al., 2024), uncertainty quantification (Greenman et al., 2025), and cross-family generalization (Groth et al., 2023). However, none of these emphasize predicting the sequence-fitness relationship across the full breadth of protein function and the types of extrapolation commonly encountered in protein engineering.

ProteinGym (Notin et al., 2023) dramatically expanded the scale and scope of protein fitness benchmarking with over 250 multiplexed assays encompassing more than 2.7 million mutated sequences across diverse proteins. It also established evaluation frameworks for both zero-shot and supervised learning. The benchmark revealed significant performance variations across protein families and highlighted the importance of model architecture, training data, and protein structure input. ProteinGym's emphasis on including as many mutational datasets as possible results in the majority of datasets only containing single- and double- mutants collected via deep mutational scanning experiments.

### 2.2. Protein Language Models for Fitness Prediction

Protein language models (pLMs) that learn patterns from unlabeled protein sequences have transformed computational protein engineering by enabling increasingly accurate predictions the fitness effects of protein variation (Freschlin et al., 2022). Early approaches leveraged subsequence co-occurence patterns in large sequence datasets (Kimothi et al., 2016; Hopf et al., 2017; Yang et al., 2018). More recently, modern pLMs were shown to learn evolutionary and structural information by training on unlabeled sequences and structures (Heinzinger et al., 2019; Rao et al., 2019; Bepler & Berger, 2021; Rives et al., 2021; Zhang et al., 2024). The likelihoods from pLMs have been shown to correlate with

mutation effects on fitness (Meier et al., 2021) and with the probability that a generated enzyme will be functional (Johnson et al., 2025). Several works demonstrated that while pLM output representations can be effective inputs to supervised sequence-fitness models, fine-tuning usually outperforms using pretrained representations (Yang et al., 2024b; Schmirler et al., 2024; Marquet et al., 2024; Li et al., 2024). However, the best performance on small labeled datasets has come from training linear regression models on the zero-shot likelihood scores and from one-hot representations of sequences (Hsu et al., 2022). Informed by this prior research demonstrating what baselines are necessary to demonstrate the usefulness of pLMs, we focus on evaluating the utility of zero-shot likelihoods, linear models, and fine-tuned pLMs in extrapolative regimes common in protein engineering.

## 3. The FLIP2 Datasets and Splits

### 3.1. Split Types

FLIP2 includes seven datasets and 16 total splits (Figure 1 and Table 1) which are grouped into five categories of generalization:

**Number**: train on variants with fewer mutations and test on variants with more mutations. This tests for the ability to extrapolate to a larger number of mutations, a common need for protein engineering applications, as data necessarily become sparser as the number of mutations increases.

**Position**: train and test on variants with mutations in different positions in the sequence. This tests for the ability to generalize to previously unperturbed positions, which may be targeted in subsequent engineering rounds.

**Mutation**: train and test on variants with different unique

mutations; mutations to different amino acids at the same position may be split across train and test.

**Fitness**: train on variants with lower fitness and test on variants with higher fitness. This simulates protein engineering campaigns that optimize protein function, as proteins later in the campaign may have higher fitness.

**Wild Type**: train and test on variants with different wild-type sequences or structural scaffolds. This tests for the ability to generalize mutational effects across wild-type sequences, and is critical especially in cases where there may only be a small number of characterized mutants per wild type.

### 3.2. Datasets

Below, we detail the FLIP2 datasets and how the splits were applied to them.

#### 3.2.1. ALPHA AMYLASE (AMYLASE)

Alpha amylases catalyze the breakdown of starches and are used in detergents to remove starch stains. We adapt a dataset by van der Flier et al. (2024) that studies how predictable stain removal activity is for variants of *Bacillus subtilis* alpha amylase when generalizing across mutations with different characteristics. The dataset includes 3,706 variants with up to eight mutations, of which 488 are single mutants, providing an opportunity to evaluate generalization for an industrially important enzyme.

**1-to-many (number)**: Train on variants with 1 mutation; test on all other variants (including the wild type).
**close-to-far (position)**: Train on variants with mutations within 7.3Å of any active site residue; test on variants with mutations more than 7.3Å from any active site residue.
**far-to-close (position)**: Train on variants with mutations more than 7.3Å from any active site residue; test on variants with mutations within 7.3Å from any active site residue.
**by-mutation (mutation)**: Train and test on different mutations.

#### 3.2.2. IMINE REDUCTASE (IRED)

Imine reductases reduce imines to amines and are employed in pharmaceutical production. We adapt work by Gantz et al. (2024), in which a microfluidic screen was used to measure the activity of 17,143 variants from an error-prone PCR library of *Streptosporangium roseum* imine reductase, including 1,207 single mutants, 3,200 double mutants, and sequences with up to 15 mutations.

**2-to-many (number)**: Train on variants with 0, 1, or 2 mutations; test on all others.

#### 3.2.3. NUCLEASE B (NUCB)

Endonucleases such as *Bacillus licheniformis* Nuclease B (NucB) degrade DNA, and thus have potential applications in chronic wound care as they can degrade extracellular DNA required for the formation of biofilms. However, enzymatic activity for the wild-type NucB drops to around 80% at physiological pH. Thomas et al. (2025) assayed nuclease activity at pH 7 for 55,760 variants from error-prone PCR. To ameliorate the effects of assay noise, we bin these measurements, with inactive variants assigned to bin 0, variants with non-zero but less than wild-type activity assigned to bin 1, variants with greater than wild-type activity but less activity than the best single mutant assigned to bin 2, and variants with more activity than the best single mutant assigned to bin 3.

**2-to-many (number)**: Train on variants with 0, 1, or 2 mutations; test on all others.

#### 3.2.4. TRYPTOPHAN SYNTHASE $\beta$-SUBUNIT (TRPB)

The $\beta$-subunit of trytophan synthase (TrpB) synthesizes tryptophan from indole and serine and is present in all kingdoms of life except animals. This function is essential to cell growth, so TrpB fitness can be approximated using a growth-based selection assay. Johnston et al. (2024) exploited this to measure all possible residue combinations for several sets of interacting positions to examine the epistatic fitness landscape. This dataset is composed of ten different sub-datasets across 20 different positions, including nine 3-site combinatorial landscapes (~8,000 variants per landscape) and one 4-site combinatorial landscape (~160,000 variants).

**1-to-many (number)**: Train on variants with 0 or 1 mutations; test on all other variants.
**2-to-many (number)**: Train on variants with 0, 1, or 2 mutations to train; test on all other variants.
**by-position (position)**: Train on all 3-site landscape variants with no positional overlap with the 4-site landscape; test on the complete 4-site landscape plus the remaining 3-site landscapes.

#### 3.2.5. HYDROPHOBIC CORE

The hydrophobic core of a protein is crucial for its function and stability, but is less studied in the context of fitness landscapes because many mutations in the core can be detrimental to stability and folding. Accurate predictions for the effects of mutations in core residues could enable an improved understanding of the core's role in protein function and better engineering strategies. Escobedo et al. (2024) randomized seven core residues in each of three proteins (UniProt entries P06241, P01053, and P0A9X9) to the hydrophobic amino acids phenylalanine, isoleucine, leucine,

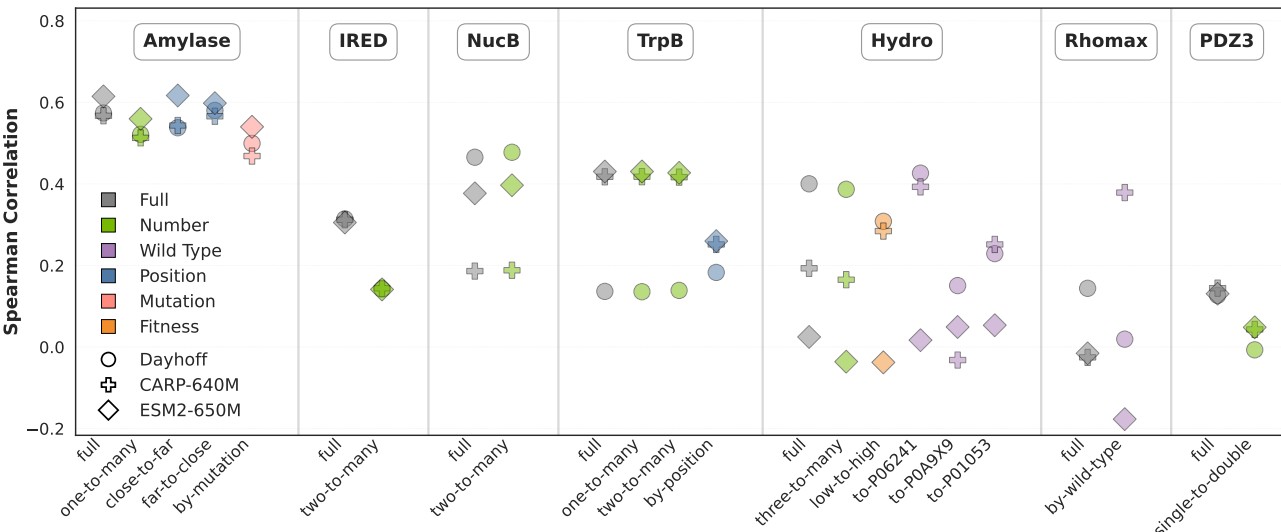

*Figure 2.* **Zero-shot pLM likelihood scores on the FLIP2 datasets and splits.**

methionine, and valine and measured the stability of these proteins, for a total of 24,935 variants assayed.

**3-to-many (number)**: Train on variants with 0, 1, 2, or 3 mutations; test on all others.

**to-P0A9X9 (wild-type)**: Train on variants of P01053 and P06241; test on variants of P0A9X9.

**to-P01053 (wild-type)** : Train on variants of P0A9X9 and P06241 to train; test on variants of P01053.

**to-P06241 (wild-type)**: Train on variants of P01053 and P0A9X9 to train; test on variants of P06241.

**low-to-high (fitness)**: Train on variants with fitness less than the median (-3.21) across all wild types; test on variants with fitness greater than the median.

### 3.2.6. RHODOPSIN

Rhodopsins are light-activated membrane proteins with applications in optogenetics, where they are used to measure and control activity in specific brain regions of living organisms. Red light penetrates deeper into brain tissue and is less damaging, but most natural rhodopsins are blue-shifted, and the sequence determinants of rhodopsin activation wavelength are complex. We use a landscape that contains 884 variants derived from 75 microbial rhodopsin sequences (Karasuyama et al., 2018; Inoue et al., 2021; Sela et al., 2024), where the goal is to predict the peak absorption wavelength for each sequence. 41 wild types have no variants. The remainder have between 1 and 181 variants, including sequences with between 1 and 6 mutations and chimeras, where segments of other wild types are substituted into the homologous location.

**by-wild-type (wild type)**: Train on the 5 most common wild types and their variants; validate on 34 wild types and

their variants; test on 36 wild types and their variants. Each split has a similar mean absorption wavelength (539.7 nm, 531.7 nm, and 534.4 nm, respectively, for train, validation, and test).

### 3.2.7. PDZ DOMAIN (PDZ3)

Despite being prevalent in the proteome, PPIs mediated by intrinsically disordered regions (IDRs) are poorly understood. To examine these in more detail, Zarin & Lehner (2024) used PSD-95/Discs-large/ZO-1 (PDZ) domains, the largest family of human protein interaction domains that can bind to short linear motifs (SLiMs) in IDRs. They measured the binding affinity of mutant variants of a domain in PDZ3 to mutant variants of a 9-amino-acid peptide representing a SLiM embedded in an IDR from the protein CRIPT, assaying >200,000 sets of double mutations (one each in the PDZ3 domain and the CRIPT peptide). Predicting these interactions would further understanding of IDRs and also improve design of functions mediated by interactions with disordered proteins.

**single-to-double (number)**: Train on single mutations in the PDZ domain; test on double mutations, one in PDZ3 and one in the CRIPT peptide. The test set is filtered to 579 sequence pairs exhibiting non-additive binding effects, where the observed affinity significantly exceeds predictions from a simple additive model (Zarin & Lehner, 2024). This focuses evaluation on variants exhibiting epistasis.

## 4. Evaluations

To characterize the behavior of different datasets and splits, and for baseline performance, we evaluated a representative set of unsupervised (zero-shot) and supervised methods (Fig-

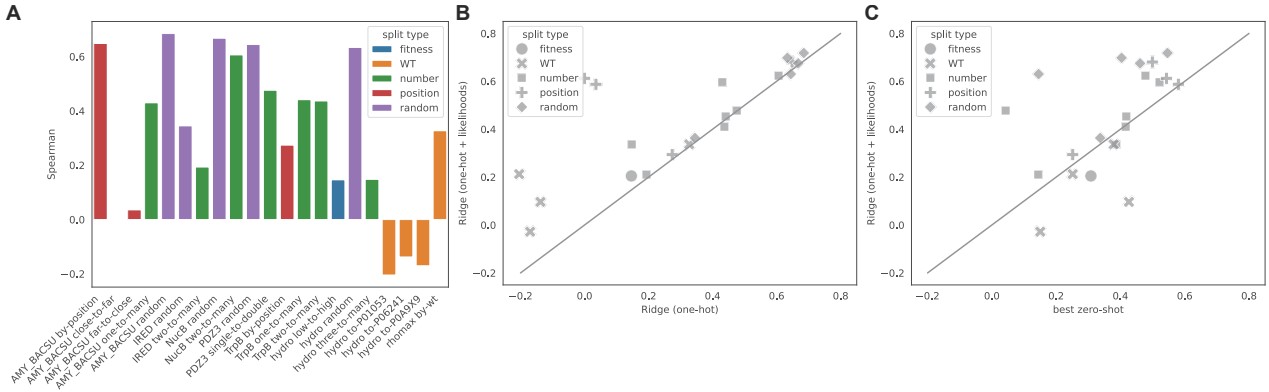

*Figure 3.* **Ridge regression performance on each FLIP2 dataset and split.** (A-G) Spearman rank correlation between ridge regression predictions and fitness for random splits of each landscape and the FLIP2 splits. Ridge regressions were performed with only sequences as inputs (one-hot) or with sequences and pLM likelihood scores as inputs (one-hot + likelihoods). Horizontal lines indicate the performance of the best zero-shot pLM likelihood score from {CARP-640M, ESM2-650M, and Dayhoff} for that landscape. (H) Comparison of Spearman rank correlation between predictions and fitness when using the best zero-shot pLM likelihood score for each FLIP2 split or ridge regression with sequences and pLM likelihood scores as input.

ure 1C). We primarily evaluated models based on Spearman correlation (Tables A1 to A16), as the ability to correctly rank-order sequences by fitness is most important when choosing sequences to prioritize for characterization (as in protein engineering use cases). We plot each model's predictions (Figures A1 to A32) and additionally report the normalized discounted cumulative gain (NDCG) (Tables A1 to A16), another ranking metric that places more weight on correctly ranking higher- than lower-fitness variants. NDCG is particularly relevant for datasets inflated with low- or non-functional variants, as the relative ranking of these variants is typically unimportant to the broader protein engineering campaign; identification and relative ranking of the most functional variants is far more important.

### 4.1. Zero-Shot Likelihood Scores

We first computed zero-shot likelihood scores to analyze how difficult it is to predict fitness in each dataset and split using only evolutionary information distilled from unlabeled amino acid sequences (Figure 2). The four representative pLMs, Dayhoff (Yang et al., 2025b), CARP (Yang et al., 2024b), ESM2 (Lin et al., 2023), and SaProt (Su et al., 2024) provide breadth in the training task (autoregressive versus masked language model), architecture (state-space-model-transformer hybrid, convolution, and transformer) and the addition of structure tokens (in SaProt). Specifically, for MLMs (ESM2-650M, CARP-640M, and SaProt-650M), we masked all mutated positions and computed the sum of the log likelihood ratios between the mutant and wild-type residues for datasets that consist of variants of a single wild type (Amylase, IRED, NucB, TrpB, and PDZ3). For the remaining datasets (Hydro and Rhomax), we computed the pseudolikelihood of each entire sequence with no positions

masked. For SaProt-650M (structure), we additionally pass structure tokens corresponding to the wild-type structures (Table A17). For Dayhoff, we averaged the N-to-C and C-to-N autoregressive likelihoods for the Dayhoff-3b-GR-HM-c and Dayhoff-3b-UR90 models. For PDZ3, we scored each of the two sequences separately and averaged the scores.

We then examined whether zero-shot likelihood scores correlate with fitness for all variants in each dataset (Figure 2, "full" column for each dataset). Zero-shot likelihood scores are much more predictive of fitness for datasets consisting of variants of a wild-type sequence (Amylase, IRED, NucB, TrpB), than for datasets with variants derived from in multiple (Hydro, Rhomax) or datasets that involve interactions between two proteins (PDZ3). Adding structure tokens in SaProt-650M does not consistently improve performance. This suggests that zero-shot likelihood scores are effective at comparing variants of the same protein, but not as effective when comparing variants for different proteins against one another. Furthermore, Dayhoff, CARP-640M, and ESM2-650M often performed very differently on each dataset, with no clear patterns based on the type of protein or the type of sequence variation. For example, while Dayhoff outperformed the other models on Hydro and Rhomax, it underperformed the other models on TrpB. This observation corroborates concurrent findings that pLM choice is task dependent (Senoner et al., 2025), and highlights the importance of reporting all results across all datasets in a benchmark – no single pLM is optimal at ranking fitness across datasets, an observation that would be obscured by a simple mean over the Spearmans. Indeed, clarity notwithstanding, and even though it is commonplace for reporting pLM zero-shot capabilities, comparing model performance via the means of bounded, asymmetrically distributed met-

rics such as correlation coefficients should be avoided. Such means are mathematically meaningless and can lead to biased, misleading results, particularly when the sample includes outliers and extreme values.

Next, we evaluated zero-shot likelihood scores for the test datasets of each dataset (Figure 2) to understand the difficulty of our splits. Notably, when splitting by wild type (i.e., Hydro and Rhomax), a model's performance can be very different on test sets versus on the full dataset, further pointing to the difficulty of harmonizing zero-shot predictions between proteins. In contrast, for number, position, and mutation splits, the zero-shot Spearman for most splits is comparable but slightly worse than over the full dataset.

## 4.2. Linear Models

We trained and evaluated ridge regression models on different numbers of randomly selected training sequences from each dataset to evaluate how well a simple linear model on one-hot sequence representations captures fitness variation (Figure 3A-G). Sequences in each dataset were zero-padded to the maximum length. These linear models performed worst on the IRED and TrpB datasets. We trained additional ridge regression models that augment the one-hot sequence with the zero-shot scores from ESM2-650M, CARP-640M, and Dayhoff, which allow the model to combine an evolutionary prior from the pLMs with information from assayed sequences (Hsu et al., 2022) (Figure 3A-G). Adding the zero-shot scores as input to the ridge regression improved performance for datasets (Amylase, IRED, NucB) and training set sizes where the best zero-shot score was more predictive than ridge regression on the sequence alone. Ridge regression on sequence alone required more data to surpasses the best zero-shot score on datasets consisting of mutations to a single wild-type enzyme (Amylase, IRED, NucB, and TrpB). Likewise, augmenting the sequence with zero-shot scores improved the performance of the ridge regression models for these datasets.

In contrast, for datasets consisting of variants derived from several backbones (Hydro and Rhomax) or to a two-protein system (PDZ3), linear models required very few training examples to outperform the zero-shot scores. On random splits linear models of one-hot sequences quickly reached high Spearman rank correlations. As a result, augmenting the sequences with zero-shot scores did not improve the performance of ridge regression models for these datasets as much.

To quantify whether each of the FLIP2 splits is more challenging than a random split of the same size, we compared ridge regression model results for our splits to the results on random splits (colored points vs. grey points, respectively, on Figure 3A-G). While almost every split is more challenging than a random split, the wild-type, position, and

fitness splits were much more challenging than the number and mutation splits. This indicates that, in many cases, linear combinations of individual mutation effects can still be predictive; but, as expected, linear models do not generalize well to new positions or to new backbones. Adding zero-shot scores as input usually resulted in similar or improved performance, especially on the more difficult position and wild-type splits. Likewise, a linear model over the zero shot scores and sequence usually matched or outperformed the best zero-shot score, except for the wild-type splits (Figure 3H). These results illustrate the danger of evaluating models using simple random splits.

## 4.3. Fine-Tuned Protein Language Models

Finally, we fine-tuned the protein language models CARP-640M, ESMC-300M, and SaProt-650M (with and without structure tokens) with both pretrained and randomly-initialized weights on each FLIP2 split to evaluate the effects of pLM pretraining and model architecture. We chose CARP-640M to compare directly to the CARP-640M zero-shot results, ESMC-300M due to the claim of being a small model optimized for learning downstream tasks (ESM Team, 2024), and SaProt-650M in order to test the effect of structure tokens. In addition, when using pretrained weights, we test both full finetuning and low-rank adaptation (LoRA) (Hu et al., 2022) with rank 32 and LoRA scaling of 64. We fine-tuned each model five times with different random seeds, using the validation set for early stopping, and report the average for each metric over those five repeats. Pretraining improved Spearman rank correlation on 14 out of 16 tasks for CARP-640M but only 9 out of 16 tasks for ESMC-300M (Figure 4A). Fine-tuning CARP-640M on supervised data only improved its zero-shot performance on 7/16 splits (Figure 4B). Notably, supervision decreased performance on every Amylase split, where the zero-shot methods generally performed well (Figure 2A-B), as well as every by-position split, indicating that fine-tuning pLMs hurts generalization to different mutational positions. Likewise, the best performing supervised pLM on each task usually outperformed ridge regression with both sequence and zero-shot likelihoods for wild type and number splits (Figure 4C).

Overall, fine-tuned pLMs underperformed in relation to their complexity and compute requirements. A fine-tuned pLM was the best-performing model on only 7/16 splits, while ridge regression with sequence and pLM likelihoods was the best on 4/16, a zero-shot pLM score was best on 4/16, and a naive supervised pLM was best on 1/16 (Table A18). Adding structure tokens or using LoRA does not consistently improve finetuning results, and no pLM consistently outperforms the others.

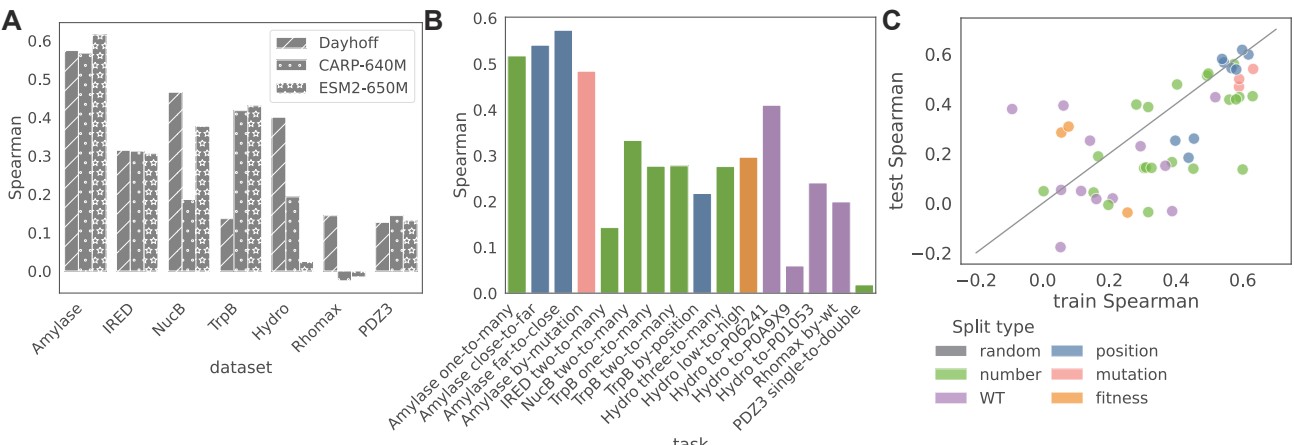

*Figure 4.* **Fine-tuned pLM performance on each FLIP2 split.** (A) Spearman rank correlation between predictions and fitness when fine-tuning pLMs with and without pretraining on each FLIP2 split. (B) Spearman rank correlation between predictions and fitness using CARP-640M zero-shot likelihood scores or else after supervised fine-tuning on the train set for each FLIP2 split. (C) Spearman rank correlation between predictions and fitness for each FLIP2 split when using ridge regression with sequences and pLM (CARP-640M, ESM2-650M, and Dayhoff) likelihood scores as input or the best fine-tuned pLM.

## 5. Discussion

We introduce FLIP2, an expanded benchmark for protein fitness prediction that is designed to probe generalization regimes that closely resemble real-world protein engineering campaigns. By adding seven diverse sequence–fitness landscapes spanning enzymes, light-sensitive proteins, and protein–protein interactions, the benchmark surfaces regimes where commonly used machine learning approaches fail, particularly when generalizing across wild types and to unseen positions. Across a broad suite of evaluated methods, simple linear models based on one-hot sequence encodings and zero-shot likelihoods from protein language models remain surprisingly competitive with and often outperform fine-tuned protein language models, especially on more difficult out-of-distribution splits.

Collectively, these findings suggest that current pLM architectures and training paradigms are not yet well matched to the challenging generalization problems that arise in protein engineering workflows. Our baselines generalize well on from fewer to more mutations or to different mutations at the same position, but they do not generalize well across wild types or to new positions (Table A18). Zero-shot likelihoods provide strong baselines in single–wild-type landscapes and can offer useful signal in distribution-shift settings, but their effectiveness deteriorates when ranking variants across different proteins or in two-protein landscapes. Ridge regression models can capture substantial fitness variation and, when augmented with zero-shot scores, often achieve the best overall performance, yet they fundamentally cannot generalize to new positions or scaffolds, as reflected in results for position and wild-type splits. Fine-tuned protein language models benefit from pretraining, but unfortunately,

they do not consistently improve over simpler methods on wild-type and position splits. In combination with previous work showing that scaling pLMs also does not improve generalization (Li et al., 2024), these results suggest that we may be reaching the limits of the current pLM transfer learning paradigm for fitness prediction.

As the field increasingly turns to using oracles trained on experimental data to steer generative models (Stocco et al., 2024; Widatalla et al., 2024; Yang et al., 2025a; Xiong et al., 2025), the ability of these oracles to generalize across increasingly large gaps in sequence space will be key to maximizing the utility of both experimental data and large, general-purpose generative models of protein sequence.

### Software and Data

FLIP2 is available at `https://flip.protein.properties`. The data is additionally hosted on Zenodo at `https://doi.org/10.5281/zenodo.18433203`.

### Impact Statement

This work aims to advance machine learning for protein engineering by providing a benchmark that better reflects the data distributions, extrapolation challenges, and practical constraints encountered in real experimental campaigns. By systematically evaluating zero-shot protein language models, linear baselines, and fine-tuned models across functional, positional, and scaffold-shift settings, the benchmark can guide researchers toward methods that are more robust in practice, thereby improving the reliability of computational tools used to design enzymes, therapeutics, and

other biotechnologically relevant proteins. At the same time, making these datasets and baselines widely available may lower barriers to entry for groups developing new methods, accelerating progress but also increasing the potential for misapplication of models to high-stakes domains, such as therapeutic design or environmental interventions.

The benchmark itself does not introduce new experimental capabilities, but it may indirectly facilitate more efficient design of proteins with altered activity, stability, or specificity, which could have substantial positive impacts in medicine, manufacturing, and sustainability (for example, by enabling better biocatalysts or more effective biologics). As with other advances in protein design, there is a dual-use dimension: improved generalization in fitness prediction could, in principle, be applied to design harmful biological agents as well as beneficial ones. To mitigate such risks, use of the benchmark and derived models should remain subject to existing biosafety and biosecurity norms and regulations, and future methodological work inspired by this benchmark should consider incorporating safety-aware objectives or filters when targeting real-world design tasks.

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

# A. Supplemental Tables and Figures

*Table A1.* Baseline performance for Amylase one-to-many

| Model | Spearman | NDCG |
|---|---|---|
| Ridge (one-hot) | 0.430 | 0.903 |
| Ridge (one-hot + likelihoods) | 0.597 | 0.947 |
| Dayhoff likelihood | 0.522 | 0.936 |
| ESM2-650M likelihood | 0.560 | 0.942 |
| CARP-640M likelihood | 0.513 | 0.931 |
| SaProt-650M likelihood (structure) | 0.560 | 0.958 |
| SaProt-650M likelihood (sequence) | $-0.015$ | 0.896 |
| CARP-640M supervised | $0.242 \pm 0.153$ | $0.873 \pm 0.029$ |
| ESMC-300M supervised | $0.148 \pm 0.181$ | $0.869 \pm 0.028$ |
| SaProt-650M supervised (structure) | $0.514 \pm 0.061$ | $0.955 \pm 0.004$ |
| SaProt-650M supervised (sequence) | $0.514 \pm 0.061$ | $0.955 \pm 0.004$ |
| CARP-640M naive supervised | $0.087 \pm 0.172$ | $0.863 \pm 0.023$ |
| ESMC-300M naive supervised | $0.044 \pm 0.028$ | $0.850 \pm 0.007$ |
| SaProt-650M naive supervised (structure) | $0.014 \pm 0.047$ | $0.910 \pm 0.009$ |
| SaProt-650M naive supervised (sequence) | $0.017 \pm 0.044$ | $0.906 \pm 0.012$ |
| SaProt-650M LoRA (structure) | $0.186 \pm 0.128$ | $0.932 \pm 0.012$ |
| SaProt-650M LoRA (sequence) | $0.149 \pm 0.112$ | $0.929 \pm 0.010$ |
| CARP-640M LoRA | $0.126 \pm 0.124$ | $0.853 \pm 0.015$ |
| ESMC-300M LoRA | $0.044 \pm 0.173$ | $0.852 \pm 0.028$ |

*Table A2.* Baseline performance for Amylase close-to-far

| Model | Spearman | NDCG |
|---|---|---|
| Ridge (one-hot) | *nan* | 0.865 |
| Ridge (one-hot + likelihoods) | 0.612 | 0.955 |
| Dayhoff likelihood | 0.538 | 0.943 |
| ESM2-650M likelihood | 0.617 | 0.955 |
| CARP-640M likelihood | 0.543 | 0.945 |
| SaProt-650M likelihood (structure) | 0.604 | 0.936 |
| SaProt-650M likelihood (sequence) | 0.085 | 0.841 |
| CARP-640M supervised | $0.081 \pm 0.055$ | $0.867 \pm 0.015$ |
| ESMC-300M supervised | $0.107 \pm 0.113$ | $0.869 \pm 0.024$ |
| SaProt-650M supervised (structure) | $0.125 \pm 0.097$ | $0.877 \pm 0.016$ |
| SaProt-650M supervised (sequence) | $0.125 \pm 0.097$ | $0.877 \pm 0.016$ |
| CARP-640M naive supervised | $0.006 \pm 0.117$ | $0.864 \pm 0.016$ |
| ESMC-300M naive supervised | $0.007 \pm 0.032$ | $0.864 \pm 0.012$ |
| SaProt-650M naive supervised (structure) | $-0.013 \pm 0.055$ | $0.860 \pm 0.005$ |
| SaProt-650M naive supervised (sequence) | $-0.013 \pm 0.055$ | $0.860 \pm 0.005$ |
| SaProt-650M LoRA (structure) | $0.047 \pm 0.035$ | $0.852 \pm 0.006$ |
| SaProt-650M LoRA (sequence) | $0.047 \pm 0.035$ | $0.852 \pm 0.006$ |
| CARP-640M LoRA | $0.229 \pm 0.058$ | $0.896 \pm 0.008$ |
| ESMC-300M LoRA | $0.145 \pm 0.082$ | $0.873 \pm 0.012$ |

*Table A3.* Baseline performance for Amylase far-to-close

| Model | Spearman | NDCG |
|---|---|---|
| Ridge (one-hot) | 0.036 | 0.836 |
| Ridge (one-hot + likelihoods) | 0.588 | 0.933 |
| Dayhoff likelihood | 0.580 | 0.947 |
| ESM2-650M likelihood | 0.598 | 0.932 |
| CARP-640M likelihood | 0.566 | 0.919 |
| SaProt-650M likelihood (structure) | 0.588 | 0.929 |
| SaProt-650M likelihood (sequence) | 0.155 | 0.813 |
| CARP-640M supervised | $0.140 \pm 0.102$ | $0.832 \pm 0.019$ |
| ESMC-300M supervised | $0.011 \pm 0.066$ | $0.819 \pm 0.011$ |
| SaProt-650M supervised (structure) | $0.178 \pm 0.098$ | $0.850 \pm 0.019$ |
| SaProt-650M supervised (sequence) | $0.184 \pm 0.102$ | $0.852 \pm 0.021$ |
| CARP-640M naive supervised | $-0.019 \pm 0.057$ | $0.825 \pm 0.009$ |
| ESMC-300M naive supervised | $0.015 \pm 0.028$ | $0.818 \pm 0.002$ |
| SaProt-650M naive supervised (structure) | $0.002 \pm 0.054$ | $0.829 \pm 0.013$ |
| SaProt-650M naive supervised (sequence) | $0.002 \pm 0.054$ | $0.829 \pm 0.013$ |
| SaProt-650M LoRA (structure) | $0.176 \pm 0.034$ | $0.834 \pm 0.004$ |
| SaProt-650M LoRA (sequence) | $0.176 \pm 0.034$ | $0.834 \pm 0.004$ |
| CARP-640M LoRA | $0.230 \pm 0.104$ | $0.846 \pm 0.015$ |
| ESMC-300M LoRA | $0.191 \pm 0.078$ | $0.842 \pm 0.016$ |

*Table A4.* Baseline performance for Amylase by-mutation

| Model | Spearman | NDCG |
|---|---|---|
| Ridge (one-hot) | 0.648 | 0.940 |
| Ridge (one-hot + likelihoods) | 0.681 | 0.940 |
| Dayhoff likelihood | 0.499 | 0.917 |
| ESM2-650M likelihood | 0.540 | 0.909 |
| CARP-640M likelihood | 0.468 | 0.906 |
| SaProt-650M likelihood (structure) | 0.510 | 0.893 |
| SaProt-650M likelihood (sequence) | 0.100 | 0.773 |
| CARP-640M supervised | $0.551 \pm 0.239$ | $0.926 \pm 0.050$ |
| ESMC-300M supervised | $0.061 \pm 0.130$ | $0.793 \pm 0.030$ |
| SaProt-650M supervised (structure) | $0.606 \pm 0.022$ | $0.928 \pm 0.011$ |
| SaProt-650M supervised (sequence) | $0.568 \pm 0.013$ | $0.921 \pm 0.006$ |
| CARP-640M naive supervised | $0.109 \pm 0.134$ | $0.819 \pm 0.029$ |
| ESMC-300M naive supervised | $0.054 \pm 0.115$ | $0.819 \pm 0.048$ |
| SaProt-650M naive supervised (structure) | $0.042 \pm 0.033$ | $0.811 \pm 0.009$ |
| SaProt-650M naive supervised (sequence) | $0.088 \pm 0.067$ | $0.830 \pm 0.026$ |
| SaProt-650M LoRA (structure) | $0.347 \pm 0.067$ | $0.881 \pm 0.016$ |
| SaProt-650M LoRA (sequence) | $0.347 \pm 0.067$ | $0.881 \pm 0.016$ |
| CARP-640M LoRA | $0.478 \pm 0.044$ | $0.888 \pm 0.024$ |
| ESMC-300M LoRA | $0.582 \pm 0.014$ | $0.928 \pm 0.006$ |

*Table A5.* Baseline performance for IRED two-to-many

| Model | Spearman | NDCG |
|---|---|---|
| Ridge (one-hot) | 0.193 | 0.960 |
| Ridge (one-hot + likelihoods) | 0.211 | 0.964 |
| Dayhoff likelihood | 0.142 | 0.955 |
| ESM2-650M likelihood | 0.141 | 0.953 |
| CARP-640M likelihood | 0.144 | 0.956 |
| SaProt-650M likelihood (structure) | 0.029 | 1.451 |
| SaProt-650M likelihood (sequence) | 0.056 | 1.436 |
| CARP-640M supervised | $0.072 \pm 0.046$ | $0.948 \pm 0.004$ |
| ESMC-300M supervised | $0.160 \pm 0.022$ | $0.954 \pm 0.002$ |
| SaProt-650M supervised (structure) | $0.203 \pm 0.010$ | $0.958 \pm 0.002$ |
| SaProt-650M supervised (sequence) | $0.192 \pm 0.021$ | $0.957 \pm 0.003$ |
| CARP-640M naive supervised | $0.012 \pm 0.017$ | $0.944 \pm 0.001$ |
| ESMC-300M naive supervised | $0.106 \pm 0.005$ | $0.949 \pm 0.001$ |
| SaProt-650M naive supervised (structure) | $0.000 \pm 0.013$ | $0.944 \pm 0.001$ |
| SaProt-650M naive supervised (sequence) | $0.019 \pm 0.036$ | $0.945 \pm 0.002$ |
| SaProt-650M LoRA (structure) | $0.185 \pm 0.026$ | $0.956 \pm 0.003$ |
| SaProt-650M LoRA (sequence) | $0.163 \pm 0.013$ | $0.954 \pm 0.002$ |
| CARP-640M LoRA | $0.150 \pm 0.009$ | $0.952 \pm 0.001$ |
| ESMC-300M LoRA | $0.220 \pm 0.006$ | $0.959 \pm 0.001$ |

*Table A6.* Baseline performance for NucB two-to-many

| Model | Spearman | NDCG |
|---|---|---|
| Ridge (one-hot) | 0.606 | 0.943 |
| Ridge (one-hot + likelihoods) | 0.623 | 0.946 |
| Dayhoff likelihood | 0.478 | 0.919 |
| ESM2-650M likelihood | 0.397 | 0.906 |
| CARP-640M likelihood | 0.188 | 0.889 |
| SaProt-650M likelihood (structure) | 0.493 | 0.898 |
| SaProt-650M likelihood (sequence) | 0.191 | 0.862 |
| CARP-640M supervised | $0.717 \pm 0.012$ | $0.970 \pm 0.001$ |
| ESMC-300M supervised | $0.723 \pm 0.002$ | $0.966 \pm 0.000$ |
| SaProt-650M supervised (structure) | $0.419 \pm 0.046$ | $0.909 \pm 0.003$ |
| SaProt-650M supervised (sequence) | $0.419 \pm 0.046$ | $0.909 \pm 0.003$ |
| CARP-640M naive supervised | $0.704 \pm 0.007$ | $0.967 \pm 0.001$ |
| ESMC-300M naive supervised | $0.635 \pm 0.011$ | $0.956 \pm 0.003$ |
| SaProt-650M naive supervised (structure) | $0.037 \pm 0.028$ | $0.866 \pm 0.005$ |
| SaProt-650M naive supervised (sequence) | $0.037 \pm 0.028$ | $0.866 \pm 0.005$ |
| SaProt-650M LoRA (structure) | $0.382 \pm 0.037$ | $0.902 \pm 0.004$ |
| SaProt-650M LoRA (sequence) | $0.328 \pm 0.017$ | $0.895 \pm 0.003$ |
| CARP-640M LoRA | $0.571 \pm 0.009$ | $0.942 \pm 0.002$ |
| ESMC-300M LoRA | $0.730 \pm 0.008$ | $0.969 \pm 0.001$ |

*Table A7.* Baseline performance for TrpB one-to-many

| Model | Spearman | NDCG |
|---|---|---|
| Ridge (one-hot) | 0.441 | 0.989 |
| Ridge (one-hot + likelihoods) | 0.453 | 0.993 |
| Dayhoff likelihood | 0.136 | 0.991 |
| ESM2-650M likelihood | 0.430 | 0.992 |
| CARP-640M likelihood | 0.418 | 0.992 |
| SaProt-650M likelihood (structure) | 0.202 | 0.714 |
| SaProt-650M likelihood (sequence) | 0.328 | 0.744 |
| CARP-640M supervised | $0.451 \pm 0.022$ | $0.992 \pm 0.002$ |
| ESMC-300M supervised | $0.306 \pm 0.063$ | $0.992 \pm 0.001$ |
| SaProt-650M supervised (structure) | $0.337 \pm 0.062$ | $0.987 \pm 0.004$ |
| SaProt-650M supervised (sequence) | $0.341 \pm 0.058$ | $0.987 \pm 0.004$ |
| CARP-640M naive supervised | $0.298 \pm 0.217$ | $0.984 \pm 0.002$ |
| ESMC-300M naive supervised | $0.369 \pm 0.028$ | $0.986 \pm 0.002$ |
| SaProt-650M naive supervised (structure) | $0.218 \pm 0.115$ | $0.984 \pm 0.001$ |
| SaProt-650M naive supervised (sequence) | $0.254 \pm 0.140$ | $0.985 \pm 0.001$ |
| SaProt-650M LoRA (structure) | $0.182 \pm 0.086$ | $0.983 \pm 0.002$ |
| SaProt-650M LoRA (sequence) | $0.184 \pm 0.084$ | $0.983 \pm 0.002$ |
| CARP-640M LoRA | $0.389 \pm 0.026$ | $0.987 \pm 0.001$ |
| ESMC-300M LoRA | $0.261 \pm 0.050$ | $0.992 \pm 0.001$ |

*Table A8.* Baseline performance for TrpB two-to-many

| Model | Spearman | NDCG |
|---|---|---|
| Ridge (one-hot) | 0.437 | 0.994 |
| Ridge (one-hot + likelihoods) | 0.410 | 0.994 |
| Dayhoff likelihood | 0.139 | 0.991 |
| ESM2-650M likelihood | 0.427 | 0.992 |
| CARP-640M likelihood | 0.417 | 0.992 |
| SaProt-650M likelihood (structure) | 0.203 | 0.714 |
| SaProt-650M likelihood (sequence) | 0.323 | 0.743 |
| CARP-640M supervised | $0.509 \pm 0.010$ | $0.996 \pm 0.000$ |
| ESMC-300M supervised | $0.429 \pm 0.042$ | $0.991 \pm 0.003$ |
| SaProt-650M supervised (structure) | $0.495 \pm 0.028$ | $0.995 \pm 0.001$ |
| SaProt-650M supervised (sequence) | $0.470 \pm 0.017$ | $0.994 \pm 0.002$ |
| CARP-640M naive supervised | $0.328 \pm 0.095$ | $0.985 \pm 0.002$ |
| ESMC-300M naive supervised | $0.398 \pm 0.020$ | $0.986 \pm 0.001$ |
| SaProt-650M naive supervised (structure) | $0.396 \pm 0.012$ | $0.986 \pm 0.000$ |
| SaProt-650M naive supervised (sequence) | $0.372 \pm 0.012$ | $0.986 \pm 0.001$ |
| SaProt-650M LoRA (structure) | $0.435 \pm 0.027$ | $0.992 \pm 0.001$ |
| SaProt-650M LoRA (sequence) | $0.444 \pm 0.013$ | $0.992 \pm 0.001$ |
| CARP-640M LoRA | $0.445 \pm 0.021$ | $0.993 \pm 0.001$ |
| ESMC-300M LoRA | $0.418 \pm 0.030$ | $0.996 \pm 0.000$ |

*Table A9.* Baseline performance for TrpB by-position

| Model | Spearman | NDCG |
|---|---|---|
| Ridge (one-hot) | 0.274 | 0.987 |
| Ridge (one-hot + likelihoods) | 0.294 | 0.988 |
| Dayhoff likelihood | 0.183 | 0.988 |
| ESM2-650M likelihood | 0.260 | 0.988 |
| CARP-640M likelihood | 0.252 | 0.988 |
| SaProt-650M likelihood (structure) | 0.109 | 0.745 |
| SaProt-650M likelihood (sequence) | 0.183 | 0.776 |
| CARP-640M supervised | $0.167 \pm 0.018$ | $0.983 \pm 0.003$ |
| ESMC-300M supervised | $0.091 \pm 0.118$ | $0.980 \pm 0.007$ |
| SaProt-650M supervised (structure) | $0.177 \pm 0.053$ | $0.982 \pm 0.003$ |
| SaProt-650M supervised (sequence) | $0.207 \pm 0.027$ | $0.980 \pm 0.003$ |
| CARP-640M naive supervised | $-0.009 \pm nan$ | $0.975 \pm 0.000$ |
| ESMC-300M naive supervised | $0.154 \pm 0.059$ | $0.978 \pm 0.004$ |
| SaProt-650M naive supervised (structure) | $0.104 \pm 0.019$ | $0.975 \pm 0.001$ |
| SaProt-650M naive supervised (sequence) | $0.088 \pm 0.071$ | $0.975 \pm 0.002$ |
| SaProt-650M LoRA (structure) | $0.146 \pm 0.030$ | $0.976 \pm 0.002$ |
| SaProt-650M LoRA (sequence) | $0.146 \pm 0.030$ | $0.976 \pm 0.002$ |
| CARP-640M LoRA | $0.241 \pm 0.048$ | $0.986 \pm 0.003$ |
| ESMC-300M LoRA | $0.191 \pm 0.026$ | $0.989 \pm 0.001$ |

*Table A10.* Baseline performance for hydro three-to-many

| Model | Spearman | NDCG |
|---|---|---|
| Ridge (one-hot) | 0.148 | 0.960 |
| Ridge (one-hot + likelihoods) | 0.337 | 0.974 |
| Dayhoff likelihood | 0.387 | 0.970 |
| ESM2-650M likelihood | $-0.036$ | 0.967 |
| CARP-640M likelihood | 0.165 | 0.963 |
| SaProt-650M likelihood (structure) | $-0.119$ | 1.014 |
| SaProt-650M likelihood (sequence) | $-0.080$ | 1.014 |
| CARP-640M supervised | $0.516 \pm 0.035$ | $0.982 \pm 0.002$ |
| ESMC-300M supervised | $0.399 \pm 0.174$ | $0.977 \pm 0.007$ |
| SaProt-650M supervised (structure) | $0.558 \pm 0.042$ | $0.982 \pm 0.001$ |
| SaProt-650M supervised (sequence) | $0.558 \pm 0.042$ | $0.982 \pm 0.001$ |
| CARP-640M naive supervised | $0.604 \pm 0.035$ | $0.984 \pm 0.001$ |
| ESMC-300M naive supervised | $0.367 \pm 0.042$ | $0.973 \pm 0.002$ |
| SaProt-650M naive supervised (structure) | $0.390 \pm 0.022$ | $0.977 \pm 0.000$ |
| SaProt-650M naive supervised (sequence) | $0.390 \pm 0.022$ | $0.977 \pm 0.000$ |
| SaProt-650M LoRA (structure) | $0.559 \pm 0.059$ | $0.984 \pm 0.002$ |
| SaProt-650M LoRA (sequence) | $0.559 \pm 0.059$ | $0.984 \pm 0.002$ |
| CARP-640M LoRA | $0.409 \pm 0.114$ | $0.973 \pm 0.005$ |
| ESMC-300M LoRA | $0.485 \pm 0.059$ | $0.978 \pm 0.006$ |

*Table A11.* Baseline performance for hydro low-to-high

| Model | Spearman | NDCG |
|---|---|---|
| Ridge (one-hot) | 0.146 | 0.887 |
| Ridge (one-hot + likelihoods) | 0.205 | 0.903 |
| Dayhoff likelihood | 0.308 | 0.924 |
| ESM2-650M likelihood | $-0.038$ | 0.865 |
| CARP-640M likelihood | 0.284 | 0.921 |
| SaProt-650M likelihood (structure) | 0.183 | 1.018 |
| SaProt-650M likelihood (sequence) | 0.300 | 1.014 |
| CARP-640M supervised | $0.188 \pm 0.071$ | $0.887 \pm 0.007$ |
| ESMC-300M supervised | $0.099 \pm 0.194$ | $0.870 \pm 0.030$ |
| SaProt-650M supervised (structure) | $0.352 \pm 0.031$ | $0.923 \pm 0.006$ |
| SaProt-650M supervised (sequence) | $0.353 \pm 0.031$ | $0.923 \pm 0.006$ |
| CARP-640M naive supervised | $0.188 \pm 0.202$ | $0.896 \pm 0.032$ |
| ESMC-300M naive supervised | $0.270 \pm 0.048$ | $0.905 \pm 0.008$ |
| SaProt-650M naive supervised (structure) | $0.181 \pm 0.029$ | $0.873 \pm 0.005$ |
| SaProt-650M naive supervised (sequence) | $0.165 \pm 0.037$ | $0.872 \pm 0.006$ |
| SaProt-650M LoRA (structure) | $0.352 \pm 0.026$ | $0.926 \pm 0.002$ |
| SaProt-650M LoRA (sequence) | $0.329 \pm 0.006$ | $0.925 \pm 0.002$ |
| CARP-640M LoRA | $0.200 \pm 0.098$ | $0.917 \pm 0.014$ |
| ESMC-300M LoRA | $0.311 \pm 0.041$ | $0.939 \pm 0.004$ |

*Table A12.* Baseline performance for hydro to-P06241

| Model | Spearman | NDCG |
|---|---|---|
| Ridge (one-hot) | $-0.138$ | 0.913 |
| Ridge (one-hot + likelihoods) | 0.097 | 0.926 |
| Dayhoff likelihood | 0.426 | 0.967 |
| ESM2-650M likelihood | 0.017 | 0.928 |
| CARP-640M likelihood | 0.393 | 0.957 |
| SaProt-650M likelihood (structure) | 0.414 | 1.017 |
| SaProt-650M likelihood (sequence) | 0.444 | 1.016 |
| CARP-640M supervised | $0.150 \pm 0.046$ | $0.933 \pm 0.008$ |
| ESMC-300M supervised | $0.161 \pm 0.184$ | $0.934 \pm 0.011$ |
| SaProt-650M supervised (structure) | $0.155 \pm 0.027$ | $0.935 \pm 0.001$ |
| SaProt-650M supervised (sequence) | $0.156 \pm 0.030$ | $0.935 \pm 0.001$ |
| CARP-640M naive supervised | $0.006 \pm 0.137$ | $0.927 \pm 0.006$ |
| ESMC-300M naive supervised | $0.098 \pm 0.024$ | $0.927 \pm 0.001$ |
| SaProt-650M naive supervised (structure) | $0.050 \pm 0.029$ | $0.929 \pm 0.002$ |
| SaProt-650M naive supervised (sequence) | $0.020 \pm 0.022$ | $0.927 \pm 0.003$ |
| SaProt-650M LoRA (structure) | $0.207 \pm 0.049$ | $0.940 \pm 0.004$ |
| SaProt-650M LoRA (sequence) | $0.250 \pm 0.062$ | $0.940 \pm 0.004$ |
| CARP-640M LoRA | $0.107 \pm 0.158$ | $0.931 \pm 0.011$ |
| ESMC-300M LoRA | $-0.160 \pm 0.429$ | $0.918 \pm 0.031$ |

*Table A13.* Baseline performance for hydro to-P0A9X9

| Model | Spearman | NDCG |
|---|---|---|
| Ridge (one-hot) | $-0.170$ | 0.957 |
| Ridge (one-hot + likelihoods) | $-0.028$ | 0.964 |
| Dayhoff likelihood | 0.151 | 0.973 |
| ESM2-650M likelihood | 0.049 | 0.964 |
| CARP-640M likelihood | $-0.032$ | 0.967 |
| SaProt-650M likelihood (structure) | 0.001 | 1.007 |
| SaProt-650M likelihood (sequence) | 0.024 | 1.008 |
| CARP-640M supervised | $0.004 \pm 0.103$ | $0.961 \pm 0.004$ |
| ESMC-300M supervised | $0.078 \pm 0.133$ | $0.970 \pm 0.001$ |
| SaProt-650M supervised (structure) | $0.171 \pm 0.075$ | $0.966 \pm 0.002$ |
| SaProt-650M supervised (sequence) | $0.134 \pm 0.063$ | $0.966 \pm 0.002$ |
| CARP-640M naive supervised | $-0.029 \pm 0.074$ | $0.960 \pm 0.002$ |
| ESMC-300M naive supervised | $-0.050 \pm 0.193$ | $0.958 \pm 0.004$ |
| SaProt-650M naive supervised (structure) | $-0.098 \pm 0.114$ | $0.957 \pm 0.004$ |
| SaProt-650M naive supervised (sequence) | $-0.113 \pm 0.113$ | $0.957 \pm 0.003$ |
| SaProt-650M LoRA (structure) | $0.115 \pm 0.035$ | $0.967 \pm 0.001$ |
| SaProt-650M LoRA (sequence) | $0.039 \pm 0.075$ | $0.968 \pm 0.002$ |
| CARP-640M LoRA | $-0.017 \pm 0.181$ | $0.960 \pm 0.006$ |
| ESMC-300M LoRA | $-0.030 \pm 0.088$ | $0.965 \pm 0.002$ |

*Table A14.* Baseline performance for hydro to-P01053

| Model | Spearman | NDCG |
|---|---|---|
| Ridge (one-hot) | $-0.204$ | 0.889 |
| Ridge (one-hot + likelihoods) | 0.213 | 0.942 |
| Dayhoff likelihood | 0.229 | 0.964 |
| ESM2-650M likelihood | 0.053 | 0.911 |
| CARP-640M likelihood | 0.252 | 0.948 |
| SaProt-650M likelihood (structure) | 0.314 | 1.009 |
| SaProt-650M likelihood (sequence) | 0.308 | 1.011 |
| CARP-640M supervised | $0.049 \pm 0.088$ | $0.909 \pm 0.008$ |
| ESMC-300M supervised | $0.329 \pm 0.190$ | $0.949 \pm 0.024$ |
| SaProt-650M supervised (structure) | $0.207 \pm 0.041$ | $0.913 \pm 0.004$ |
| SaProt-650M supervised (sequence) | $0.260 \pm 0.078$ | $0.919 \pm 0.009$ |
| CARP-640M naive supervised | $0.061 \pm 0.094$ | $0.907 \pm 0.009$ |
| ESMC-300M naive supervised | $-0.086 \pm 0.135$ | $0.895 \pm 0.006$ |
| SaProt-650M naive supervised (structure) | $-0.002 \pm 0.057$ | $0.905 \pm 0.009$ |
| SaProt-650M naive supervised (sequence) | $0.003 \pm 0.080$ | $0.906 \pm 0.007$ |
| SaProt-650M LoRA (structure) | $0.203 \pm 0.046$ | $0.910 \pm 0.004$ |
| SaProt-650M LoRA (sequence) | $0.252 \pm 0.041$ | $0.920 \pm 0.008$ |
| CARP-640M LoRA | $0.064 \pm 0.188$ | $0.904 \pm 0.012$ |
| ESMC-300M LoRA | $0.394 \pm 0.006$ | $0.957 \pm 0.004$ |

*Table A15.* Baseline performance for Rhomax by-wild-type

| Model | Spearman | NDCG |
|---|---|---|
| Ridge (one-hot) | 0.327 | 0.939 |
| Ridge (one-hot + likelihoods) | 0.337 | 0.936 |
| Dayhoff likelihood | 0.019 | 0.902 |
| ESM2-650M likelihood | $-0.177$ | 0.904 |
| CARP-640M likelihood | 0.379 | 0.937 |
| SaProt-650M likelihood (structure) | 0.201 | 0.177 |
| SaProt-650M likelihood (sequence) | 0.018 | 0.175 |
| CARP-640M supervised | $0.072 \pm 0.244$ | $0.934 \pm 0.029$ |
| ESMC-300M supervised | $-0.016 \pm 0.113$ | $0.942 \pm 0.008$ |
| SaProt-650M supervised (structure) | $- - -$ | $0.921 \pm 0.000$ |
| SaProt-650M supervised (sequence) | $0.253 \pm 0.150$ | $0.938 \pm 0.007$ |
| CARP-640M naive supervised | $0.056 \pm 0.125$ | $0.934 \pm 0.009$ |
| ESMC-300M naive supervised | $0.036 \pm 0.218$ | $0.936 \pm 0.022$ |
| SaProt-650M naive supervised (structure) | $- - -$ | $0.921 \pm 0.000$ |
| SaProt-650M naive supervised (sequence) | $- - -$ | $0.911 \pm 0.024$ |
| SaProt-650M LoRA (structure) | $- - -$ | $0.921 \pm 0.000$ |
| SaProt-650M LoRA (sequence) | $0.118 \pm 0.185$ | $0.928 \pm 0.010$ |
| CARP-640M LoRA | $0.085 \pm 0.187$ | $0.933 \pm 0.008$ |
| ESMC-300M LoRA | $-0.105 \pm 0.281$ | $0.922 \pm 0.036$ |

*Table A16.* Baseline performance for PDZ3 single-to-double

| Model | Spearman | NDCG |
|---|---|---|
| Ridge (one-hot) | 0.476 | 0.873 |
| Ridge (one-hot + likelihoods) | 0.478 | 0.875 |
| Dayhoff likelihood | $-0.007$ | 0.856 |
| ESM2-650M likelihood | 0.048 | 0.854 |
| CARP-640M likelihood | 0.043 | 0.863 |
| SaProt-650M likelihood (structure) | 0.057 | 1.153 |
| SaProt-650M likelihood (sequence) | $-0.123$ | 1.159 |
| CARP-640M supervised | $0.512 \pm 0.014$ | $0.882 \pm 0.015$ |
| ESMC-300M supervised | $0.499 \pm 0.018$ | $0.889 \pm 0.011$ |
| SaProt-650M supervised (structure) | $0.488 \pm 0.006$ | $0.874 \pm 0.009$ |
| SaProt-650M supervised (sequence) | $0.488 \pm 0.006$ | $0.874 \pm 0.009$ |
| CARP-640M naive supervised | $0.464 \pm 0.022$ | $0.870 \pm 0.008$ |
| ESMC-300M naive supervised | $0.502 \pm 0.016$ | $0.880 \pm 0.006$ |
| SaProt-650M naive supervised (structure) | $0.338 \pm 0.194$ | $0.875 \pm 0.015$ |
| SaProt-650M naive supervised (sequence) | $0.338 \pm 0.194$ | $0.875 \pm 0.015$ |
| SaProt-650M LoRA (structure) | $0.329 \pm 0.155$ | $0.878 \pm 0.023$ |
| SaProt-650M LoRA (sequence) | $0.329 \pm 0.155$ | $0.878 \pm 0.023$ |
| CARP-640M LoRA | $0.124 \pm 0.121$ | $0.864 \pm 0.025$ |
| ESMC-300M LoRA | $0.319 \pm 0.221$ | $0.863 \pm 0.022$ |

*Table A17.* PDB codes used for visualisation and SaProt structure encoding.

| Protein | PDB Code |
|---|---|
| PDZ3 | 5HEB |
| NucB | 5OMT |
| IRED | 5OCM |
| TrpB | 8VHH |
| P06241 | 5ZAU |
| P0A9X9 | 3MEF |
| P01053 | 3CI2 |
| Amylase | 1UA7 |
| Rhomax | 1F88 |

*Table A18.* Best-performing model for each split.

| Dataset | Split | Split type | Best model | Best Spearman |
|---|---|---|---|---|
| Alpha amylase (Amylase) | one-to-many | number | Ridge (one-hot + likelihoods) | 0.597 |
| | close-to-far | position | ESM2-650M likelihood | 0.617 |
| | far-to-close | position | ESM2-650M likelihood | 0.598 |
| | by-mutation | mutation | Ridge (one-hot + likelihoods) | 0.681 |
| Imine reductase (IRED) | two-to-many | number | ESMC-300M LoRA | 0.220 |
| Nuclease B (NucB) | two-to-many | number | ESMC-300M LoRA | 0.730 |
| Tryptophan B (TrpB) | one-to-many | number | Ridge (one-hot + likelihoods) | 0.453 |
| | two-to-many | number | CARP-640M supervised | 0.509 |
| | by-position | position | Ridge (one-hot + likelihoods) | 0.294 |
| Hydrophobic core (hydro) | three-to-many | number | CARP-640M naive supervised | 0.604 |
| | low-to-high | fitness | SaProt-650M supervised (sequence) | 0.353 |
| | to-P06241 | wild-type | SaProt-650M likelihood (sequence) | 0.444 |
| | to-P0A9X9 | wild-type | SaProt-650M supervised (structure) | 0.171 |
| | to-P01053 | wild-type | ESMC-300M LoRA | 0.394 |
| Rhodopsin (rhomax) | by-wild-type | wild-type | CARP-640M likelihood | 0.379 |
| PDZ3 | single-to-double | number | CARP-640M supervised | 0.512 |

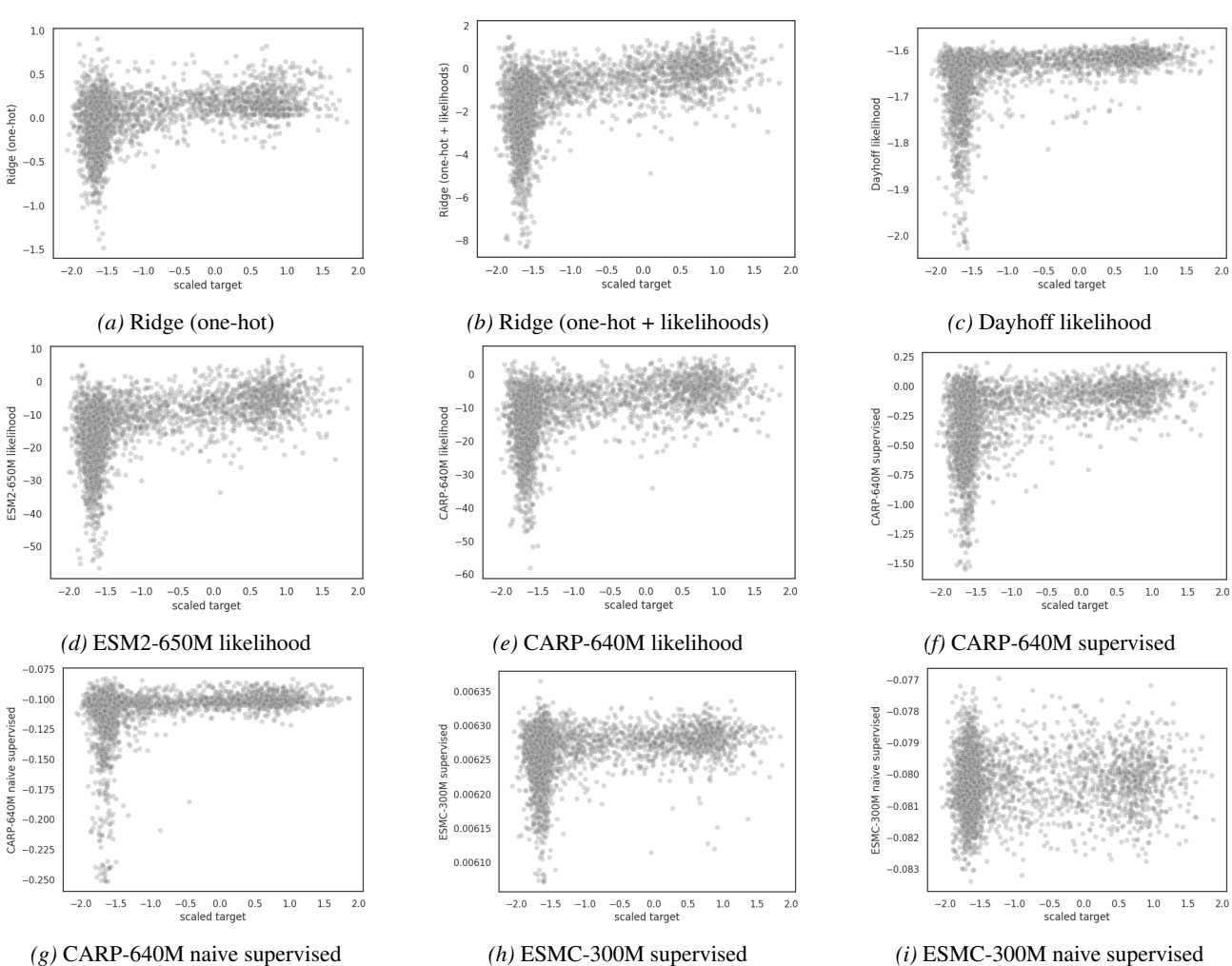

*(a)* Ridge (one-hot)    *(b)* Ridge (one-hot + likelihoods)    *(c)* Dayhoff likelihood

*(d)* ESM2-650M likelihood    *(e)* CARP-640M likelihood    *(f)* CARP-640M supervised

*(g)* CARP-640M naive supervised    *(h)* ESMC-300M supervised    *(i)* ESMC-300M naive supervised

*Figure A1.* Predictions for amylase one-to-many. Finetuned PLM predictions are averaged over 5 random seeds.

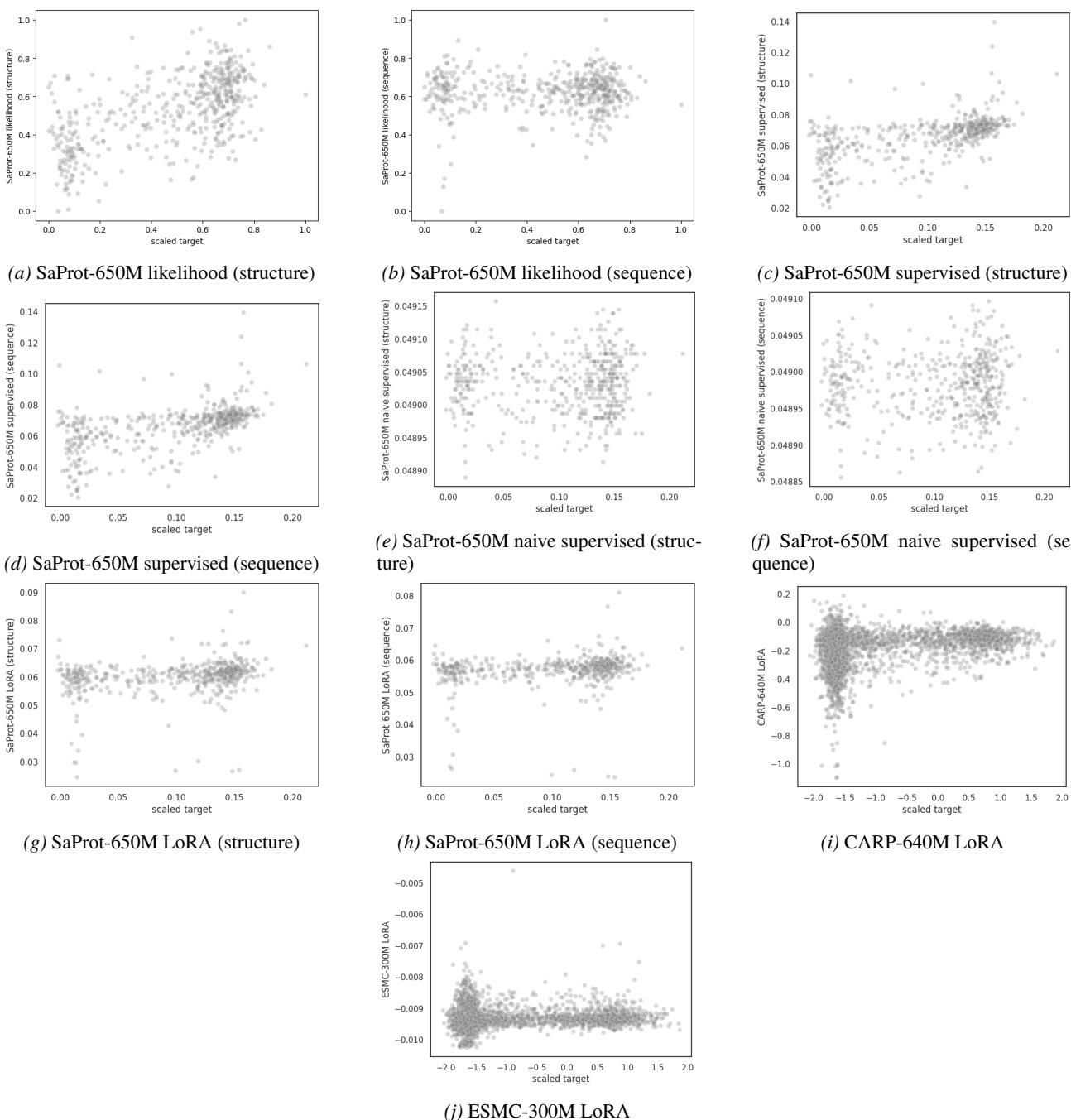

*Figure A2.* Predictions for amylase one-to-many (continued). Finetuned PLM predictions are averaged over 5 random seeds.

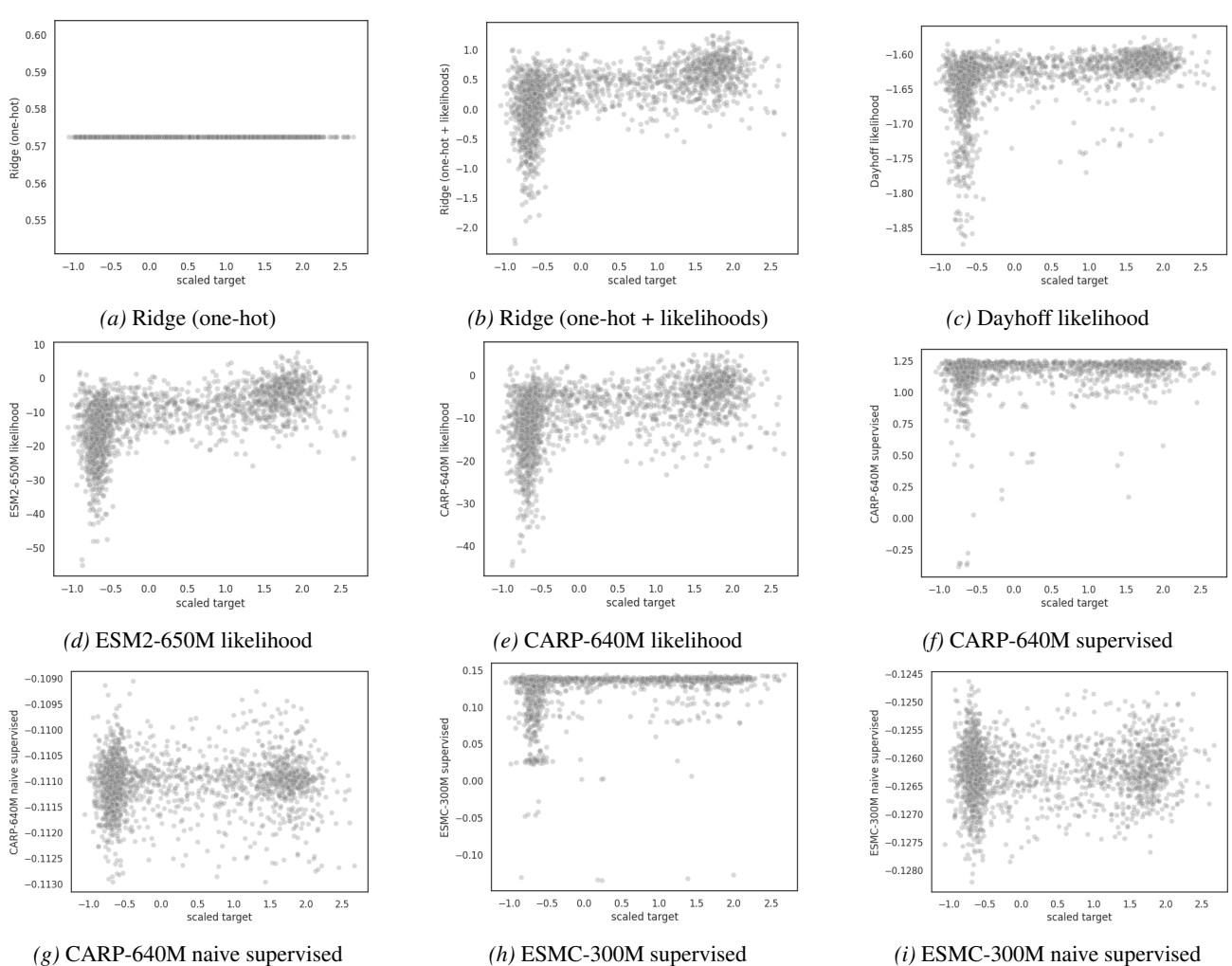

*(a)* Ridge (one-hot)  *(b)* Ridge (one-hot + likelihoods)  *(c)* Dayhoff likelihood

*(d)* ESM2-650M likelihood  *(e)* CARP-640M likelihood  *(f)* CARP-640M supervised

*(g)* CARP-640M naive supervised  *(h)* ESMC-300M supervised  *(i)* ESMC-300M naive supervised

*Figure A3.* Predictions for amylase close-to-far. Finetuned PLM predictions are averaged over 5 random seeds.

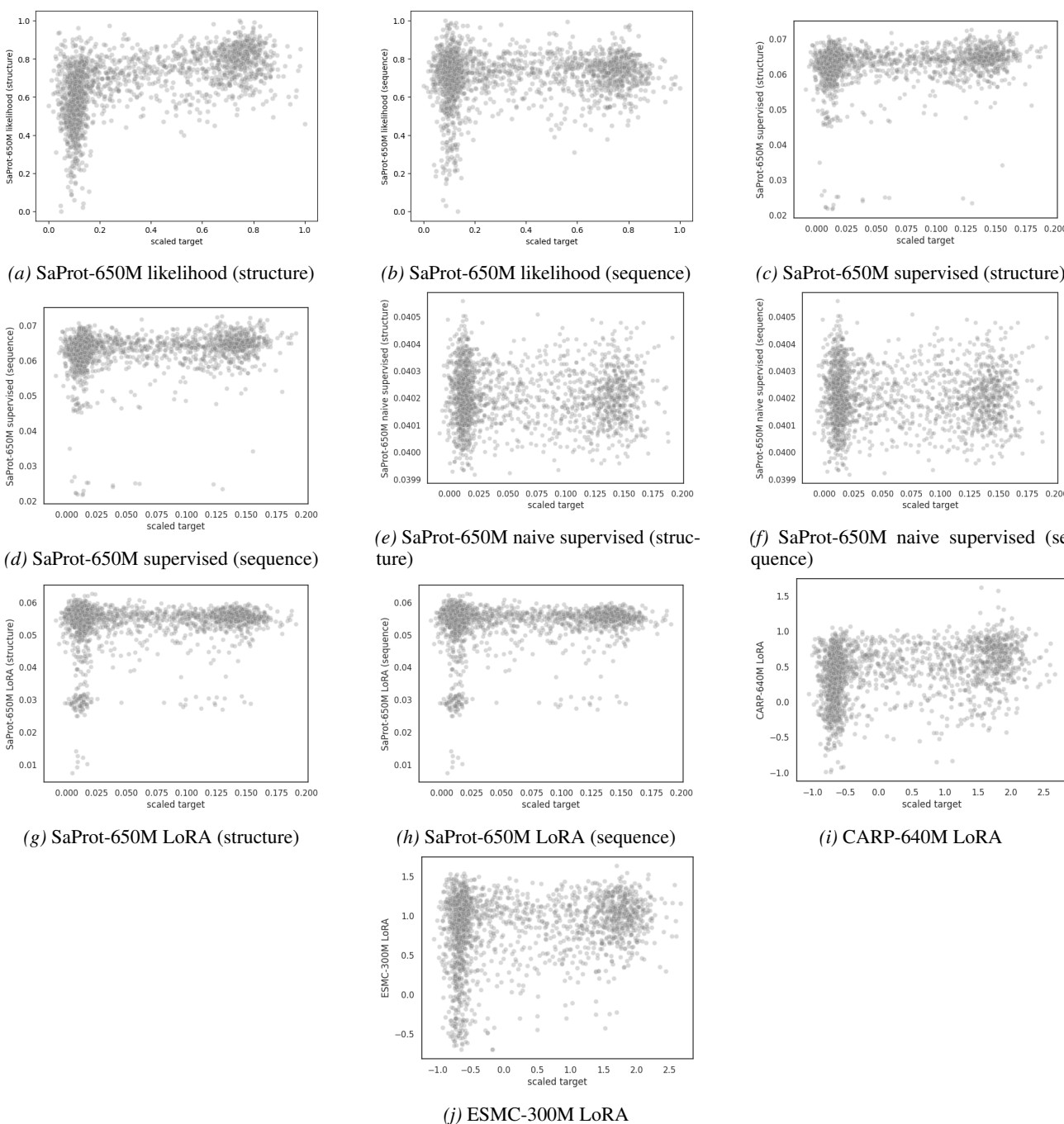

*Figure A4.* Predictions for amylase close-to-far (continued). Finetuned PLM predictions are averaged over 5 random seeds.

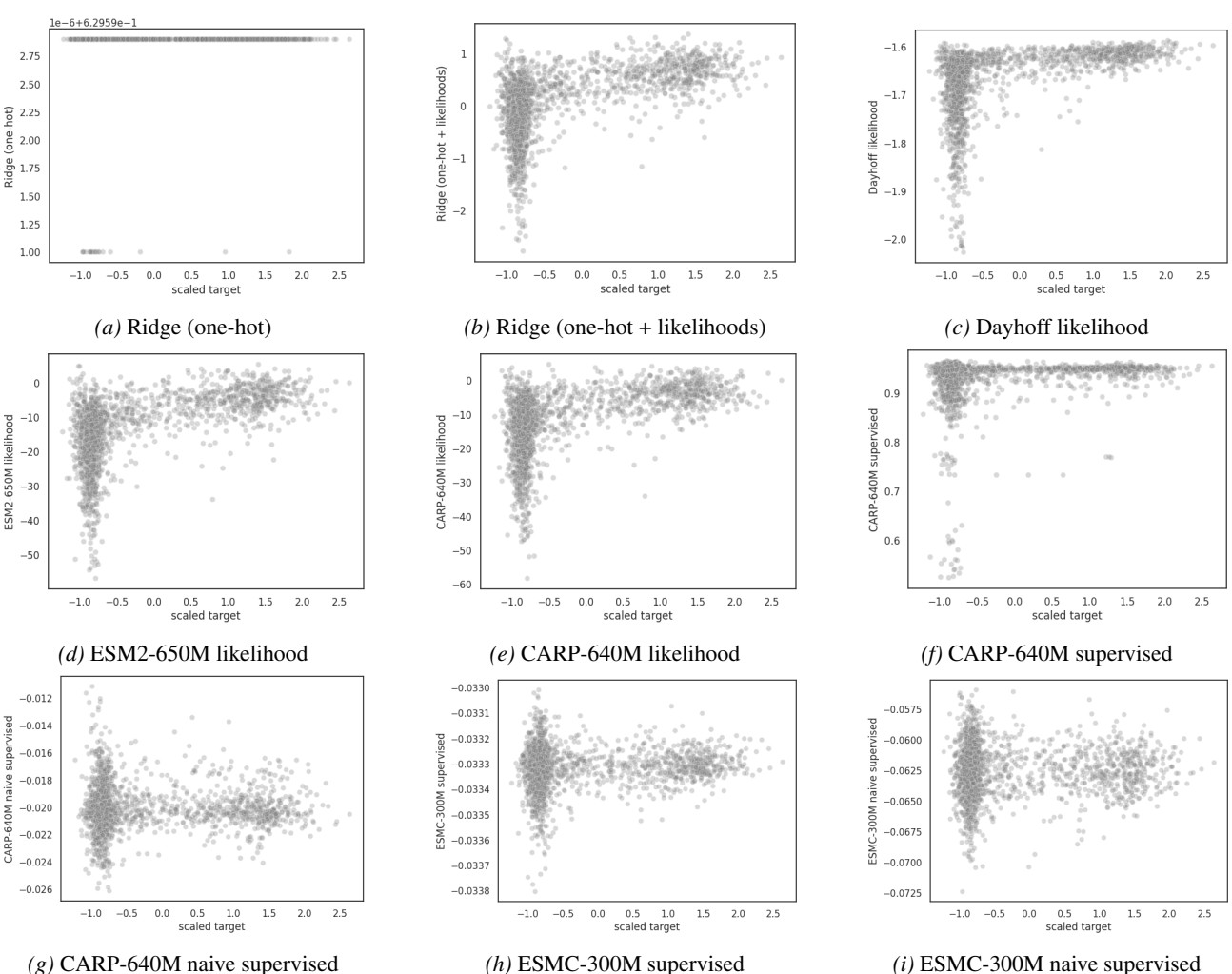

*(a)* Ridge (one-hot)     *(b)* Ridge (one-hot + likelihoods)     *(c)* Dayhoff likelihood

*(d)* ESM2-650M likelihood     *(e)* CARP-640M likelihood     *(f)* CARP-640M supervised

*(g)* CARP-640M naive supervised     *(h)* ESMC-300M supervised     *(i)* ESMC-300M naive supervised

*Figure A5.* Predictions for amylase far-to-close. Finetuned PLM predictions are averaged over 5 random seeds.

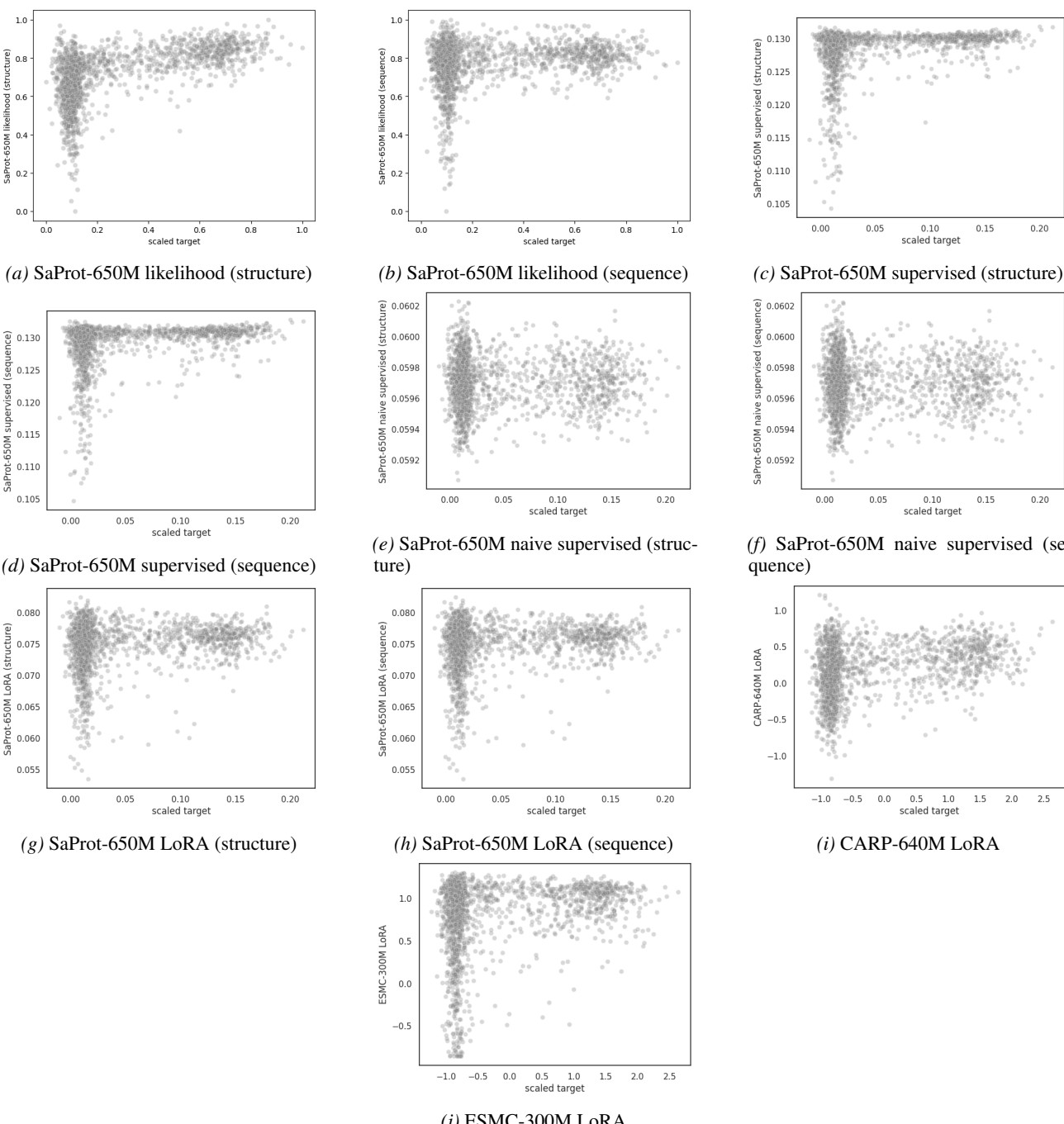

*(a)* SaProt-650M likelihood (structure)

*(b)* SaProt-650M likelihood (sequence)

*(c)* SaProt-650M supervised (structure)

*(d)* SaProt-650M supervised (sequence)

*(e)* SaProt-650M naive supervised (structure)

*(f)* SaProt-650M naive supervised (sequence)

*(g)* SaProt-650M LoRA (structure)

*(h)* SaProt-650M LoRA (sequence)

*(i)* CARP-640M LoRA

*(j)* ESMC-300M LoRA

*Figure A6.* Predictions for amylase far-to-close (continued). Finetuned PLM predictions are averaged over 5 random seeds.

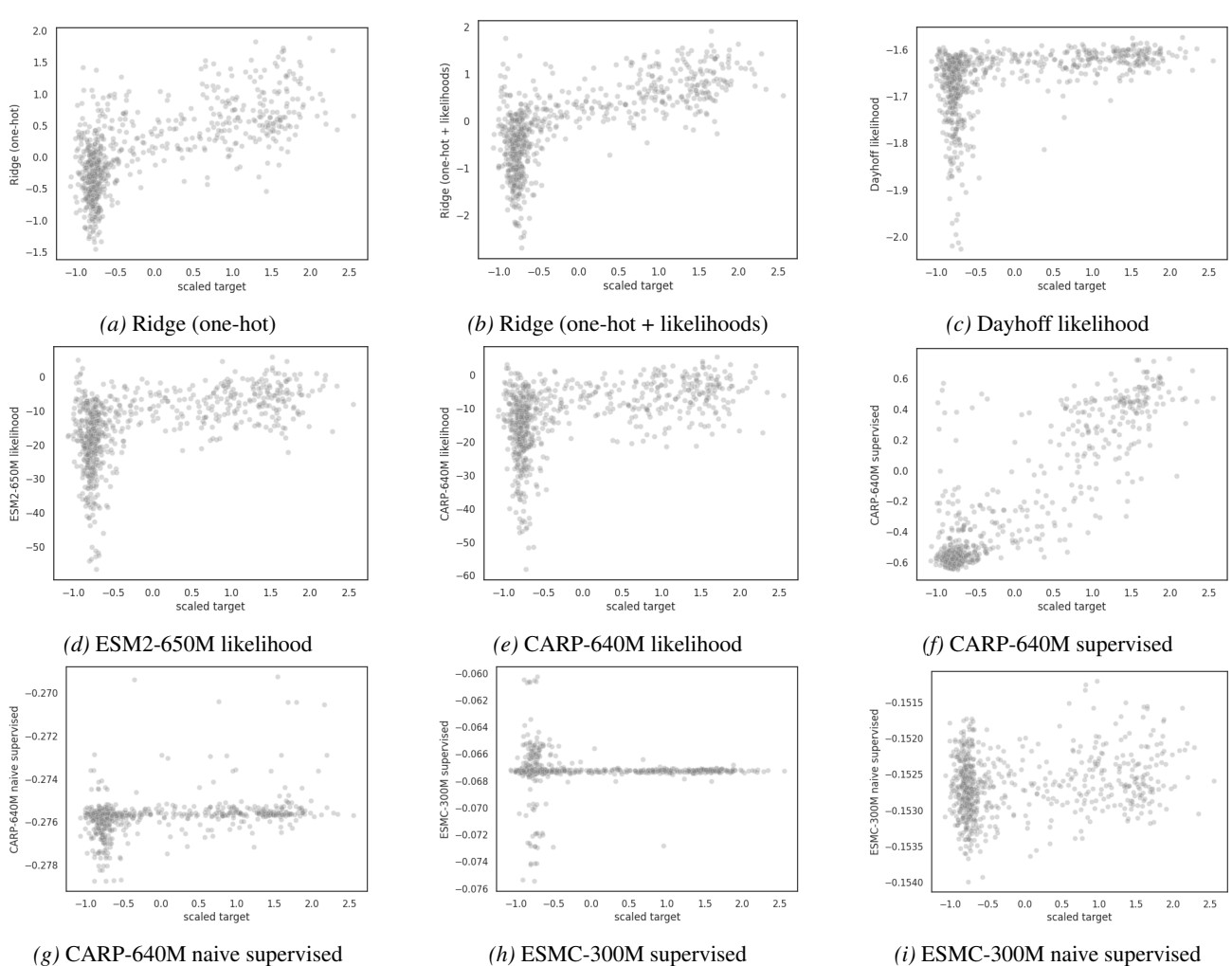

*(a)* Ridge (one-hot)  *(b)* Ridge (one-hot + likelihoods)  *(c)* Dayhoff likelihood

*(d)* ESM2-650M likelihood  *(e)* CARP-640M likelihood  *(f)* CARP-640M supervised

*(g)* CARP-640M naive supervised  *(h)* ESMC-300M supervised  *(i)* ESMC-300M naive supervised

*Figure A7.* Predictions for amylase by-mutation. Finetuned PLM predictions are averaged over 5 random seeds.

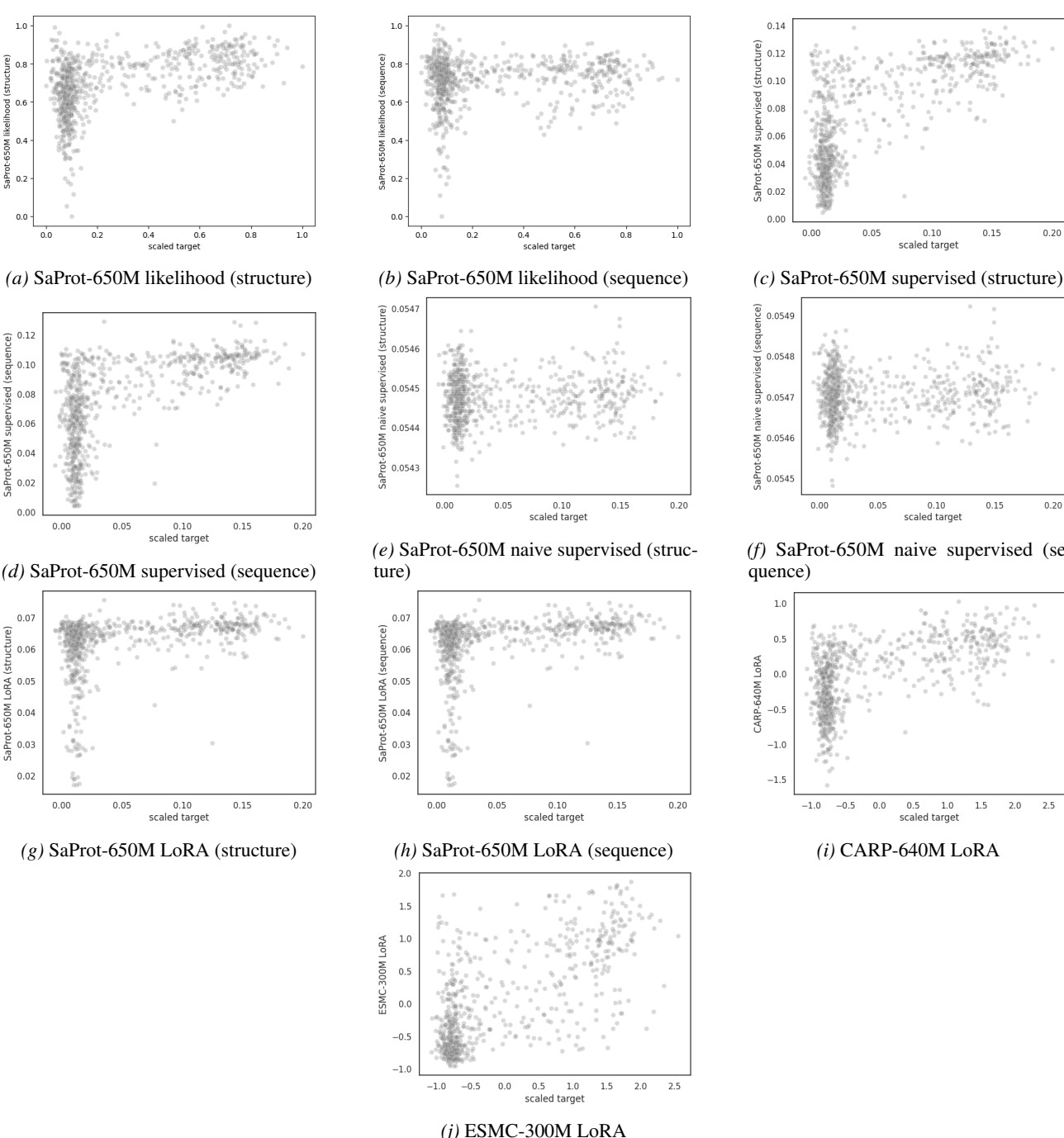

*(a)* SaProt-650M likelihood (structure)

*(b)* SaProt-650M likelihood (sequence)

*(c)* SaProt-650M supervised (structure)

*(d)* SaProt-650M supervised (sequence)

*(e)* SaProt-650M naive supervised (structure)

*(f)* SaProt-650M naive supervised (sequence)

*(g)* SaProt-650M LoRA (structure)

*(h)* SaProt-650M LoRA (sequence)

*(i)* CARP-640M LoRA

*(j)* ESMC-300M LoRA

*Figure A8.* Predictions for amylase by-mutation (continued). Finetuned PLM predictions are averaged over 5 random seeds.

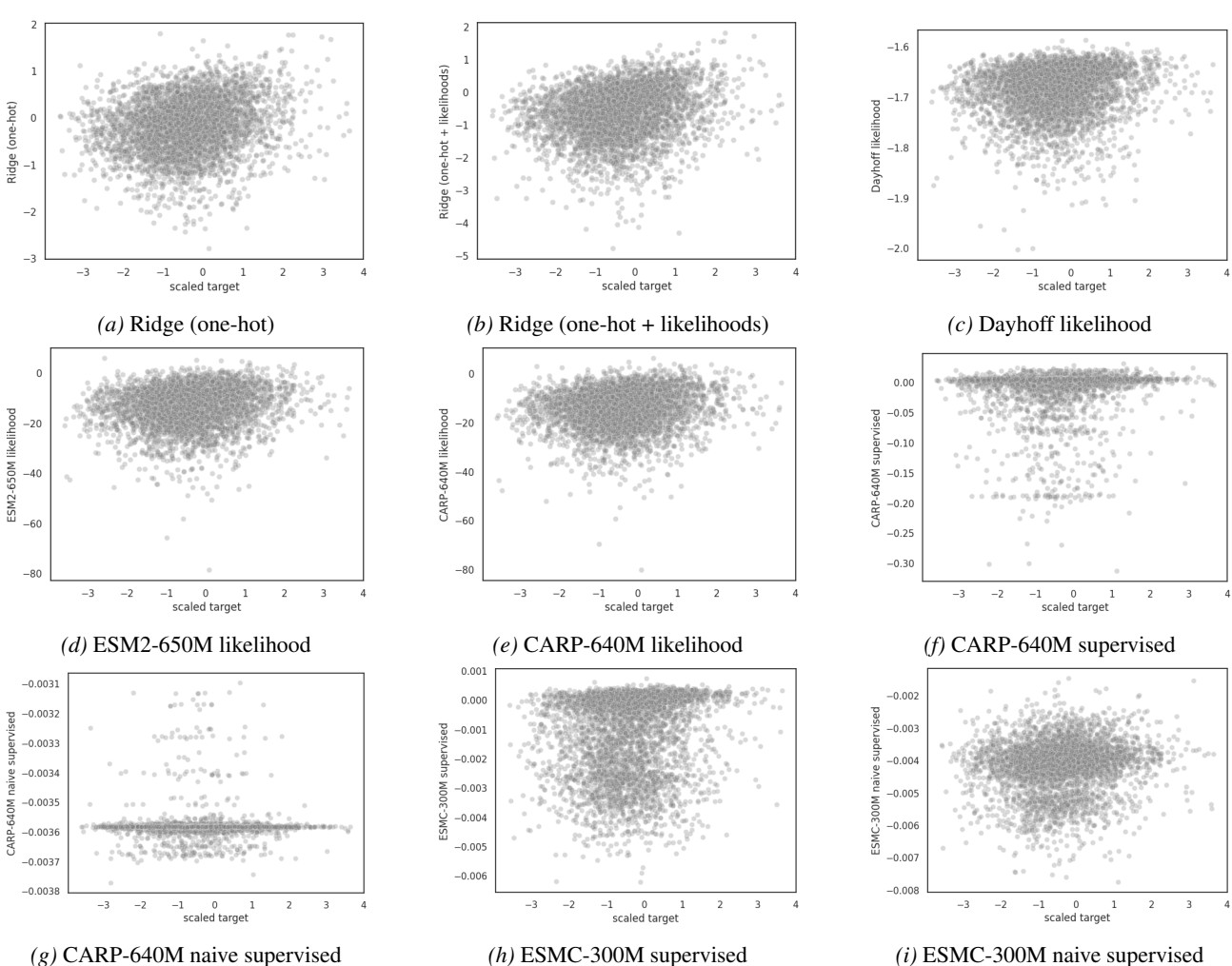

*(a)* Ridge (one-hot)  *(b)* Ridge (one-hot + likelihoods)  *(c)* Dayhoff likelihood

*(d)* ESM2-650M likelihood  *(e)* CARP-640M likelihood  *(f)* CARP-640M supervised

*(g)* CARP-640M naive supervised  *(h)* ESMC-300M supervised  *(i)* ESMC-300M naive supervised

*Figure A9.* Predictions for IRED two-to-many. Finetuned PLM predictions are averaged over 5 random seeds.

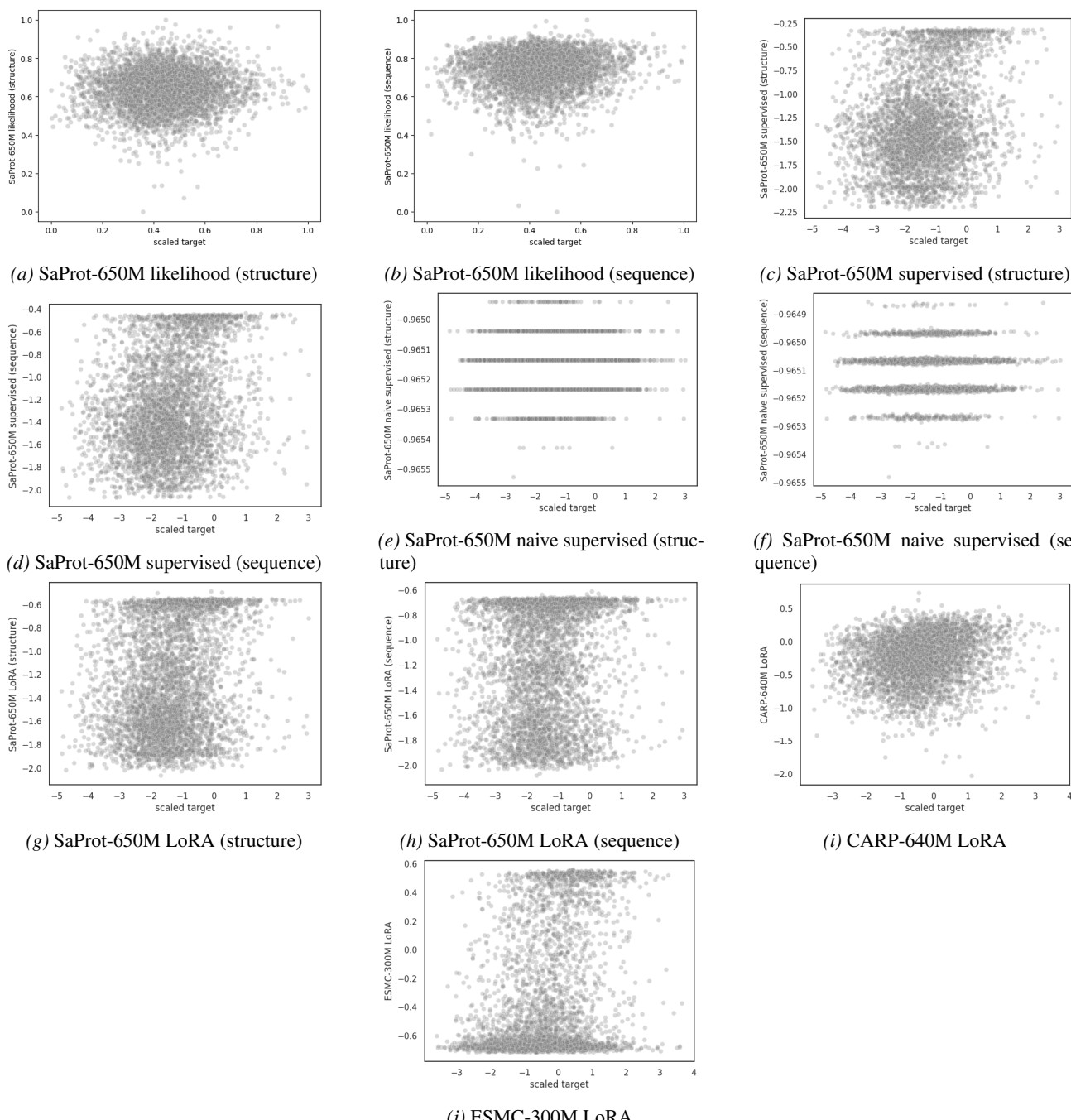

*(a)* SaProt-650M likelihood (structure)

*(b)* SaProt-650M likelihood (sequence)

*(c)* SaProt-650M supervised (structure)

*(d)* SaProt-650M supervised (sequence)

*(e)* SaProt-650M naive supervised (structure)

*(f)* SaProt-650M naive supervised (sequence)

*(g)* SaProt-650M LoRA (structure)

*(h)* SaProt-650M LoRA (sequence)

*(i)* CARP-640M LoRA

*(j)* ESMC-300M LoRA

*Figure A10.* Predictions for IRED two-to-many (continued). Finetuned PLM predictions are averaged over 5 random seeds.

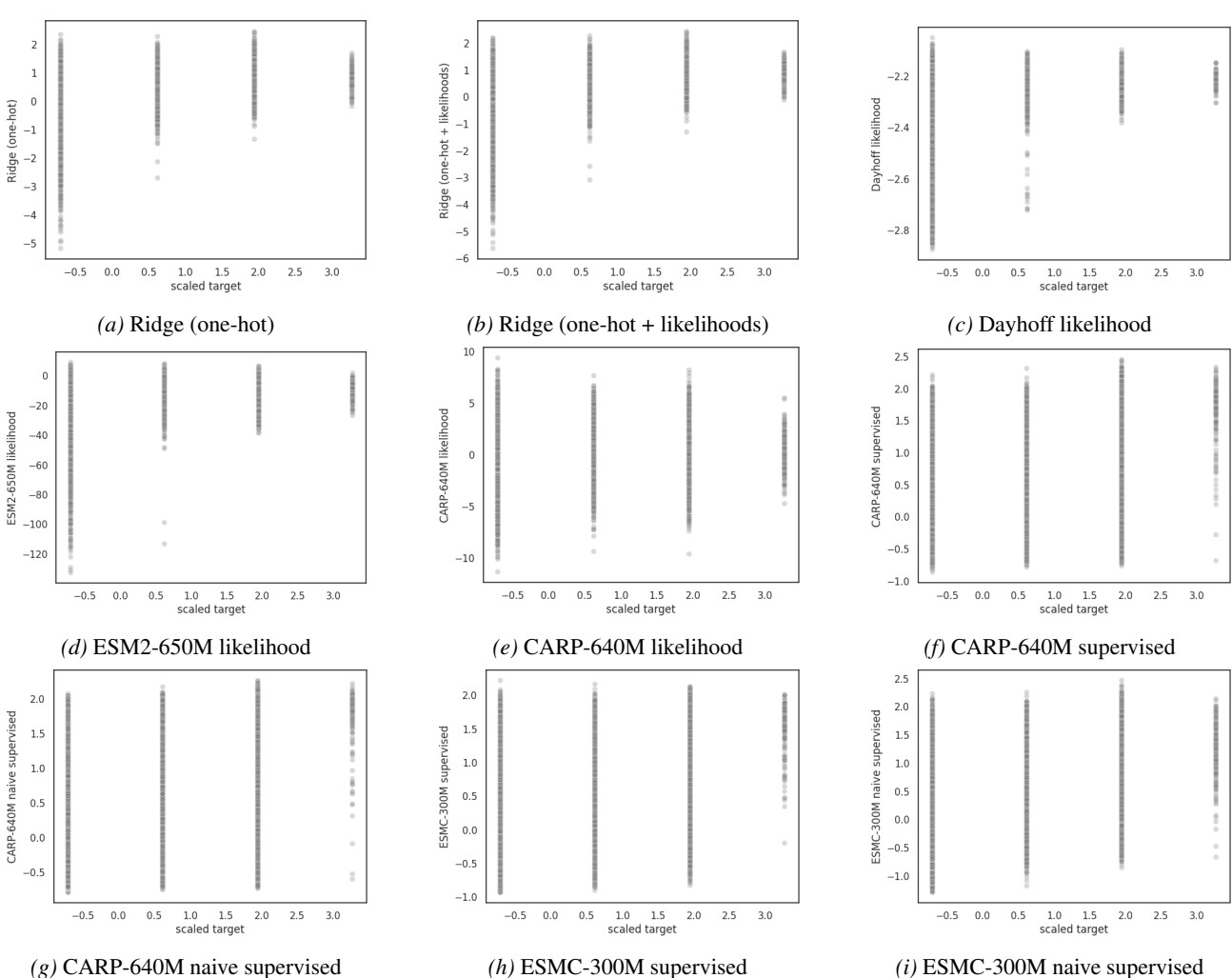

*(a)* Ridge (one-hot)  *(b)* Ridge (one-hot + likelihoods)  *(c)* Dayhoff likelihood

*(d)* ESM2-650M likelihood  *(e)* CARP-640M likelihood  *(f)* CARP-640M supervised

*(g)* CARP-640M naive supervised  *(h)* ESMC-300M supervised  *(i)* ESMC-300M naive supervised

*Figure A11.* Predictions for NucB two-to-many. Finetuned PLM predictions are averaged over 5 random seeds.

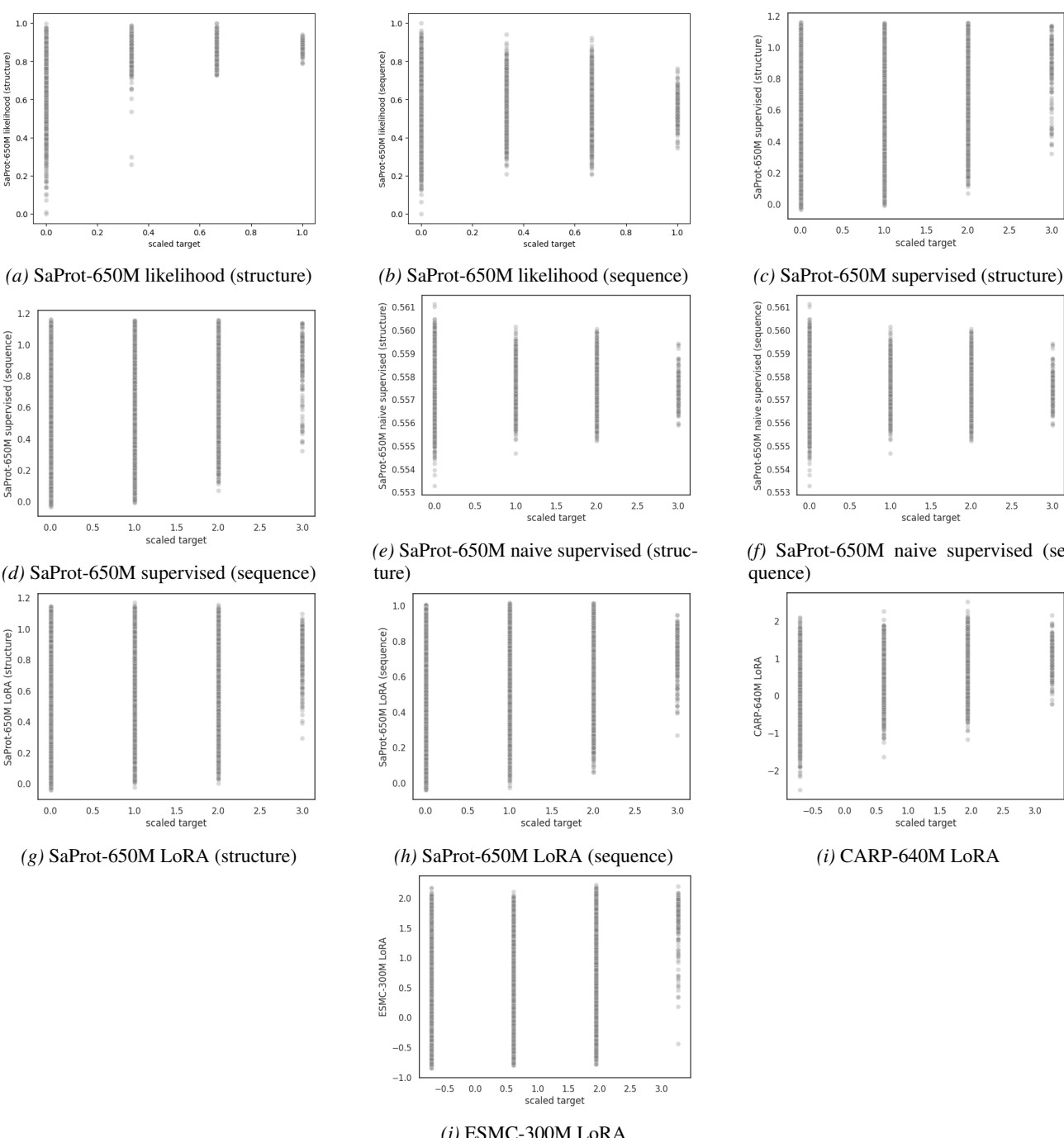

*(a)* SaProt-650M likelihood (structure)

*(b)* SaProt-650M likelihood (sequence)

*(c)* SaProt-650M supervised (structure)

*(d)* SaProt-650M supervised (sequence)

*(e)* SaProt-650M naive supervised (structure)

*(f)* SaProt-650M naive supervised (sequence)

*(g)* SaProt-650M LoRA (structure)

*(h)* SaProt-650M LoRA (sequence)

*(i)* CARP-640M LoRA

*(j)* ESMC-300M LoRA

*Figure A12.* Predictions for NucB two-to-many (continued). Finetuned PLM predictions are averaged over 5 random seeds.

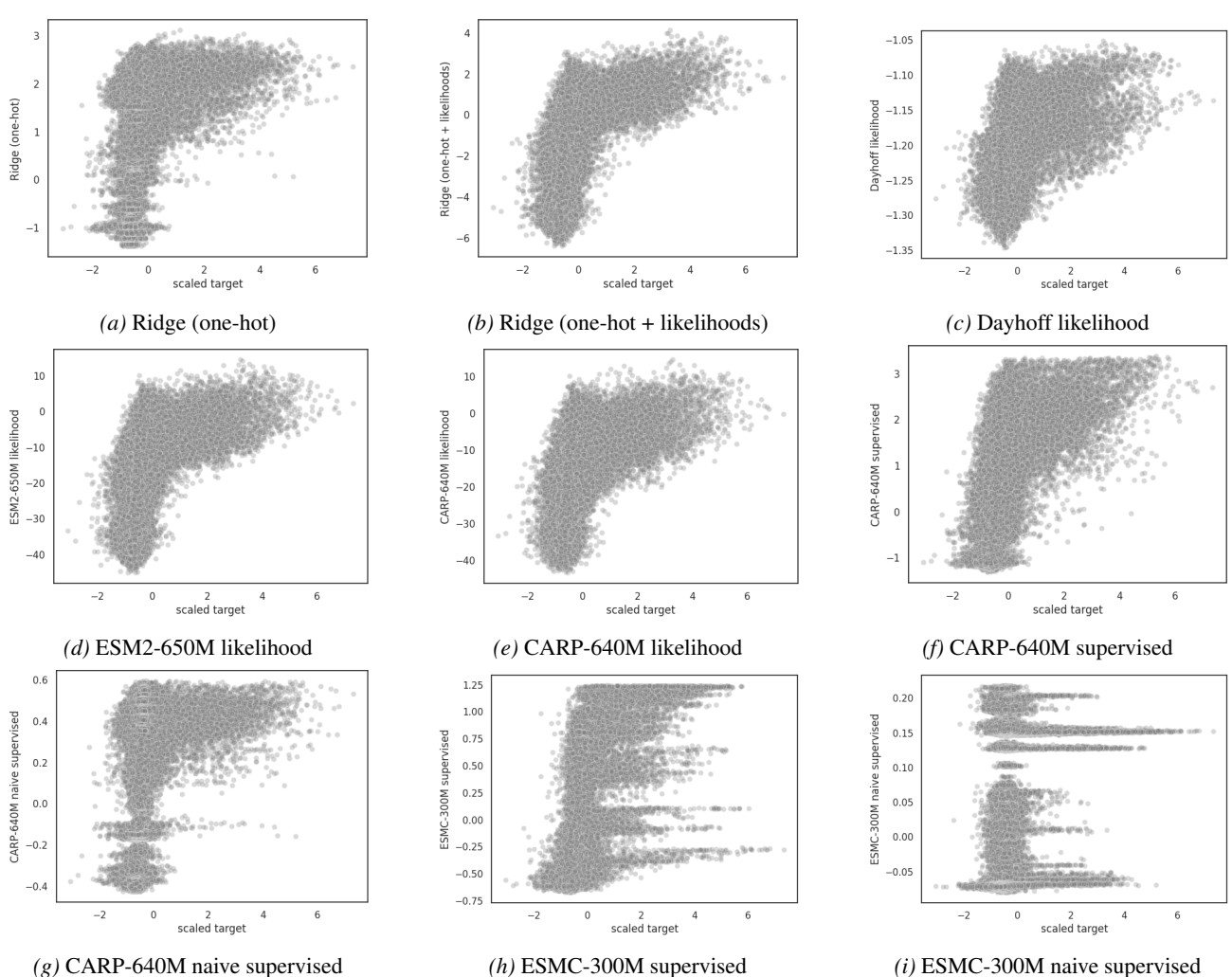

*Figure A13.* Predictions for TrpB one-to-many. Finetuned PLM predictions are averaged over 5 random seeds.

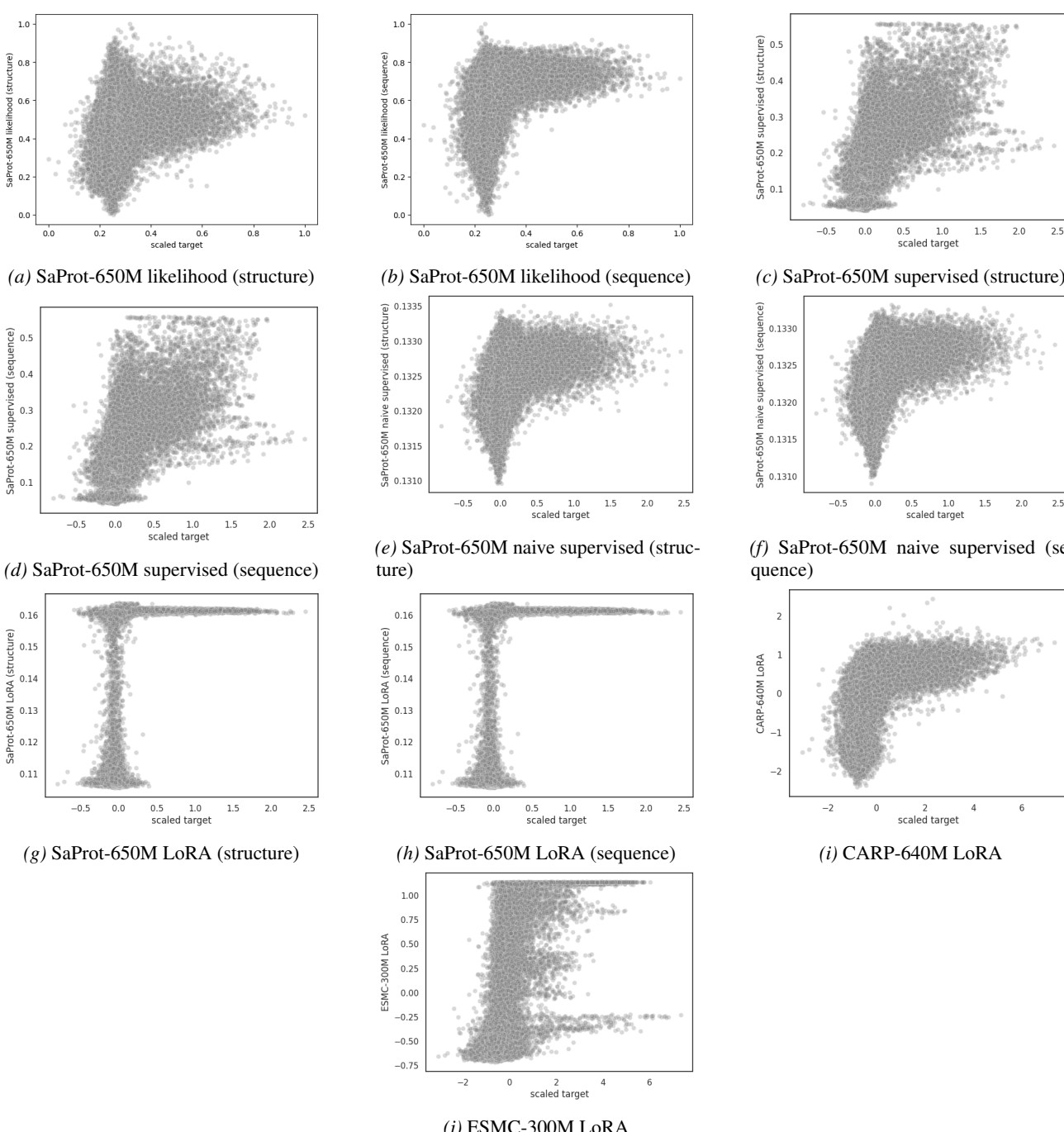

*(a)* SaProt-650M likelihood (structure)

*(b)* SaProt-650M likelihood (sequence)

*(c)* SaProt-650M supervised (structure)

*(d)* SaProt-650M supervised (sequence)

*(e)* SaProt-650M naive supervised (structure)

*(f)* SaProt-650M naive supervised (sequence)

*(g)* SaProt-650M LoRA (structure)

*(h)* SaProt-650M LoRA (sequence)

*(i)* CARP-640M LoRA

*(j)* ESMC-300M LoRA

*Figure A14.* Predictions for TrpB one-to-many (continued). Finetuned PLM predictions are averaged over 5 random seeds.

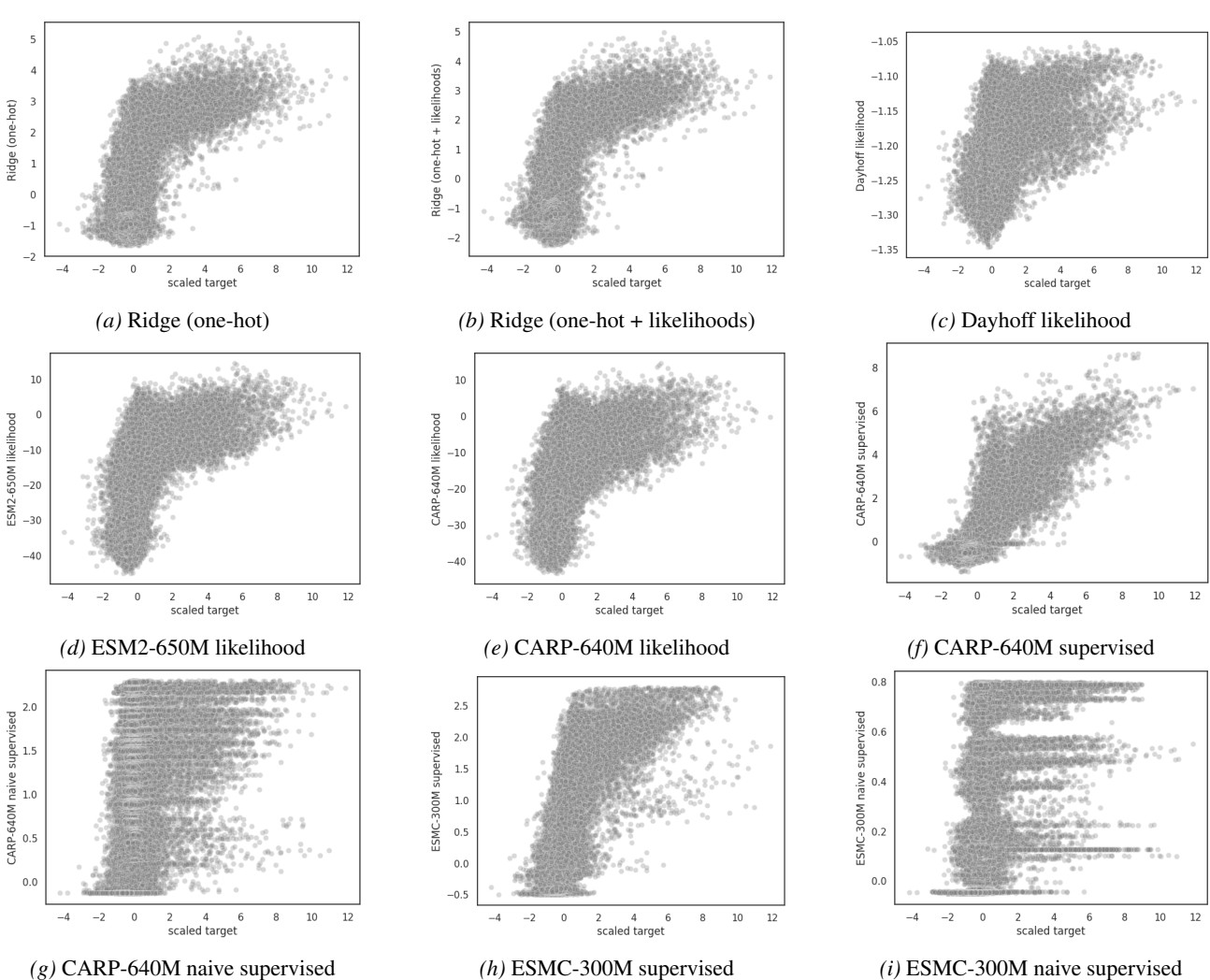

*(a)* Ridge (one-hot)                *(b)* Ridge (one-hot + likelihoods)                *(c)* Dayhoff likelihood

*(d)* ESM2-650M likelihood           *(e)* CARP-640M likelihood                        *(f)* CARP-640M supervised

*(g)* CARP-640M naive supervised     *(h)* ESMC-300M supervised                        *(i)* ESMC-300M naive supervised

*Figure A15.* Predictions for TrpB two-to-many. Finetuned PLM predictions are averaged over 5 random seeds.

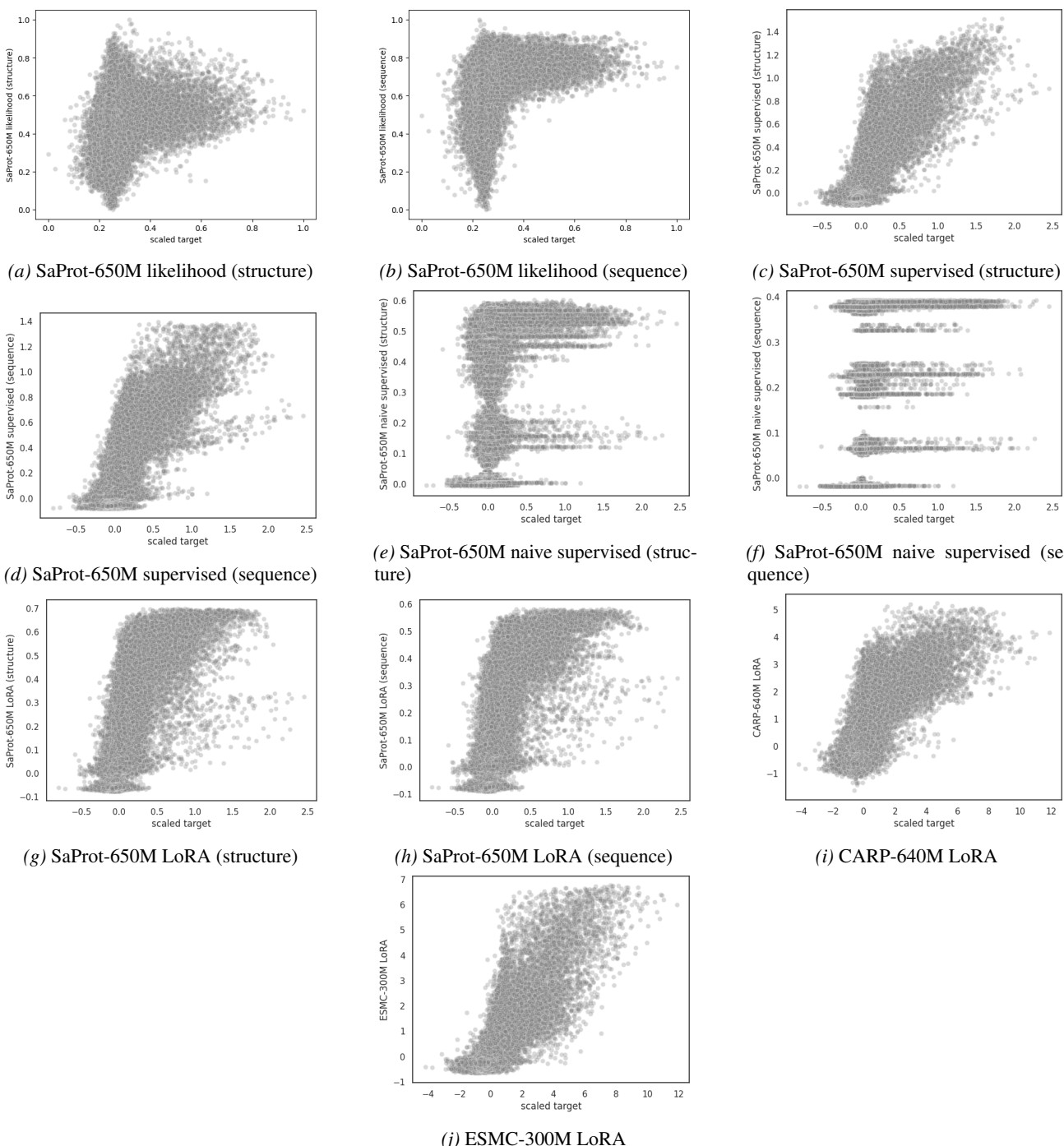

*(a)* SaProt-650M likelihood (structure)

*(b)* SaProt-650M likelihood (sequence)

*(c)* SaProt-650M supervised (structure)

*(d)* SaProt-650M supervised (sequence)

*(e)* SaProt-650M naive supervised (structure)

*(f)* SaProt-650M naive supervised (sequence)

*(g)* SaProt-650M LoRA (structure)

*(h)* SaProt-650M LoRA (sequence)

*(i)* CARP-640M LoRA

*(j)* ESMC-300M LoRA

*Figure A16.* Predictions for TrpB two-to-many (continued). Finetuned PLM predictions are averaged over 5 random seeds.

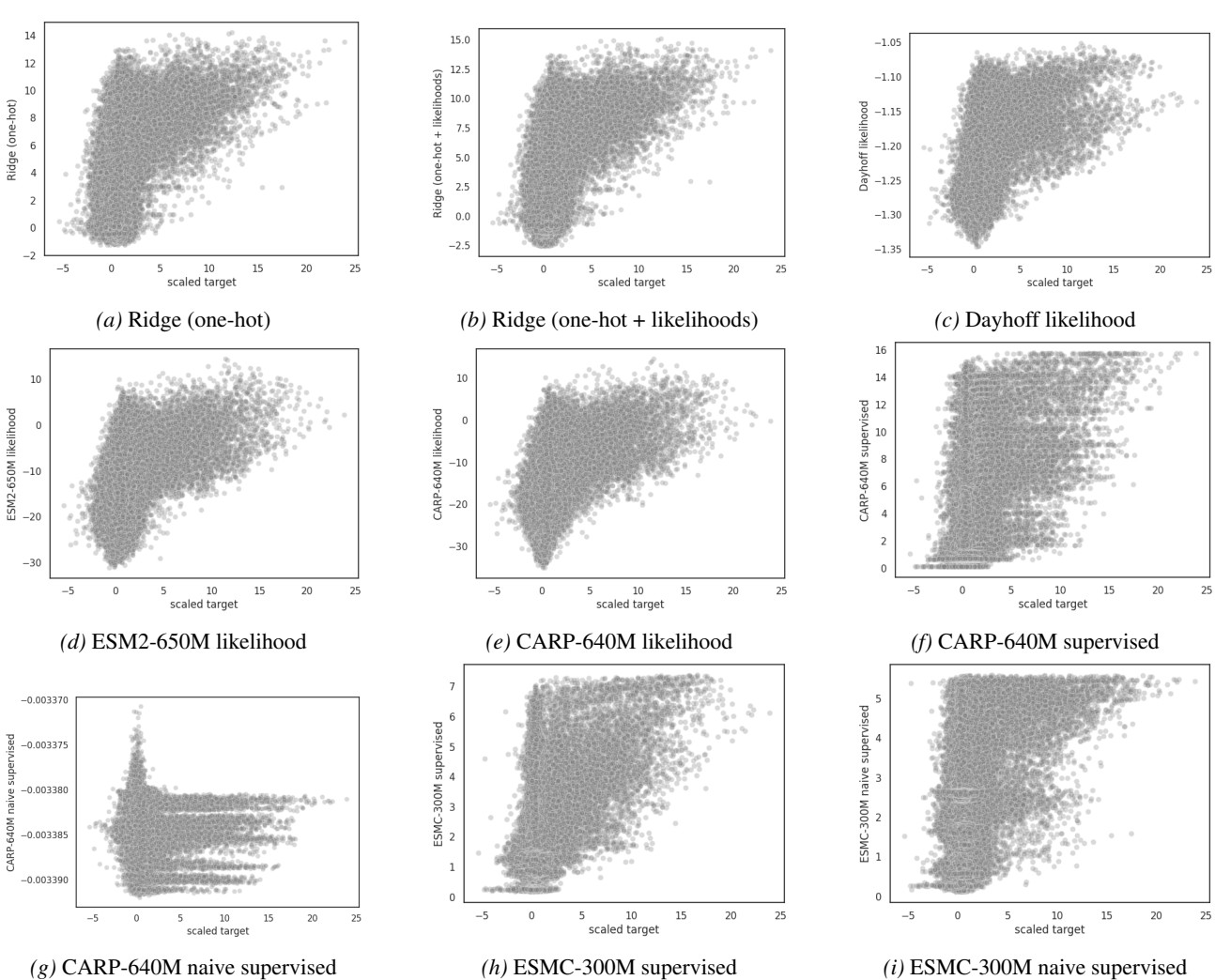

*Figure A17.* Predictions for TrpB by-position. Finetuned PLM predictions are averaged over 5 random seeds.

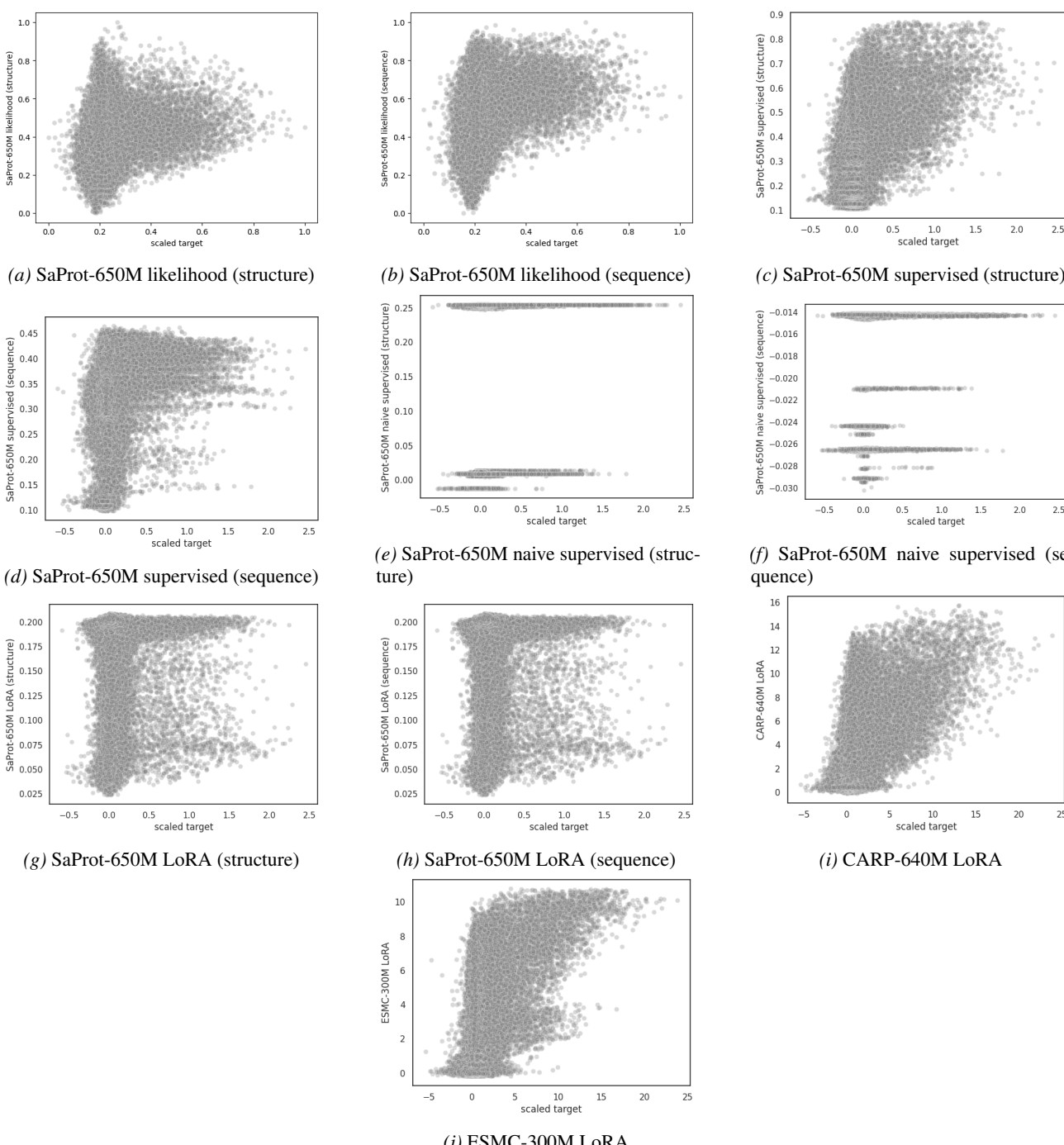

*(a)* SaProt-650M likelihood (structure)

*(b)* SaProt-650M likelihood (sequence)

*(c)* SaProt-650M supervised (structure)

*(d)* SaProt-650M supervised (sequence)

*(e)* SaProt-650M naive supervised (structure)

*(f)* SaProt-650M naive supervised (sequence)

*(g)* SaProt-650M LoRA (structure)

*(h)* SaProt-650M LoRA (sequence)

*(i)* CARP-640M LoRA

*(j)* ESMC-300M LoRA

*Figure A18.* Predictions for TrpB by-position (continued). Finetuned PLM predictions are averaged over 5 random seeds.

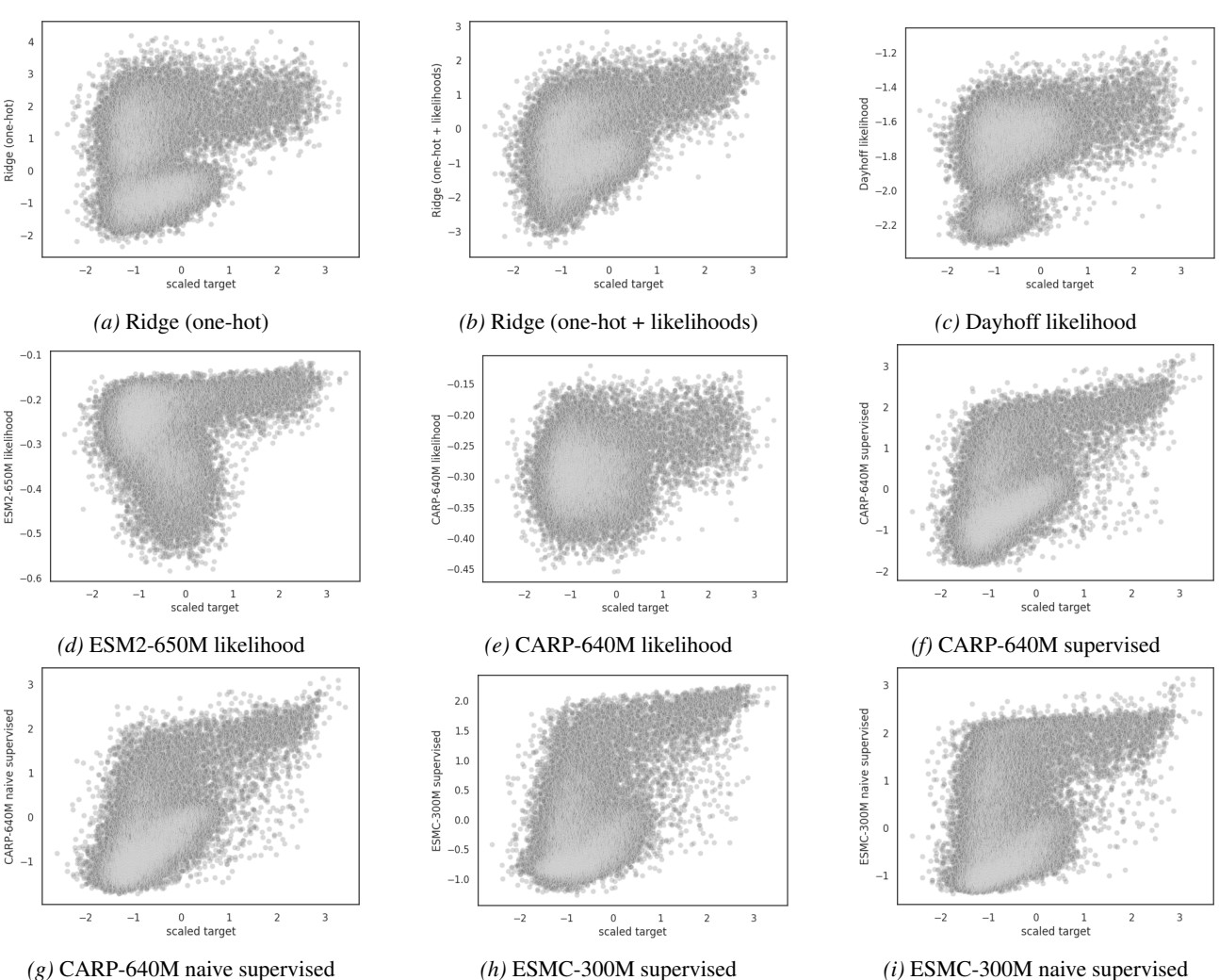

*(a)* Ridge (one-hot)  *(b)* Ridge (one-hot + likelihoods)  *(c)* Dayhoff likelihood

*(d)* ESM2-650M likelihood  *(e)* CARP-640M likelihood  *(f)* CARP-640M supervised

*(g)* CARP-640M naive supervised  *(h)* ESMC-300M supervised  *(i)* ESMC-300M naive supervised

*Figure A19.* Predictions for hydrophobic core three-to-many. Finetuned PLM predictions are averaged over 5 random seeds.

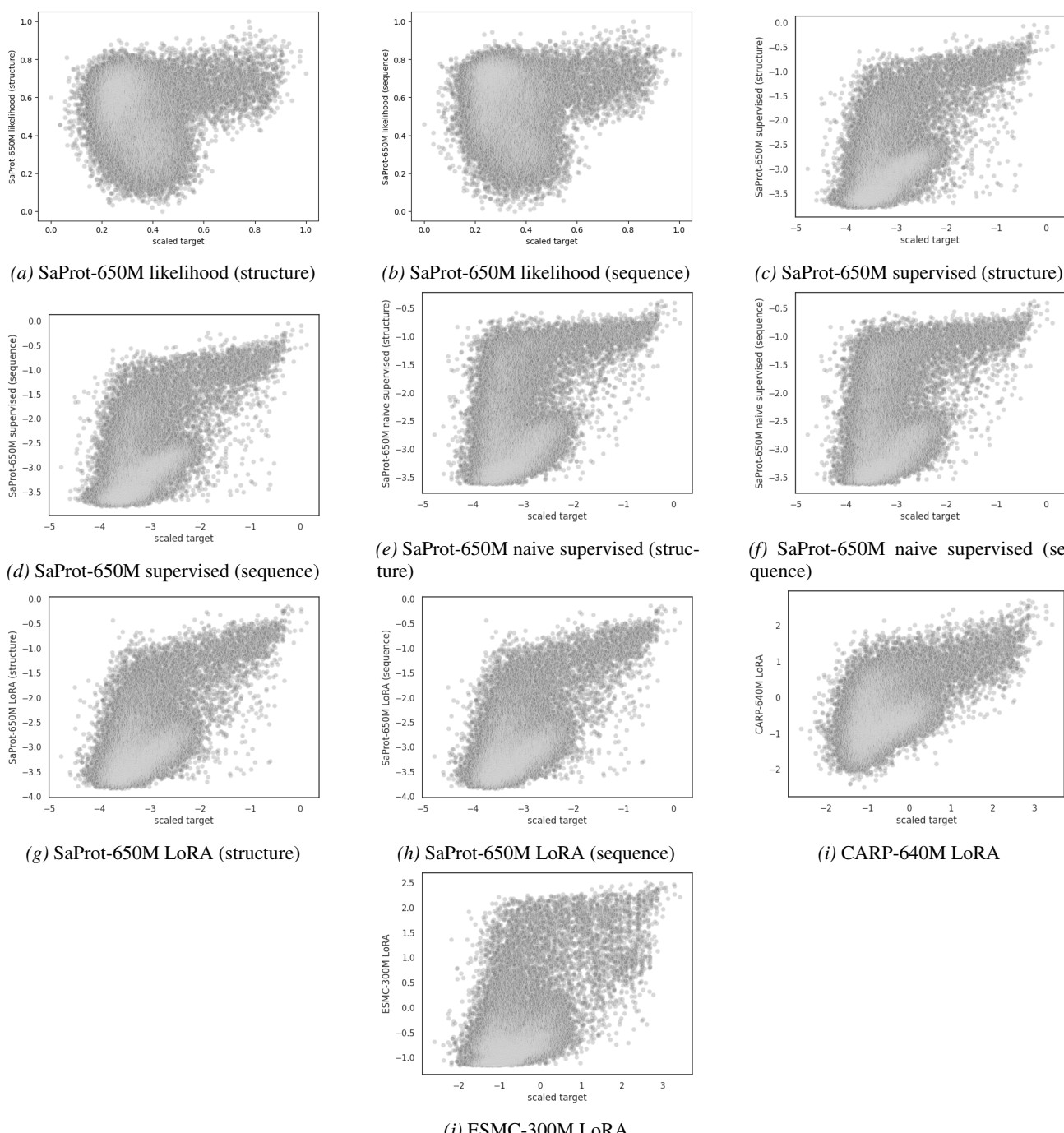

*(a)* SaProt-650M likelihood (structure)

*(b)* SaProt-650M likelihood (sequence)

*(c)* SaProt-650M supervised (structure)

*(d)* SaProt-650M supervised (sequence)

*(e)* SaProt-650M naive supervised (structure)

*(f)* SaProt-650M naive supervised (sequence)

*(g)* SaProt-650M LoRA (structure)

*(h)* SaProt-650M LoRA (sequence)

*(i)* CARP-640M LoRA

*(j)* ESMC-300M LoRA

*Figure A20.* Predictions for hydrophobic core three-to-many (continued). Finetuned PLM predictions are averaged over 5 random seeds.

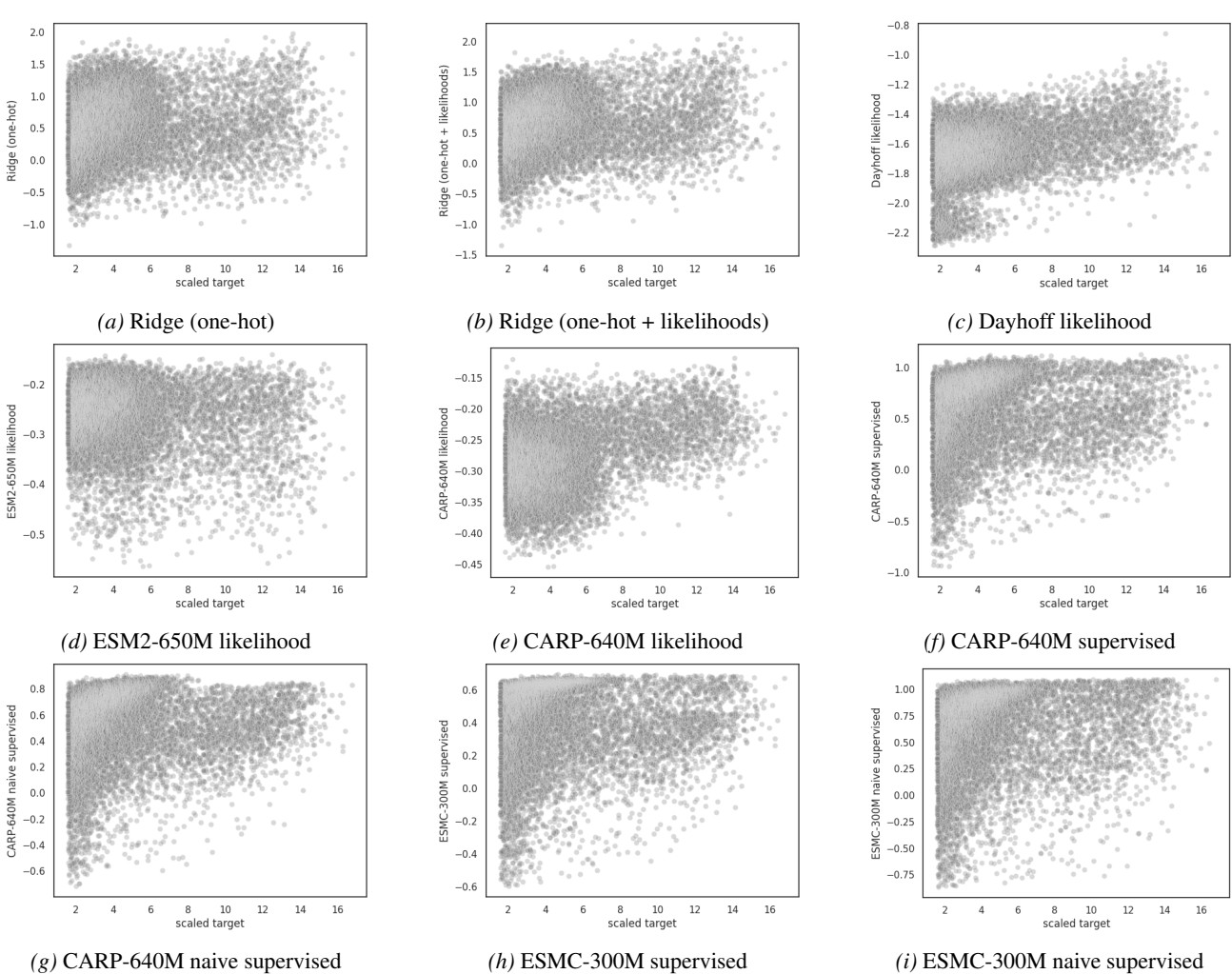

*(a)* Ridge (one-hot)    *(b)* Ridge (one-hot + likelihoods)    *(c)* Dayhoff likelihood

*(d)* ESM2-650M likelihood    *(e)* CARP-640M likelihood    *(f)* CARP-640M supervised

*(g)* CARP-640M naive supervised    *(h)* ESMC-300M supervised    *(i)* ESMC-300M naive supervised

*Figure A21.* Predictions for hydrophobic core low-to-high. Finetuned PLM predictions are averaged over 5 random seeds.

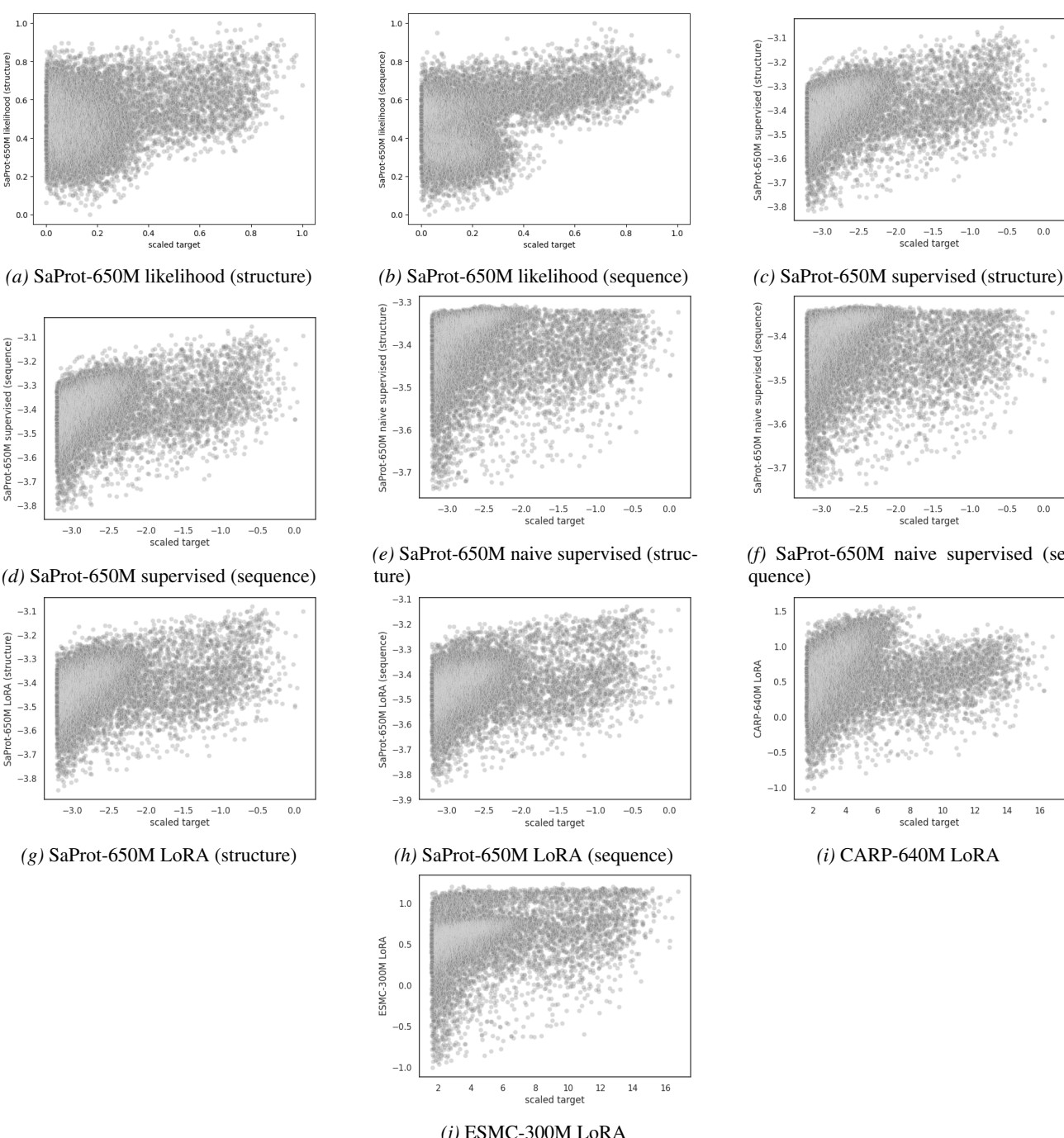

*Figure A22.* Predictions for hydrophobic core low-to-high (continued). Finetuned PLM predictions are averaged over 5 random seeds.

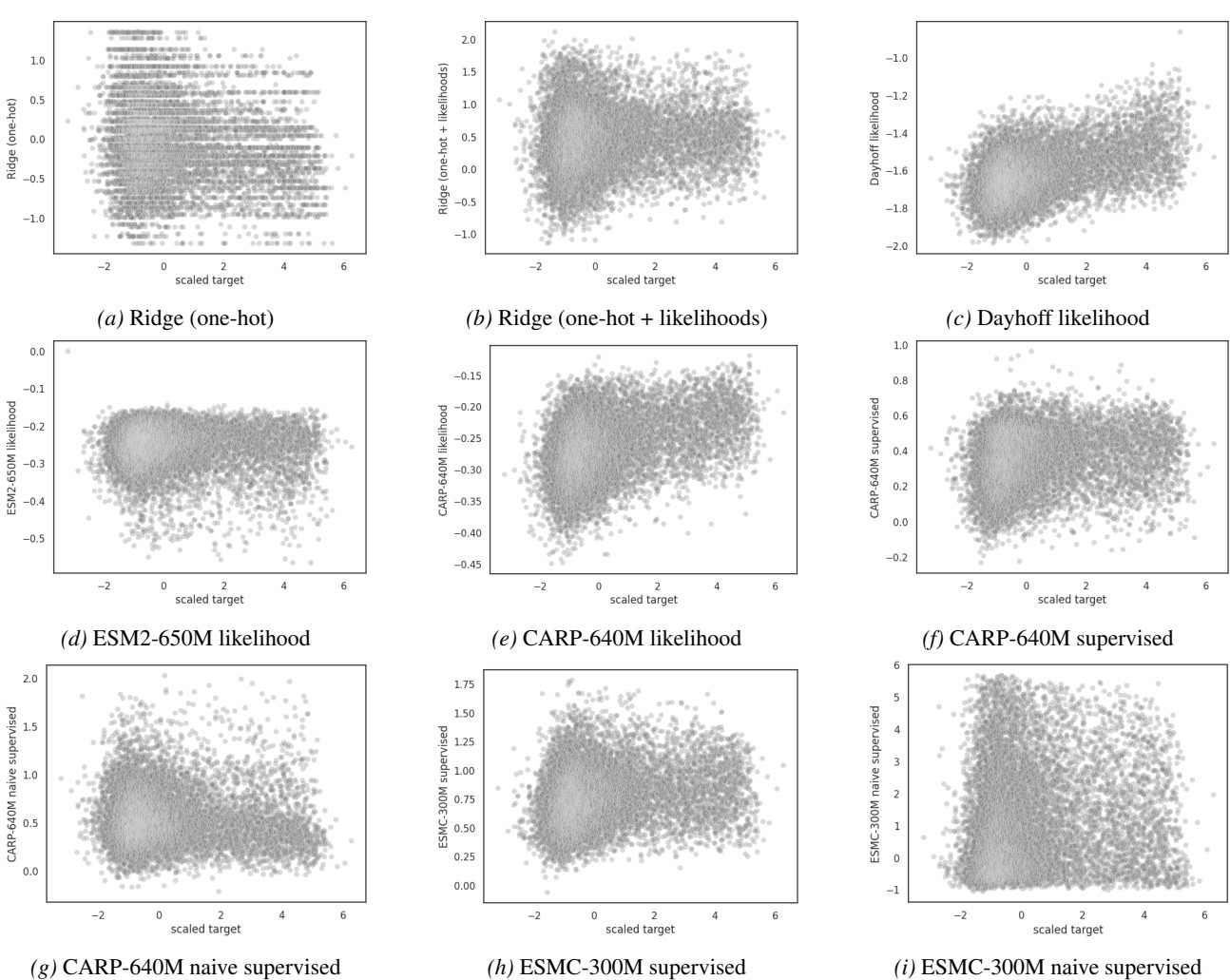

*(a)* Ridge (one-hot)  *(b)* Ridge (one-hot + likelihoods)  *(c)* Dayhoff likelihood

*(d)* ESM2-650M likelihood  *(e)* CARP-640M likelihood  *(f)* CARP-640M supervised

*(g)* CARP-640M naive supervised  *(h)* ESMC-300M supervised  *(i)* ESMC-300M naive supervised

*Figure A23.* Predictions for hydrophobic core to-P06241. Finetuned PLM predictions are averaged over 5 random seeds.

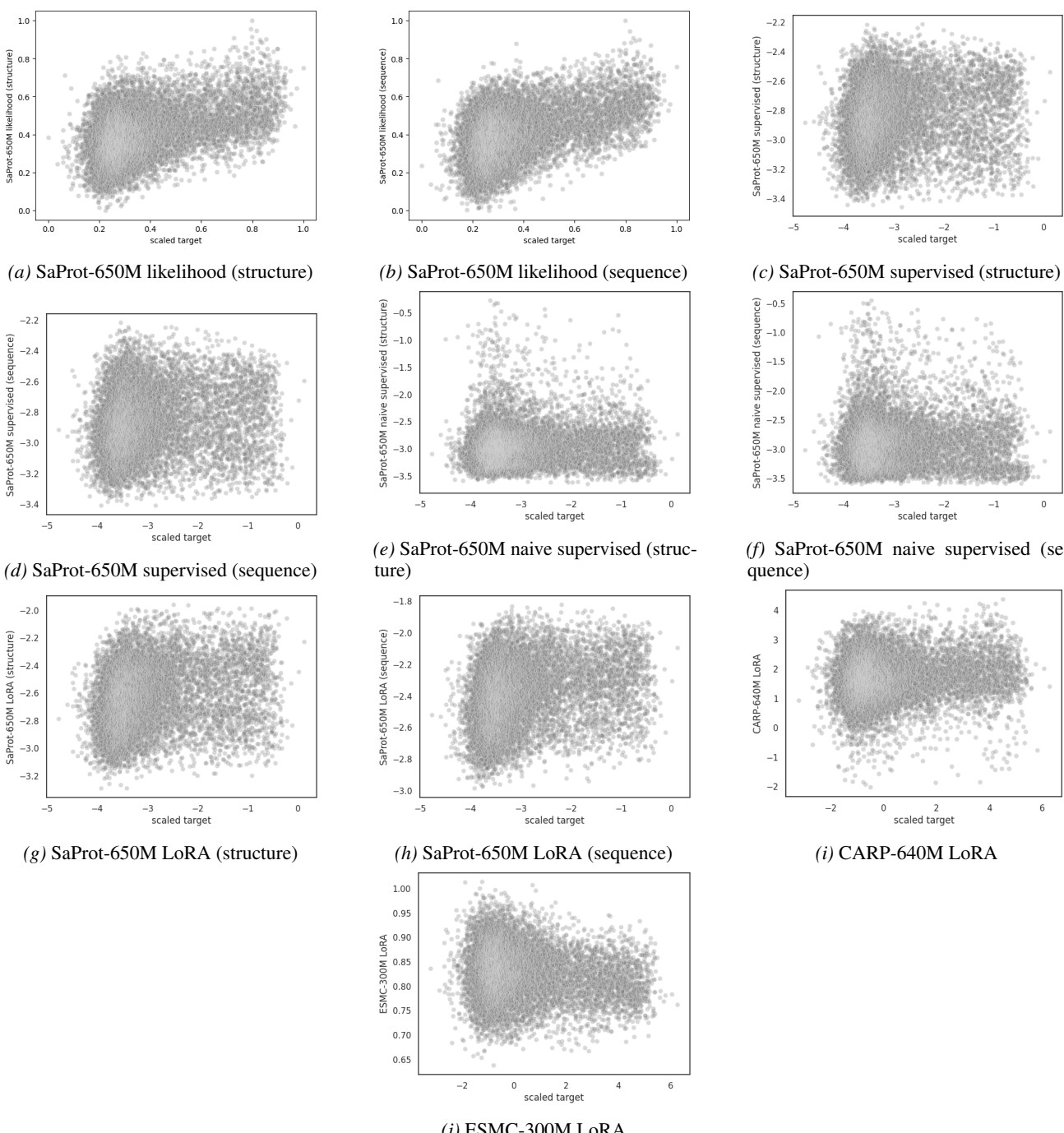

*(a)* SaProt-650M likelihood (structure)

*(b)* SaProt-650M likelihood (sequence)

*(c)* SaProt-650M supervised (structure)

*(d)* SaProt-650M supervised (sequence)

*(e)* SaProt-650M naive supervised (structure)

*(f)* SaProt-650M naive supervised (sequence)

*(g)* SaProt-650M LoRA (structure)

*(h)* SaProt-650M LoRA (sequence)

*(i)* CARP-640M LoRA

*(j)* ESMC-300M LoRA

*Figure A24.* Predictions for hydrophobic core to-P06241 (continued). Finetuned PLM predictions are averaged over 5 random seeds.

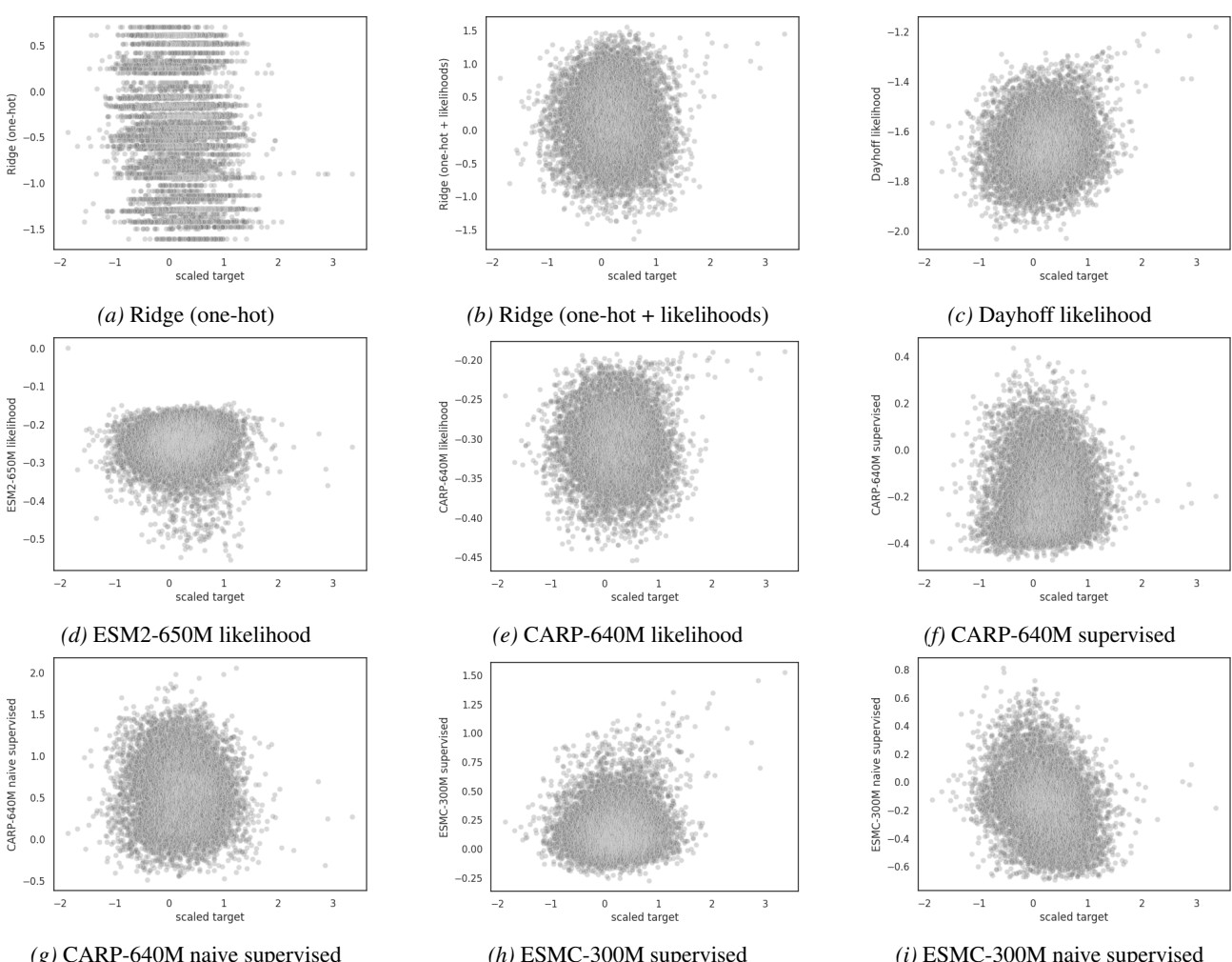

*(a)* Ridge (one-hot)          *(b)* Ridge (one-hot + likelihoods)          *(c)* Dayhoff likelihood

*(d)* ESM2-650M likelihood          *(e)* CARP-640M likelihood          *(f)* CARP-640M supervised

*(g)* CARP-640M naive supervised          *(h)* ESMC-300M supervised          *(i)* ESMC-300M naive supervised

*Figure A25.* Predictions for hydrophobic core to-P0A9X9. Finetuned PLM predictions are averaged over 5 random seeds.

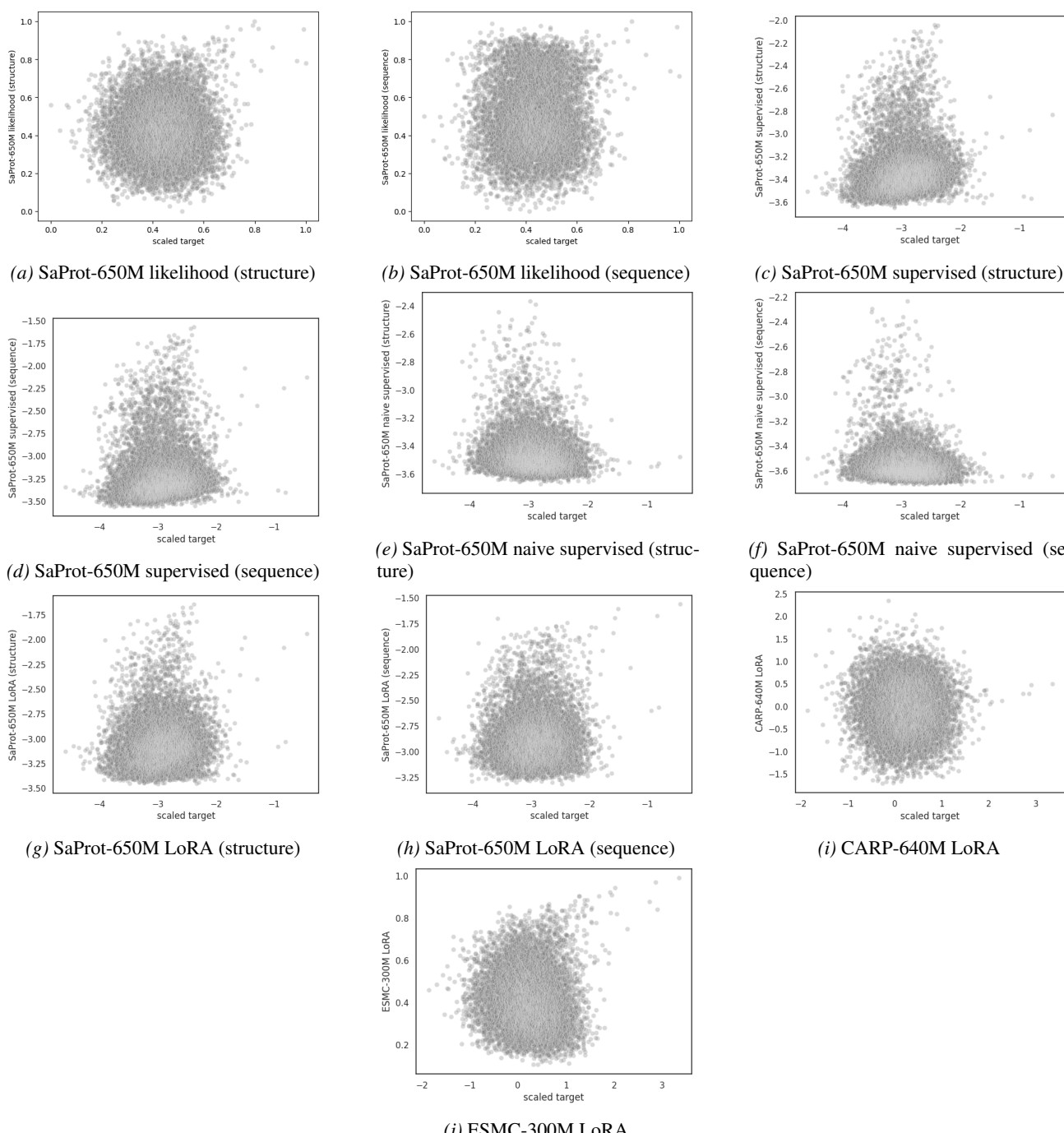

*(a)* SaProt-650M likelihood (structure)

*(b)* SaProt-650M likelihood (sequence)

*(c)* SaProt-650M supervised (structure)

*(d)* SaProt-650M supervised (sequence)

*(e)* SaProt-650M naive supervised (structure)

*(f)* SaProt-650M naive supervised (sequence)

*(g)* SaProt-650M LoRA (structure)

*(h)* SaProt-650M LoRA (sequence)

*(i)* CARP-640M LoRA

*(j)* ESMC-300M LoRA

*Figure A26.* Predictions for hydrophobic core to-P0A9X9 (continued). Finetuned PLM predictions are averaged over 5 random seeds.

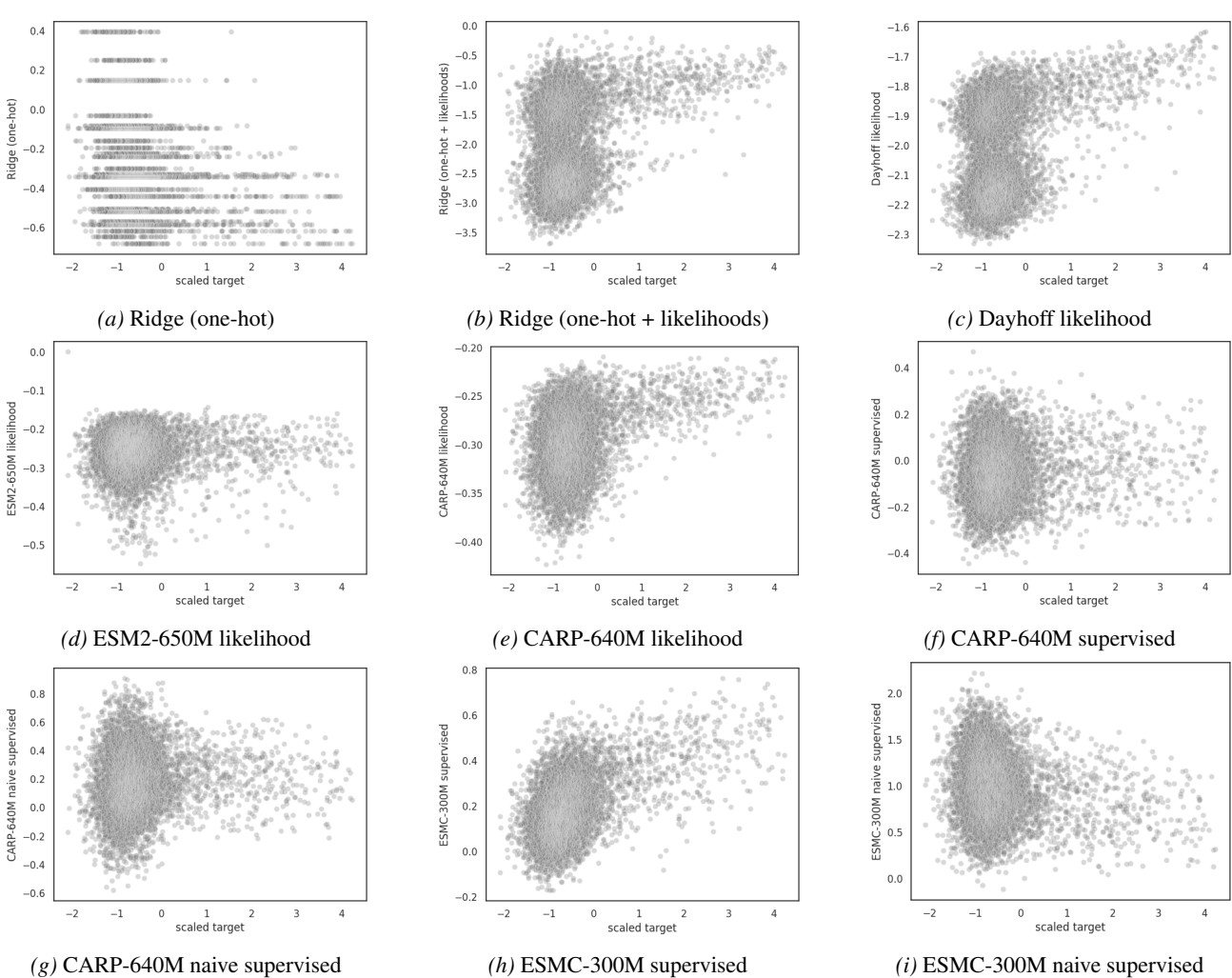

*(a)* Ridge (one-hot)  *(b)* Ridge (one-hot + likelihoods)  *(c)* Dayhoff likelihood

*(d)* ESM2-650M likelihood  *(e)* CARP-640M likelihood  *(f)* CARP-640M supervised

*(g)* CARP-640M naive supervised  *(h)* ESMC-300M supervised  *(i)* ESMC-300M naive supervised

*Figure A27.* Predictions for hydrophobic core to-P01053. Finetuned PLM predictions are averaged over 5 random seeds.

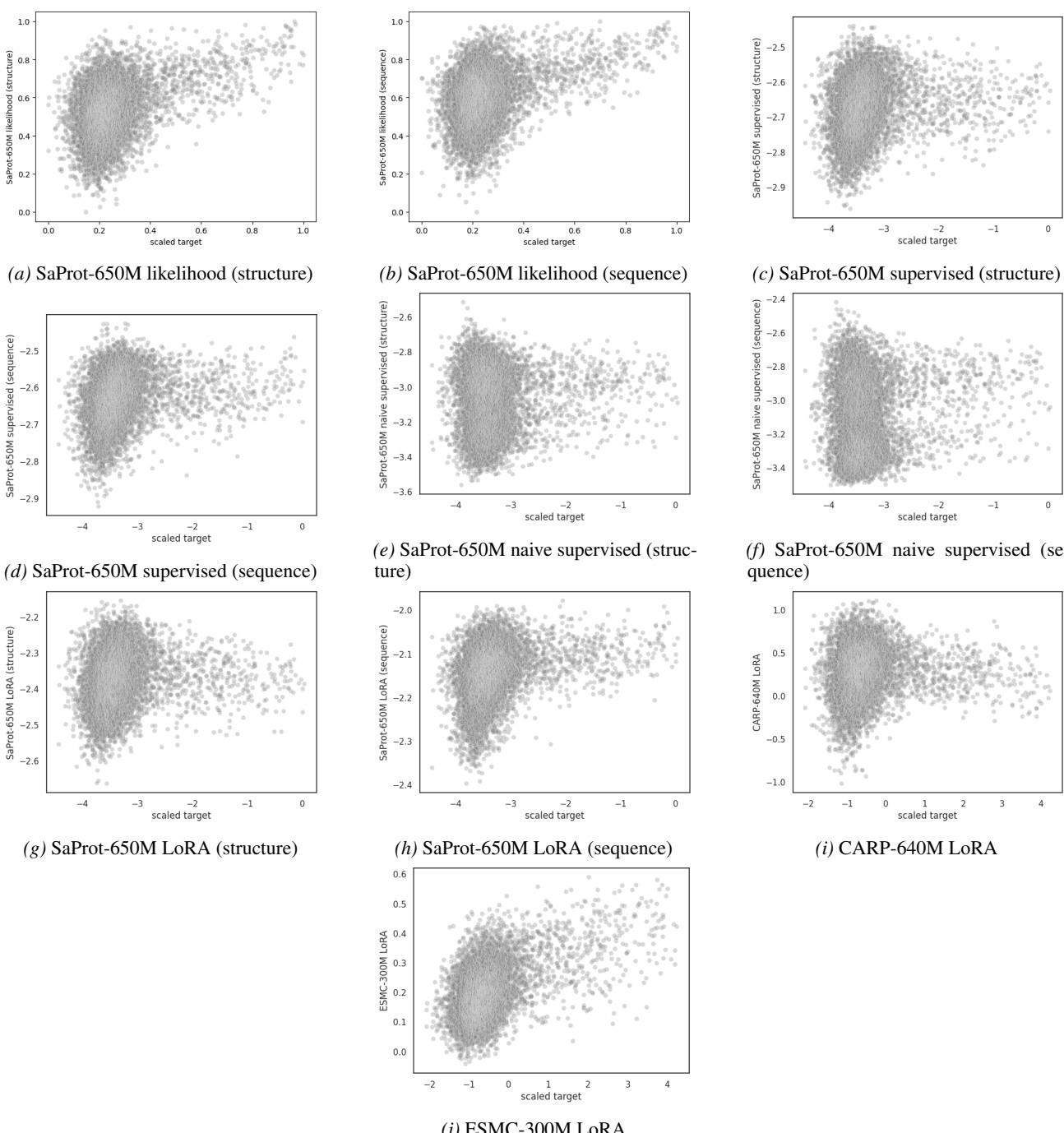

*(a)* SaProt-650M likelihood (structure)

*(b)* SaProt-650M likelihood (sequence)

*(c)* SaProt-650M supervised (structure)

*(d)* SaProt-650M supervised (sequence)

*(e)* SaProt-650M naive supervised (structure)

*(f)* SaProt-650M naive supervised (sequence)

*(g)* SaProt-650M LoRA (structure)

*(h)* SaProt-650M LoRA (sequence)

*(i)* CARP-640M LoRA

*(j)* ESMC-300M LoRA

*Figure A28.* Predictions for hydrophobic core to-P01053 (continued). Finetuned PLM predictions are averaged over 5 random seeds.

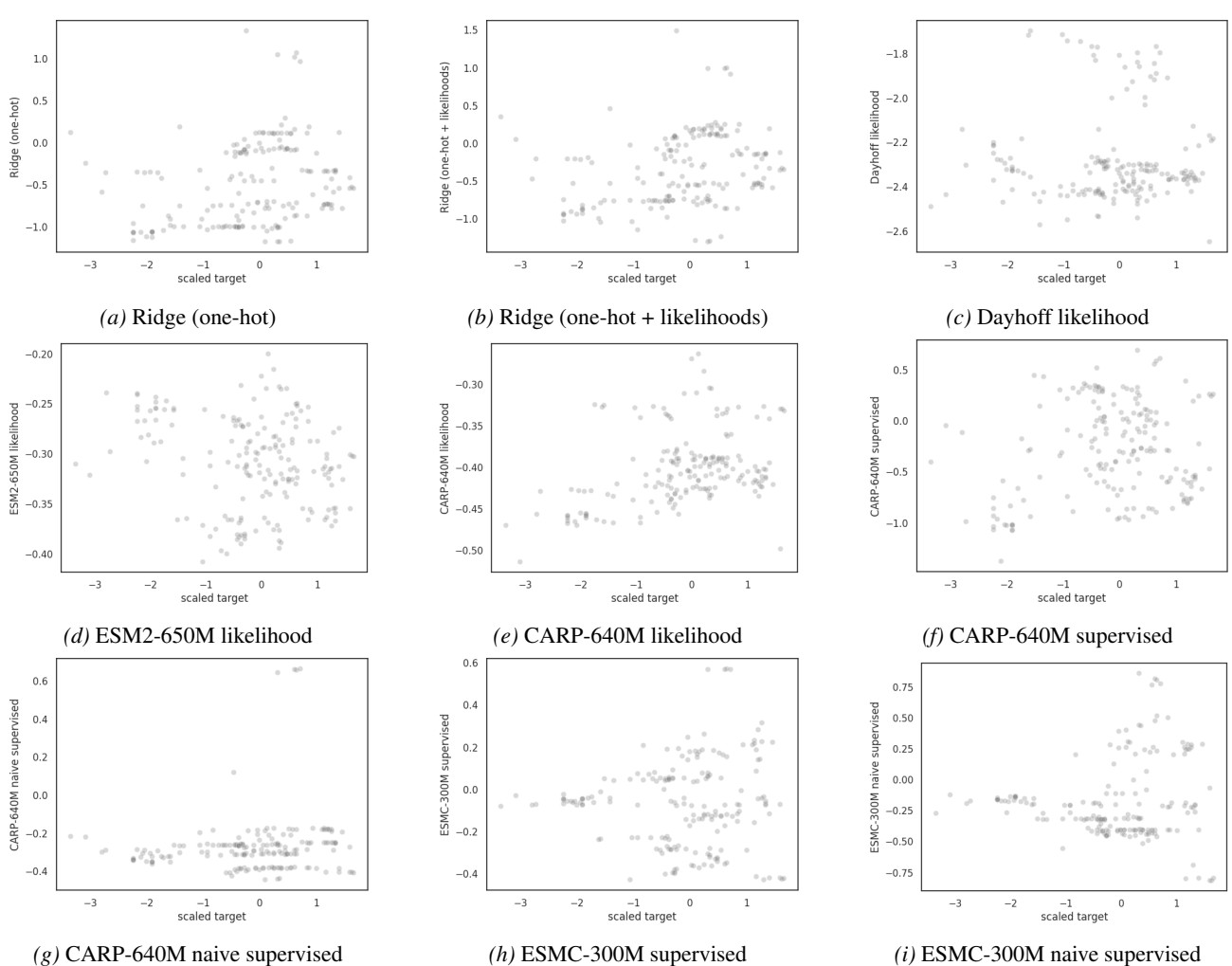

*(a)* Ridge (one-hot)     *(b)* Ridge (one-hot + likelihoods)     *(c)* Dayhoff likelihood

*(d)* ESM2-650M likelihood     *(e)* CARP-640M likelihood     *(f)* CARP-640M supervised

*(g)* CARP-640M naive supervised     *(h)* ESMC-300M supervised     *(i)* ESMC-300M naive supervised

*Figure A29.* Predictions for Rhomax by-wild-type. Finetuned PLM predictions are averaged over 5 random seeds.

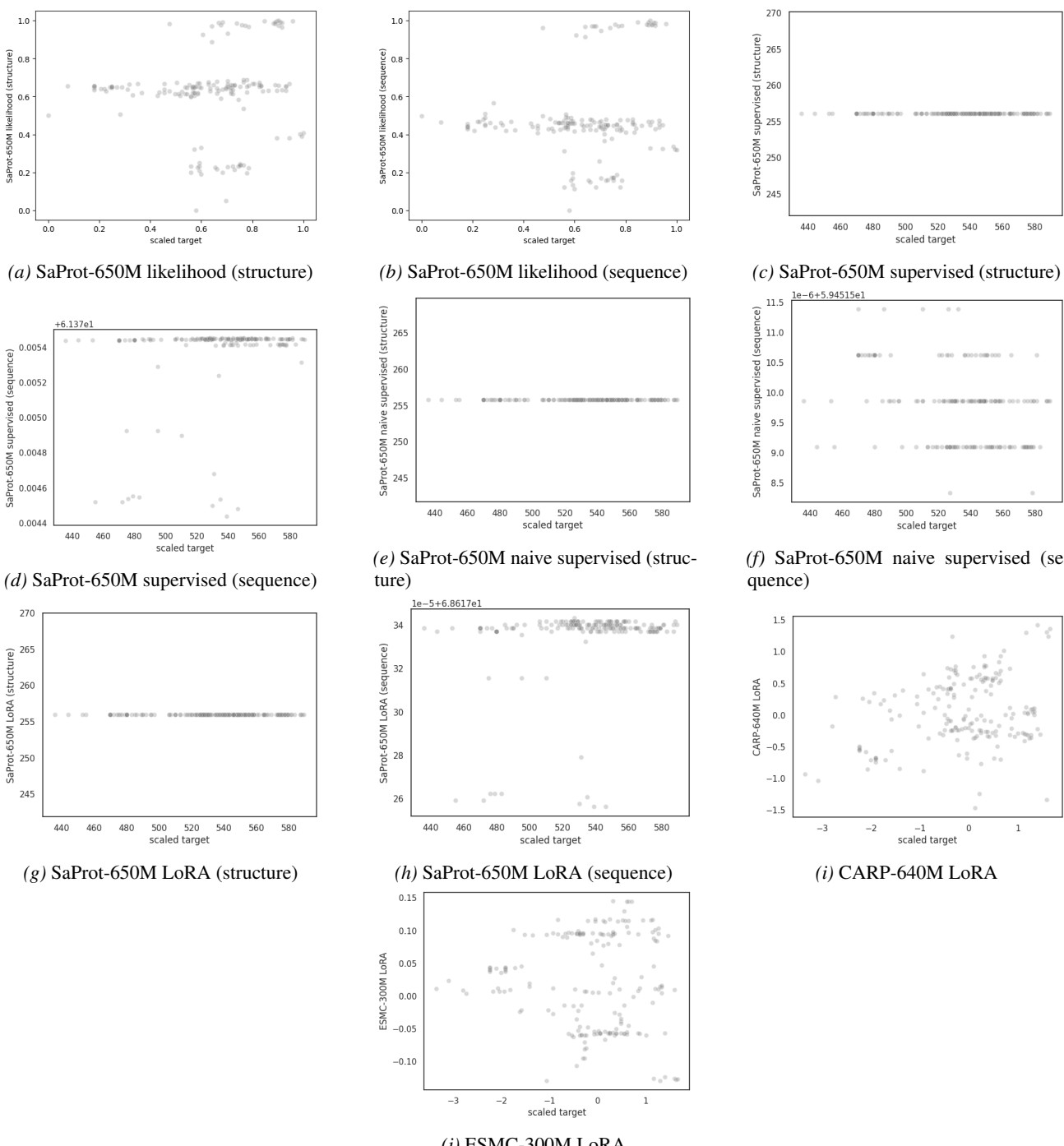

*(a)* SaProt-650M likelihood (structure)

*(b)* SaProt-650M likelihood (sequence)

*(c)* SaProt-650M supervised (structure)

*(d)* SaProt-650M supervised (sequence)

*(e)* SaProt-650M naive supervised (structure)

*(f)* SaProt-650M naive supervised (sequence)

*(g)* SaProt-650M LoRA (structure)

*(h)* SaProt-650M LoRA (sequence)

*(i)* CARP-640M LoRA

*(j)* ESMC-300M LoRA

*Figure A30.* Predictions for Rhomax by-wild-type (continued). Finetuned PLM predictions are averaged over 5 random seeds.

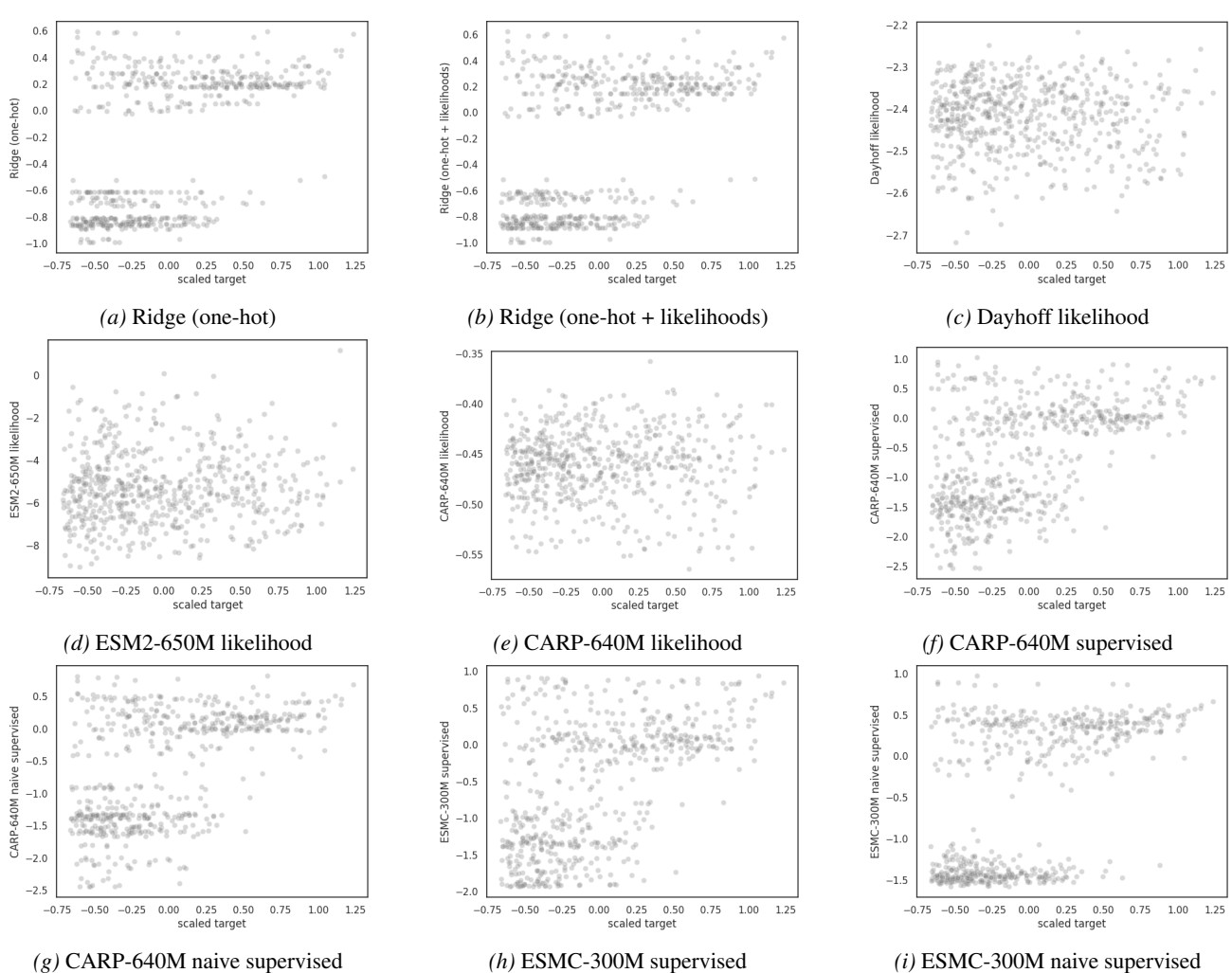

*(a)* Ridge (one-hot)  *(b)* Ridge (one-hot + likelihoods)  *(c)* Dayhoff likelihood

*(d)* ESM2-650M likelihood  *(e)* CARP-640M likelihood  *(f)* CARP-640M supervised

*(g)* CARP-640M naive supervised  *(h)* ESMC-300M supervised  *(i)* ESMC-300M naive supervised

*Figure A31.* Predictions for PDZ3 single-to-double. Finetuned PLM predictions are averaged over 5 random seeds.

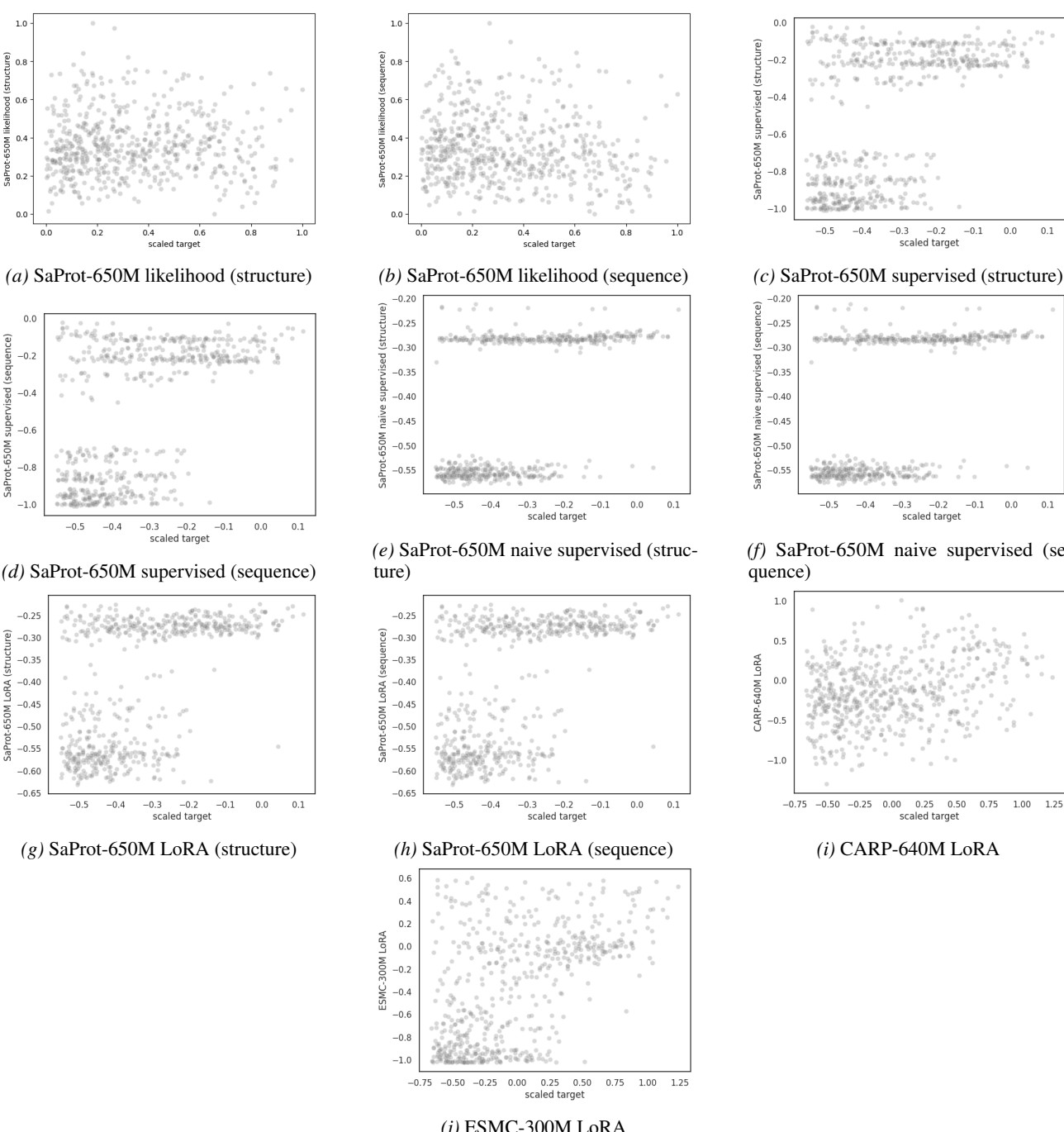

*(a)* SaProt-650M likelihood (structure)

*(b)* SaProt-650M likelihood (sequence)

*(c)* SaProt-650M supervised (structure)

*(d)* SaProt-650M supervised (sequence)

*(e)* SaProt-650M naive supervised (structure)

*(f)* SaProt-650M naive supervised (sequence)

*(g)* SaProt-650M LoRA (structure)

*(h)* SaProt-650M LoRA (sequence)

*(i)* CARP-640M LoRA

*(j)* ESMC-300M LoRA

*Figure A32.* Predictions for PDZ3 single-to-double (continued). Finetuned PLM predictions are averaged over 5 random seeds.

