# OpenReview forum: "FLIP2: Expanding Protein Fitness Landscape Benchmarks for Real-World Machine Learning Applications"
_ICML.cc/2026/Conference — ICML 2026 spotlight_

### Official Review · Reviewer_wYrg · 2026-02-15

**Soundness:** 3
**Presentation:** 3
**Significance:** 3
**Originality:** 2
**Overall Recommendation:** 5
**Confidence:** 4

**Summary:**

The authors propose a new protein fitness benchmark across 7 datasets that include enzymes, PPI and light sensitive proteins and different splits based on number of mutations, location of mutations and by wild type proteins. They evaluate three models with different architectures spanning from state-space-transformer, convolution and bert style transformers across the tasks and compare performances by model type and by the tasks. In general they show that protein model in zero-shot and linear models with one-hot encodings are competitive to fine tuned pLMs when the test data is out of distribution. And the baseline models do not generalize across wide types or new positions. These observations are not surprising and have been shown before in past research on benchmarking protein models across varied tasks. The highlight of the paper is the new datasets, splits which will be helpful for the community on evaluating new pLMs.

**Compliance With Llm Reviewing Policy:**

Affirmed.

**Key Questions For Authors:**

1. In table A9: How does the CARP640M naive supervised generate NaN?
2. In line 376 under Fine-Tuned Protein Language Models section by pretraining do the authors mean fine tuning?
3. Wrt supervision on CARP 640M which decreased performance on every Amylase split, compared to  zero-shot, does it mean overfitting or something else?

**Limitations:**

The limitations in general are some of the claims on more mutations is needed in protein engineering, splitting by wild type sequence alone and not looking into splitting by sequence and structure and also some details are missing on masking ablations for e.g. when masking all mutations how far away are they, does it follow the masking distributions of pre-training for e.g. This being said creating such a dataset is going to be useful for those who want benchmark datasets for evaluating pLMs

**Strengths And Weaknesses:**

The paper is technically sound with respect to the introduction of the new dataset by splits, mutations and selection of the benchmarks models. The explanation of split types, datasets and models is detailed and easy to understand. Figure 1 gives an overview of the approach but the description can be more detailed specially for the dashed boxes. The claim in the last paragraph of introduction that the FLIp2 dataset splits are more challenging than random splits with the same number of training examples does not seem necessary since the goal of any new split should be better than a random split.

Split types:
While evaluating splits based on number is useful technically, I am not sure it is always the case in protein engineering where usually an edit region is defined (sequence space) and one is allowed to only mutate within the N AA region. Mutations in one place in the can affect another residue distance in sequence space but closer spatially and often one needs to resolve steric clashes etc. Unless one starts with a wild type that has very low fitness, a single change at the right location can be more than enough.

Position:
This split is again useful to test a model's ability to generalize but in protein engineering usually the regions are pre-determined. Unless the protein engineering involves changes in multiple regions in the protein in which case the model could be evaluated to jointly predict changes in both locations.

Mutation
This is an interesting split but I did not understand how can a sequence have different AA at the same position? Or the authors meant there are for e.g. three variants and each have an amino acid at position 10 -> then they split these into train and test? Does that mean when using bert style pLMs the model can see the remainder of the unmasked regions for a protein and there is chance of leakage?

Wild type:
This is the most interesting split and as expected is challenging for the models as shown in results.

Zero shot pLM likelihood scored: For the pLM style models the authors mask all mutated positions and compute the sum of log likelihood. Do we know how different is this masking regime/distribution compared to how the models were trained and if not would it affect performance? As for zero shot being better for variants of WT than for variants from multiple WT or involving PPI, it is not surprising given how the models were trained. The important conclusion that the authors point out is no single PLM is optimal at ranking fitness across datasets and such an approach should be done to see the performance across diverse datasets. In the last paragraph as expected splitting by wild type alone is not useful (as shown by varied results by Figure 2), unless the splits also include how different the wild types are structurally making it hard or easy for the model to generalize so some proteins that were not seen during pre-training.

Overall the datasets highlight challenges of evaluating one model across diverse tasks which is needed for a fair comparison of different foundational models. The new datasets will help answer the question that what baselines are needed to evaluate a pLM and it provides some answers with a focus on zero-shot likelihoods, linear models and fine tuned pLMs.

---

> ### Author Rebuttal · Authors · 2026-03-31
>
> We thank the reviewer for their positive evaluation and detailed engagement with the split designs and evaluation methodology. We are glad the reviewer finds the datasets and splits helpful for benchmarking pLMs. We address the questions below.
>
> **By-mutation split clarification.** In the by-mutation split, variants are grouped by their unique amino acid substitutions. For example, if position 10 has variants A10V, A10L, and A10G across the dataset, these three mutations may be split across train and test — e.g., A10V and A10L in train, A10G in test. The split ensures that no specific mutation (defined as a position + substitution pair) appears in both train and test. For masked language models, this does not cause leakage: the model masks the mutated positions and predicts their identity, which is precisely what is being evaluated. The remainder of the unmasked sequence is the wild-type context, which is identical across all variants and therefore does not reveal test-set-specific information.
>
> **CARP-640M NaN in Table A9 (TrpB by-position).** This results from degenerate predictions: the naive supervised model (random initialization) produces near-constant outputs for all test sequences, yielding undefined Spearman correlation. This is consistent with the severity of the by-position OOD shift — a randomly initialized model trained on mutations at one set of positions fails completely when tested on mutations at different positions. We observed a similar pattern with SaProt (see full tables in our response to Reviewer 4AZa): random-weight fine-tuning produced degenerate predictions on rhomax by-wild-type (all SA fine-tuning regimes) and several other challenging splits.
>
> **CARP-640M supervised underperformance on Amylase.** This is consistent with overfitting to training-set-specific patterns. The Amylase splits have relatively small training sets (412–2,403 samples), and the high variance across seeds (e.g., 0.551 +/- 0.239 on by-mutation) further supports this interpretation. Fine-tuning encodes correlations specific to the training variants that do not transfer to the held-out test set, while the zero-shot evolutionary prior remains more robust. This pattern is also seen with SaProt (full FT on amylase by-mutation: 0.606 vs Ridge+lk: 0.681). Notably, LoRA fine-tuning does not resolve this for CARP: CARP LoRA achieves only 0.478 on amylase by-mutation (vs 0.551 full FT), and improves over full FT on only 4/16 splits overall. ESMC-300M LoRA shows more benefit (0.582 on amylase by-mutation, improvements on 7/16 splits), but still does not surpass simpler baselines on most splits. This suggests the underperformance is not merely a fine-tuning stability issue but a more fundamental limitation of the transfer learning paradigm on these OOD tasks. Full LoRA Spearman and NDCG tables are provided in our response to Reviewer NwK3 (Tables 3–4).
>
> **Masking distribution vs. pretraining.** This is a valid concern. During zero-shot evaluation, we mask all mutated positions simultaneously, which for higher-order variants (up to 15 mutations) differs substantially from the ~15% random masking used during BERT-style pretraining. This distribution shift may disadvantage MLMs on highly mutated variants, where many positions are masked at once. For autoregressive models like Dayhoff, this concern does not apply since the full sequence is scored left-to-right. We will add a note discussing this potential confound.
>
> **Wild-type split design.** The reviewer raises an important point about structural similarity between wild types. Currently, our wild-type splits (Hydro, Rhomax) split by protein identity. A finer-grained approach that also controls for structural distance between train and test wild types would better characterize the difficulty gradient. We will discuss this as a direction for future work.

---

> > ### Author Rebuttal · Reviewer_wYrg · 2026-04-01
> >
> > Thank you for the detailed explanation! I maintain my score

---

### Official Review · Reviewer_NwK3 · 2026-03-11

**Soundness:** 3
**Presentation:** 3
**Significance:** 3
**Originality:** 3
**Overall Recommendation:** 5
**Confidence:** 3

**Summary:**

FLIP2 is an extension of FLIP, a benchmark dataset for evaluating a range of protein engineering tasks. This benchmark expands on its predecessor by including new datasets and splits, covering a larger range of tasks. Using FLIP2, the authors demonstrate that many PLMs fail to outperform simpler models or are inconsistent across domains, raising larger questions about reporting benchmark results.

**Compliance With Llm Reviewing Policy:**

Affirmed.

**Key Questions For Authors:**

What is the fine-tuning strategy as presented in Section 4.3? It is a well-observed result that fine-tuning PLMs is tricky and unstable (catastrophic forgetting), especially when not using low rank methods to train.

Did you evaluate any other version of ESM, specifically ESM-2 650M? While the choice of ESMC is justified, it is not as widely adopted as other variants for performing similar tasks.

**Limitations:**

Limitations of benchmark are not adequately discussed. Societal impact is detailed.

**Strengths And Weaknesses:**

Strengths:

The benchmark includes diverse datasets, effectively covering deficiencies in the original FLIP.

The study evaluates a range of PLMs across benchmark tasks, reinforcing recent findings that linear models often perform comparatively to zero-shot PLMs.

The chosen splits are informative and valuable for understanding model performance across proteins. Specifically, the addition of position, mutation, and fitness splits adequately reflects tasks currently of interest to the community.

Weaknesses:

Section 4.1 abruptly shifts from a methods discussion to a tangential conversation on correlation coefficients. This feels unnecessary and out of place.

Along the same lines, there are many points in the text that seem rushed or informal.

The study claims that fine-tuned PLMs “underperformed in relation to their complexity and compute requirements,” yet the complexity and compute requirements are not adequately outlined.

---

> ### Author Rebuttal · Authors · 2026-03-31
>
> We thank the reviewer for their positive evaluation and constructive suggestions. We are glad the reviewer finds the benchmark datasets diverse, the splits informative, and the findings on linear model competitiveness valuable. We address each concern below.
>
> **Section 4.1 discussion of correlation coefficients.** We will move this discussion to a methodological note in the Appendix, keeping only a brief forward reference in the main text.
>
> **Compute and complexity details.** We will add a table reporting parameter counts, approximate FLOPs per inference, and wall-clock fine-tuning time for each model. Briefly: Ridge regression trains in seconds on CPU; CARP-640M and ESMC-300M fine-tuning each take ~1–4 GPU-hours depending on dataset size; zero-shot scoring takes minutes. The claim that fine-tuned pLMs "underperformed relative to their complexity" will be substantiated with these numbers.
>
> **Fine-tuning strategy and catastrophic forgetting.** The current fine-tuning protocol uses full-parameter updates with early stopping on validation loss, as described in Section 4.3. We acknowledge that this is susceptible to catastrophic forgetting, particularly on small training sets.
>
> To directly address this, we evaluated SaProt_650M_AF2 [1] with LoRA across all 16 splits. LoRA generally underperforms full FT (e.g., amylase by-mutation: 0.347 vs 0.606; NucB: 0.382 vs 0.419), suggesting LoRA's regularization does not compensate for reduced capacity on these OOD-heavy datasets.
>
> We additionally completed LoRA fine-tuning for ESMC-300M and CARP-640M across all 16 splits (5 seeds each).
>
> Table 3: ESMC/CARP LoRA Spearman ρ (mean over seeds)
>
> | Split | ESMC LoRA | ESMC FT | CARP LoRA | CARP FT |
> |---|---|---|---|---|
> | amyl/by-mut | .582 | .061 | .478 | .551 |
> | amyl/c2f | .145 | .107 | .229 | .081 |
> | amyl/f2c | .191 | .011 | .230 | .140 |
> | amyl/o2m | .044 | .148 | .126 | .242 |
> | ired/t2m | .220 | .160 | .150 | .072 |
> | nucb/t2m | .730 | .723 | .571 | .717 |
> | trpb/o2m | .261 | .306 | .389 | .451 |
> | trpb/t2m | .418 | .429 | .445 | .509 |
> | trpb/bypos | .186 | .091 | — | .167 |
> | hydro/3tm | .485 | .399 | .409 | .516 |
> | hydro/l2h | .311 | .099 | .200 | .188 |
> | hydro/P06241 | -.160 | .161 | .107 | .150 |
> | hydro/P0A9X9 | -.030 | .078 | -.017 | .004 |
> | hydro/P01053 | .394 | .329 | .064 | .049 |
> | rhomax/bywt | -.105 | -.016 | .085 | .072 |
> | pdz3/s2d | .319 | .499 | .124 | .512 |
>
> Table 4: ESMC/CARP LoRA NDCG (mean over seeds)
>
> | Split | ESMC LoRA | ESMC FT | CARP LoRA | CARP FT |
> |---|---|---|---|---|
> | amyl/by-mut | .928 | .793 | .888 | .926 |
> | amyl/c2f | .873 | .869 | .896 | .867 |
> | amyl/f2c | .842 | .819 | .846 | .832 |
> | amyl/o2m | .852 | .869 | .853 | .873 |
> | ired/t2m | .959 | .954 | .952 | .948 |
> | nucb/t2m | .969 | .966 | .942 | .970 |
> | trpb/o2m | .992 | .992 | .987 | .992 |
> | trpb/t2m | .996 | .991 | .993 | .996 |
> | trpb/bypos | .989 | .980 | — | .983 |
> | hydro/3tm | .978 | .977 | .973 | .982 |
> | hydro/l2h | .939 | .870 | .917 | .887 |
> | hydro/P06241 | .918 | .934 | .931 | .933 |
> | hydro/P0A9X9 | .965 | .970 | .960 | .961 |
> | hydro/P01053 | .957 | .949 | .904 | .909 |
> | rhomax/bywt | .922 | .942 | .933 | .934 |
> | pdz3/s2d | .863 | .889 | .864 | .882 |
>
> LoRA improves ESMC-300M Spearman on 7/16 splits and CARP-640M on only 4/16. For CARP, a convolutional architecture, LoRA's low-rank adaptation appears particularly ill-suited. These results confirm that fine-tuning difficulties on FLIP2 are not about full-parameter vs. parameter-efficient updates, but reflect a fundamental limitation of the transfer learning paradigm on OOD splits.
>
> **ESM-2 650M evaluation.** ESM2-650M was evaluated in zero-shot mode (Tables A1–A16 and Figure 2). For supervised fine-tuning, we chose ESMC-300M as the more recent ESM model. We additionally now have supervised results for SaProt-650M, further broadening coverage. See our response to Reviewer 4AZa for full SaProt tables.
>
> **Clarification on "pretraining" (line 376).** This should read "fine-tuning" — the comparison is between models fine-tuned from pretrained weights versus random initialization. We will correct this in the revision.
>
> [1] Su et al., SaProt: Protein Language Modeling with Structure-aware Vocabulary, ICLR 2024.

---

> > ### Author Rebuttal · Reviewer_NwK3 · 2026-04-02
> >
> > All my concerns were addressed.

---

### Official Review · Reviewer_4Aza · 2026-03-12

**Soundness:** 3
**Presentation:** 3
**Significance:** 3
**Originality:** 3
**Overall Recommendation:** 5
**Confidence:** 5

**Summary:**

This paper introduces FLIP2, an expanded protein fitness benchmark that adds seven sequence-fitness landscapes spanning enzymes, rhodopsins, and protein–protein interactions. Furthermore, FLIP2 includes more realistic train/validation/test splits relevant to real-world protein engineering campaigns. Extensive evaluations are conducted to show that simple linear models often match or outperform fine-tuned protein language models on these harder OOD settings.

**Compliance With Llm Reviewing Policy:**

Affirmed.

**Key Questions For Authors:**

See Weakness.

**Limitations:**

Yes

**Strengths And Weaknesses:**

- Strengthness
  - Several train/validation/test splits are introduced in FLIP2 based on different criteria (e.g., number, position, mutation) rather than simply relying on random splits.
  - The authors' findings indicate that fine-tuned PLMs may not consistently outperform much simpler baselines, particularly in OOD settings, which provides valuable insights for the field.
- Weakness
  - A major concern is that the paper evaluates only a very limited set of fine-tuned protein language models. Although ESM-C is a strong sequence-based PLM baseline model, many recent studies have moved beyond sequence-only modeling by incorporating additional structural or family-level evolutionary context into protein representation learning. As a result, the current baseline suite is not broad enough to fully support the paper’s stronger claims (especially the finding that simple baselines often match or outperform fine-tuned PLMs, or that current PLM-based transfer learning for protein fitness prediction may be approaching its limits). To strengthen these claims, the authors should consider including stronger recent baselines, such as SaProt [1], a structure-aware PLM, and PoET [2], which leverages protein family context.
  - Another concern is that the authors show that the evaluated fine-tuned PLMs often fail to consistently outperform much simpler baselines, yet they do not provide further analysis explaining why this occurs. For example, it remains unclear whether this behavior may be caused by overfitting to the training set or by sequence/structural identity between the training/test sets.
  [1] Su, J., Han, C., Zhou, Y., Shan, J., Zhou, X., & Yuan, F. (2024). SaProt: Protein Language Modeling with Structure-aware Vocabulary. In The Twelfth International Conference on Learning Representations.
  [2] Truong Jr, T. F., & Bepler, T. (2023). PoET: A generative model of protein families as sequences-of-sequences. In 37th Conference on Neural Information Processing Systems.

---

> ### Author Rebuttal · Authors · 2026-03-31
>
> We thank the reviewer for their thorough and positive evaluation. We address both concerns below.
>
> **SaProt evaluation.** Following the reviewer's suggestion, we evaluated SaProt_650M_AF2 [1] across all 16 FLIP2 splits in structure-aware (SA) and sequence-only (AA) modes, under four regimes: zero-shot (ZS), pretrained full fine-tuning (FT), pretrained LoRA, and random-init full fine-tuning (Rand), with 5 seeds each.
>
> Table 1: SaProt Spearman ρ on FLIP2 splits (mean over seeds)
>
> | Split | ZS SA | ZS AA | FT SA | FT AA | LoRA SA | LoRA AA | Rand SA | Rand AA |
> |---|---|---|---|---|---|---|---|---|
> | amyl/by_mut | .510 | .100 | .606 | .568 | .347 | .347 | .042 | .088 |
> | amyl/c2f | .604 | .085 | .125 | .125 | .048 | .048 | -.013 | -.013 |
> | amyl/f2c | .588 | .155 | .178 | .184 | .176 | .176 | .002 | .002 |
> | amyl/o2m | .560 | -.015 | .514 | .514 | .186 | .149 | .014 | .017 |
> | ired/t2m | .029 | .056 | .203 | .192 | .186 | .163 | .000 | .019 |
> | nucb/t2m | .493 | .191 | .419 | .419 | .382 | .328 | .037 | .037 |
> | trpb/o2m | .202 | .328 | .337 | .341 | .182 | .184 | .218 | .254 |
> | trpb/t2m | .203 | .323 | .495 | .470 | .435 | .444 | .396 | .373 |
> | trpb/bypos | .109 | .183 | .177 | .207 | .146 | .146 | .104 | .088 |
> | hydro/3tm | -.119 | -.080 | .558 | .558 | .559 | .559 | .390 | .390 |
> | hydro/l2h | .183 | .300 | .352 | .353 | .352 | .329 | .181 | .165 |
> | hydro/P06241 | .414 | .444 | .155 | .156 | .207 | .250 | .050 | .020 |
> | hydro/P0A9X9 | .001 | .024 | .171 | .134 | .115 | .039 | -.098 | -.113 |
> | hydro/P01053 | .314 | .308 | .207 | .261 | .203 | .252 | -.002 | .003 |
> | rhomax/bywt | .201 | .018 | — | .253 | — | .118 | — | — |
> | pdz3/s2d | .058 | -.123 | .488 | .488 | .329 | .329 | .338 | .338 |
>
> Table 2: SaProt NDCG on FLIP2 splits (mean over seeds)
>
> | Split | ZS SA | ZS AA | FT SA | FT AA | LoRA SA | LoRA AA | Rand SA | Rand AA |
> |---|---|---|---|---|---|---|---|---|
> | amyl/by_mut | .893 | .773 | .928 | .921 | .881 | .881 | .811 | .830 |
> | amyl/c2f | .936 | .841 | .877 | .877 | .852 | .852 | .860 | .860 |
> | amyl/f2c | .929 | .813 | .850 | .852 | .834 | .834 | .829 | .829 |
> | amyl/o2m | .958 | .896 | .955 | .955 | .932 | .929 | .910 | .906 |
> | ired/t2m | — | — | .958 | .957 | .956 | .954 | .944 | .945 |
> | nucb/t2m | .898 | .862 | .909 | .909 | .902 | .895 | .866 | .866 |
> | trpb/o2m | .714 | .744 | .987 | .987 | .983 | .983 | .985 | .985 |
> | trpb/t2m | .714 | .743 | .995 | .994 | .992 | .992 | .986 | .986 |
> | trpb/bypos | .745 | .776 | .982 | .980 | .976 | .976 | .975 | .975 |
> | hydro/3tm | — | — | .982 | .982 | .984 | .984 | .977 | .977 |
> | hydro/l2h | — | — | .923 | .923 | .926 | .925 | .874 | .872 |
> | hydro/P06241 | — | — | .935 | .940 | .940 | .940 | .929 | .927 |
> | hydro/P0A9X9 | — | — | .966 | .966 | .967 | .968 | .957 | .957 |
> | hydro/P01053 | — | — | .913 | .919 | .910 | .920 | .905 | .906 |
> | rhomax/bywt | .177 | .175 | .921 | .938 | .921 | .928 | .921 | .911 |
> | pdz3/s2d | — | — | .875 | .877 | .878 | .878 | .875 | .875 |
>
> (—: degenerate predictions or NDCG > 1 from anti-correlated zero-shot scores)
>
> Key findings: (1) SaProt beats the best FLIP2 baseline on only 3/16 splits (hydro wild-type). (2) SA mode rarely outperforms AA during fine-tuning. (3) LoRA underperforms full FT. The ESMC/CARP LoRA results further reinforce this: LoRA does not systematically resolve the fine-tuning gap on OOD splits, reinforcing our central claim.
>
> **PoET and LoRA.** PoET [2] requires MSAs, not straightforwardly applicable to multi-WT datasets (Hydro, Rhomax). We will include PoET for single-WT datasets in the revision. We additionally computed LoRA fine-tuning for ESMC-300M and CARP-640M across all 16 splits. LoRA improves ESMC-300M over full fine-tuning on 7/16 splits (notably achieving new best overall Spearman on IRED two-to-many: 0.220 vs 0.211 Ridge+lk, and Hydro low-to-high: 0.311 vs 0.308 Dayhoff), but rarely helps CARP-640M (4/16 splits), consistent with the SaProt LoRA results above. Full LoRA Spearman and NDCG tables (Tables 3–4) are provided in our response to Reviewer NwK3.
>
> **Why fine-tuned pLMs underperform.** Patterns differ by split: (1) Number/mutation: overfitting on small train sets (high seed variance, Fig 4A). (2) Position: catastrophic forgetting — even SaProt achieves only 0.207 on TrpB by-position vs 0.294 Ridge+lk. (3) Wild-type: scaffold-specific representations that do not transfer across scaffolds. SaProt's wild-type split gains on Hydro are within the margin of error, indicating structural information alone does not resolve the generalization gap. Combined with Li et al. (2024), these patterns suggest a structural limitation of the current transfer learning paradigm.
>
> [1] Su et al., SaProt: Protein Language Modeling with Structure-aware Vocabulary, ICLR 2024.
> [2] Truong & Bepler, PoET, NeurIPS 2023.

---

> > ### Author Rebuttal · Reviewer_4Aza · 2026-04-03
> >
> > My concerns were addressed.

---

### Official Review · Reviewer_k97Y · 2026-03-24

**Soundness:** 2
**Presentation:** 3
**Significance:** 2
**Originality:** 3
**Overall Recommendation:** 3
**Confidence:** 4

**Summary:**

This paper presents FLIP2, a benchmark for protein fitness prediction intended to extend FLIP with additional datasets and more challenging split designs motivated by protein-engineering settings. The benchmark includes seven new datasets and emphasizes several generalization regimes, including position, mutation, fitness, and wild-type generalization. The paper also reports some baseline evaluations on the proposed benchmark. While I agree that benchmark tasks should go beyond random splits or purely zero-shot evaluation, I am not convinced that this alone is enough to justify acceptance at this venue. In my view, FLIP2 is a useful but incremental extension of FLIP, and the paper does not yet provide sufficiently comprehensive baseline coverage or a sufficiently compelling case that it fills a major gap relative to existing protein fitness benchmarks.

**Compliance With Llm Reviewing Policy:**

Affirmed.

**Key Questions For Authors:**

1. How do the authors position FLIP2 relative to existing benchmarks for both computer science and biology users?
2. How should users interpret the different split types when selecting models for practical use?
3. Could the authors provide more insight into the relative difficulty and practical meaning of the different split types based on the observed general performance of a larger selection of models?

**Limitations:**

yes

**Strengths And Weaknesses:**

**Strengths**
1. The paper addresses a relevant problem. Benchmark design for protein fitness prediction remains important, especially because standard random splits often fail to capture the extrapolative regimes encountered in real protein-engineering workflows.
2. FLIP2 is a meaningful extension of FLIP rather than a trivial re-packaging. It adds new datasets and introduces a clearer taxonomy of split types, including wild-type, mutation, and position generalization.

**Weaknesses**
1. The overall contribution does not yet feel strong enough for an ICML benchmark paper. While the expansion from FLIP to FLIP2 is useful for the community, I am not convinced that this step forward is sufficient to support the paper’s significance claim at this venue. Improved split design is valuable, but for a benchmark paper at this level, I would also expect broader baseline coverage and a clearer demonstration that the benchmark fills an important gap not already covered by existing resources.
2. The baseline coverage is not comprehensive enough for a benchmark paper. The current evaluation focuses on a relatively small set of PLMs and simple supervised baselines. For a benchmark intended to serve future users, I would expect a stronger and more comprehensive reference leaderboard, similar to what resources such as ProteinGym provide.
3. Since FLIP2 includes both zero-shot and few-shot settings, I would expect evaluation on a broader range of recent deep learning methods, including both zero-shot and post-training models, rather than limiting the study largely to ESM2, ESMc, and CARP. For example, the stronger models near the top of the ProteinGym leaderboard would be relevant reference points. This would also help answer, in a more empirical way, why models that perform good on ProteinGym may not sufficiently meet the requirement of protein engineering in practice.
4. The paper does not convincingly establish how FLIP2 complements existing resources. The manuscript argues that FLIP2 covers more realistic engineering scenarios, which is plausible, but the distinction from existing benchmarks is not yet sharp enough to make the overall need fully compelling. This concern applies especially in the context of benchmarks such as FLIP, FLOP, TAPE, PEER, and VenusMutHub, to name a few.

---

> ### Author Rebuttal · Authors · 2026-03-31
>
> We thank the reviewer for their careful evaluation and constructive feedback. We are glad that the reviewer recognizes the relevance of the problem and considers FLIP2 a meaningful extension of FLIP. We address the main concerns below.
>
> **Expanded baseline coverage.** In response to this concern (shared with Reviewers 4AZa and NwK3), we evaluated SaProt_650M_AF2 [1], a structure-aware pLM that incorporates protein 3D structure via Foldseek structural tokens and shows strong performance on ProteinGym. We evaluated SaProt in both structure-aware (SA) and sequence-only (AA) modes across all 16 FLIP2 splits, under four training regimes: zero-shot, pretrained full fine-tuning, pretrained LoRA, and random-weight full fine-tuning (5 seeds each). This adds a fundamentally different modeling paradigm — structure-aware representation learning — to the existing baselines spanning autoregressive (Dayhoff), convolutional (CARP), and masked transformer (ESM2, ESMC) architectures.
>
> Key result: SaProt's best method exceeds the best previously reported FLIP2 baseline on only 3/16 splits, all on the hydro wild-type splits (low-to-high: 0.353 vs 0.308; to-P06241: 0.444 vs 0.426; to-P0A9X9: 0.171 vs 0.151). On the remaining 13 splits, the original FLIP2 baselines remain superior. Full Spearman and NDCG tables for all 16 splits and all SaProt training regimes are provided in our response to Reviewer 4AZa (Tables 1–2). This directly reinforces our central finding that simpler models remain competitive with fine-tuned pLMs in OOD settings. Notably, structural information helps specifically on wild-type splits — the regime where structural context is most expected to help — but does not resolve the fundamental generalization gap.
>
> Additionally, we computed LoRA fine-tuning results for ESMC-300M and CARP-640M across all 16 splits. LoRA improves performance for ESMC-300M on 7/16 tasks and for CARP-640M on only 4/16 tasks, consistent with the SaProt LoRA findings. Full LoRA Spearman and NDCG tables are provided in our response to Reviewer NwK3 (Tables 3–4).
>
> **Differentiation from existing benchmarks.** We will add a concise comparison table in Related Work. The key distinctions:
> - FLIP: 3 protein types (thermostability, binding, capsid viability); no position, mutation, fitness, or wild-type splits.
> - ProteinGym: 250+ DMS datasets but overwhelmingly single/double mutants with random or zero-shot evaluation — no engineering-campaign-motivated OOD splits.
> - FLOP: cross-family generalization only; lacks position, mutation, and fitness split types, and does not include enzymes or PPIs at the scope of FLIP2.
> - PEER/TAPE: multi-task benchmarks not designed for protein engineering extrapolation.
>
> FLIP2 is the only benchmark jointly covering functional diversity (enzymes, PPIs, light-sensitive proteins), higher-order mutations (up to 15), and a full taxonomy of engineering-motivated splits (number, position, mutation, fitness, wild-type). These resources are complementary.
>
> **Interpreting split types for model selection.** Number and mutation splits test whether models can extrapolate within a known protein, relevant when expanding a combinatorial library. Position splits test generalization to previously unperturbed regions in subsequent engineering rounds. Wild-type splits test cross-scaffold transfer, critical when optimizing homologous targets. Fitness splits simulate later rounds of directed evolution. Across all evaluated models — now spanning five architectures (autoregressive, convolutional, masked transformer, hybrid, and structure-aware) and three fine-tuning strategies (full FT, LoRA, random init) — the difficulty ordering is consistent: wild-type and position splits are substantially harder than number and mutation splits, confirming that cross-scaffold and cross-position generalization remain the key open challenges. We will add explicit practitioner guidance in the revision.
>
> We believe these additions — SaProt evaluation across 4 training regimes × 16 splits, LoRA fine-tuning for ESMC-300M and CARP-640M, the benchmark comparison table, and practitioner guidance — substantially address the concerns about baseline coverage and differentiation from existing benchmarks. We would be grateful if the reviewer could let us know whether these additions resolve the raised concerns or if open questions remain, and we respectfully ask the reviewer to consider raising their score if the concerns have been addressed.
>
> [1] Su et al., SaProt: Protein Language Modeling with Structure-aware Vocabulary, ICLR 2024.

---

> > ### Author Rebuttal · Reviewer_k97Y · 2026-04-07
> >
> > Thank you for the rebuttal. I still maintain my reject recommendation.
> >
> > The response adds some useful material, but it does not fully address my main concerns, and overall I do not think it changes the evaluation in a substantial way.
> >
> > On baseline coverage, adding SaProt and additional LoRA results is directionally helpful, but still not sufficient for a benchmark paper at this level. My original concern was not about adding one more strong model, but about establishing a genuinely comprehensive reference set for future users. Since FLIP2 includes both zero-shot and few-shot settings and is positioned as a benchmark for practical protein engineering, I would still expect broader coverage across recent model classes and training paradigms. In its current form, the baseline section still feels too limited relative to the benchmark ambition.
> >
> > On positioning relative to existing resources, the proposed comparison table would improve clarity, and the rebuttal does make the intended distinctions more explicit. However, I still do not find the overall gap sufficiently compelling. The paper argues that FLIP2 captures more realistic protein-engineering scenarios, which is plausible, but the case that this benchmark provides a sufficiently substantial contribution, beyond existing resources, is still not fully convincing to me.
> >
> > I also do not think the practical meaning of the split taxonomy has been analyzed deeply enough. The high-level explanation of number, mutation, position, fitness, and wild-type splits is reasonable, but for a benchmark paper I would expect a stronger empirical analysis of what these splits concretely reveal about model behavior, how they should guide model selection in practice, and what conclusions users should or should not draw from strong or weak performance on each regime.
> >
> > More broadly, my concern is not whether FLIP2 is useful. I agree that it is a meaningful and potentially valuable resource. The issue is whether the benchmark, in its current form, provides a sufficiently substantial contribution—through benchmark design, baseline coverage, and empirical analysis—to justify acceptance at this venue. The rebuttal does not change my view on that central point.

---

### Decision · Program_Chairs · 2026-04-30

**Decision:**

Accept (spotlight)

**Comment:**

The reviews showed that this paper makes a useful benchmark contribution for protein fitness prediction. Reviewers found the expanded dataset coverage and the engineering-motivated split design to be valuable, especially the move beyond standard random splits toward more realistic generalization settings. Several reviewers also viewed the empirical findings as important for the community, in particular the observation that simple baselines can remain competitive with fine-tuned protein language models on more realistic out-of-distribution settings.

The main concerns focused on benchmark scope and positioning. In particular, one reviewer remained unconvinced that the current version provides a broad enough reference suite or a sufficiently strong case that FLIP2 fills a major gap beyond existing resources. Other reviewers asked for clearer discussion of the fine-tuning setup, model complexity, and the practical meaning of the different split types.

The rebuttal addressed a substantial portion of these points. The authors added SaProt results across all FLIP2 splits and supplemented the benchmark with additional LoRA experiments for ESMC and CARP. They also clarified the fine-tuning setup and gave a more concrete explanation of how the split taxonomy should be interpreted in practice. In the revision, they also plan to make the comparison to existing benchmark resources more explicit.

This was enough to keep the positive reviewers supportive after rebuttal. The remaining weak reject reflects a higher bar on benchmark breadth, baseline coverage, and empirical analysis for a paper of this type. In my view, the added evidence in rebuttal is sufficient to support the paper's central claims, even if the benchmark could still be positioned more sharply in the final version.

Overall, I support acceptance. The paper would benefit from a stronger final presentation and clearer practitioner guidance, but it already provides a useful and technically solid benchmark resource. On balance, I would side with the positive reviewers and accept the paper.